# Online Robust Reinforcement Learning with General Function Approximation

**Debamita Ghosh** [1]  **George K. Atia** [1 2]  **Yue Wang** [1 2]

## Abstract

Reinforcement learning (RL) in real-world tasks often suffers from performance degradation due to distribution shift between training and deployment environments. Distributionally Robust RL (DR-RL) addresses this issue by optimizing the worst-case performance over an uncertainty set of transition dynamics, providing an optimized baseline performance upon deployment. However, existing methods typically require strong data access assumptions (e.g., a generative model or comprehensive offline datasets) and mostly focus on tabular settings. In this paper, we introduce a purely online DR-RL algorithm with general function approximation that learns a robust policy directly from interaction, without any prior knowledge or pre-collected data. Our method uses a dual-based fitted robust Bellman update to jointly learn the value function and the robust backup operator. We establish the first regret guarantee for online DR-RL in terms of an intrinsic complexity measure—the robust Bellman–Eluder (BE) dimension, for general $\phi$-divergence uncertainty sets. Our regret bound is sublinear and independent of $|\mathcal{S}|$ and $|\mathcal{A}|$, and recovers sharp rates in structured regimes, providing a scalable method for practical DR-RL.

## 1. Introduction

Reinforcement learning (RL) is a powerful framework for solving sequential decision-making tasks. As one of RL's fundamental training schemes, in the online learning setting, an agent learns an optimal policy through trial-and-error interactions with an unknown environment, without relying on pre-collected datasets or high-fidelity simulators. This paradigm has driven major successes in complex simulator-based domains, including video games (Silver et al., 2016; Zha et al., 2021; Berner et al., 2019; Vinyals et al., 2017) and generative AI (Ouyang et al., 2022; Cao et al., 2023; Black et al., 2023; Uehara et al., 2024; Zhang et al., 2024; Du et al., 2023; Cao et al., 2024). A core vulnerability of conventional online RL, however, is its implicit assumption that the environment's dynamics, though stochastic, remain fixed and do not change from training to deployment. In practice, however, this assumption is often violated due to, e.g., non-stationary environments or modeling errors. Policies learned by standard RL can be highly specialized to the training conditions and brittle to even small shifts, potentially leading to severe performance degradations. For instance, in autonomous driving (Kiran et al., 2021) or healthcare (Wang et al., 2018), an agent may face unanticipated changes such as reduced road friction due to weather, and a standard RL policy—never trained to account for such possibilities—can take unsafe or costly decisions.

The core issue is that vanilla online RL optimizes expected return under the training dynamics, without accounting for perturbations or model mismatch at deployment. DR-RL (Iyengar, 2005; Pinto et al., 2017; Hu et al., 2022) addresses this by optimizing worst-case performance over a prescribed uncertainty set of plausible transition models. This worst-case objective encourages agents to learn policies that are inherently resilient to environmental shifts, leading to more reliable and safer behavior when post-deployment conditions differ from those seen during training (Goodfellow et al., 2014; Vinitsky et al., 2020; Abdullah et al., 2019; Hou et al., 2020; Rajeswaran et al., 2017; Atkeson & Morimoto, 2003; Morimoto & Doya, 2005; Huang et al., 2017; Kos & Song, 2017; Lin et al., 2017; Pattanaik et al., 2018; Mandlekar et al., 2017). In *online* DR-RL (He et al., 2025; Liu et al., 2024; Liu & Xu, 2024b; Lu et al., 2024; Ghosh et al., 2026), the agent learns directly from interaction with an unknown environment while optimizing the worst-case criterion over the uncertainty set, which is centered at the interaction environment. It thus offers a principled way to retain the benefits of online learning while improving robustness to model mismatch.

Despite its promise, online DR-RL poses two fundamental theoretical challenges. First, the objective is inherently off-

---

[1]Department of Electrical and Computer Engineering, University of Central Florida, Orlando, FL 32816 [2]Department of Computer Science, University of Central Florida, Orlando, FL 32816. Correspondence to: Debamita Ghosh <debamita.ghosh@ucf.edu>, George K. Atia <george.atia@ucf.edu>, Yue Wang <yue.wang@ucf.edu>.

*Proceedings of the 43rd International Conference on Machine Learning*, Seoul, South Korea. PMLR 306, 2026. Copyright 2026 by the author(s).

target: data are collected under the nominal dynamics, while robustness is defined with respect to worst-case dynamics, which typically differ from the training environment. As a result, the agent must solve an off-dynamics learning problem (Eysenbach et al., 2020; Liu & Xu, 2024a; Holla, 2021). This mismatch can create an information bottleneck—samples that are crucial under the worst-case model may be rarely (or never) observed under the nominal dynamics the agent interacts with (Lu et al., 2024; Ghosh et al., 2026). Second, because learning occurs through real interaction, naive exploration is often unacceptable: the agent must maintain safe and satisfactory performance—even under worst-case dynamics—throughout learning, not just at convergence. Due to these difficulties, existing DR-RL methods generally assume extra data access, such as a generative model that can freely produce samples (Panaganti & Kalathil, 2022; Xu et al., 2023; Shi et al., 2023), a large offline dataset that sufficiently covers relevant dynamics (Blanchet et al., 2023; Shi & Chi, 2024; Tang et al., 2024; Wang et al., 2024c; Liu & Xu, 2024a; Panaganti et al., 2022; Wang et al., 2024a), or more recently hybrid regimes that combine substantial offline data with limited online interaction (Panaganti et al., 2024). In many practical settings, however, such simulators or datasets are unavailable or prohibitively costly, making purely online DR-RL essential.

Another obstacle of DR-RL is scalability. Most existing DR-RL algorithms are developed for small, tabular settings, whereas real applications typically involve enormous state–action spaces where tabular methods are infeasible. In standard RL, this gap is bridged by function approximation (Mnih et al., 2013; Silver et al., 2016; Kober et al., 2013; Li et al., 2016), which represents value functions within a low-dimensional hypothesis class. Extending this idea to DR-RL, however, is theoretically delicate. Because robustness targets worst-case dynamics that differ from the nominal data-generating process, an accurate low-dimensional approximation of the robust value function need not exist; for example, the worst-case value may fail to admit a good linear approximation even when the nominal value does (Tamar et al., 2014). Existing approaches that attempt to scale DR-RL with function approximation therefore often impose strong, hard-to-verify conditions—such as requiring a small discount factor (Xu & Mannor, 2010; Zhou et al., 2024; Badrinath & Kalathil, 2021) or assuming a linear-MDP structure (Ma et al., 2022; Liu & Xu, 2024b;a; Liu et al., 2024; Wang et al., 2024a).

These gaps naturally lead to one fundamental question: ***Can we develop a sample-efficient online DR-RL algorithm that scales up to large problems?***

In this paper, we answer this question by developing an online DR-RL framework with general function approximation and deriving its convergence guarantees. Our main contributions are summarized as follows.

**Sample-efficient algorithm for purely online DR-RL with general function approximation.** We propose RFL-$\phi$, the first algorithm for *purely online* distributionally robust RL with general function approximation under $\phi$-divergence uncertainty sets. The algorithm integrates optimism into a fitted-learning framework via a novel functional reformulation of the robust Bellman operator. Instead of using per–state–action bonuses (as in tabular UCB methods), RFL-$\phi$ constructs a *global uncertainty quantifier* over the function class, which more effectively aggregates estimation error and guides exploration. This yields a computationally efficient method for large-scale problems and, to our knowledge, the first polynomial-time, polynomial-sample algorithm for purely online DR-RL beyond tabular and offline/hybrid settings.

**Robust BE dimension as the intrinsic complexity measure.** We introduce the *robust BE dimension* as the intrinsic complexity measure governing learnability in online DR-RL with function approximation. It is defined as the distributional Eluder dimension of the robust Bellman residual class $(\mathcal{I} - \mathcal{T}_h^{\phi,\sigma})\mathcal{F}$ under on-policy distributions, capturing how many statistically independent policy-induced distributions can arise before generalization becomes inevitable. Unlike prior robust RL analyses that rely on coverage or concentrability assumptions (e.g., (Panaganti et al., 2022; 2024; He et al., 2025)), our framework depends solely on this *functional complexity measure*. We show that the robust BE dimension is finite for broad problem classes (including structured RMDPs such as tabular) and fully characterizes the sample complexity of online DR-RL, paralleling the role of BE dimension in non-robust RL (Jin et al., 2021; 2022).

**Dual robust fitted learning and global confidence sets.** We design a dual-based robust Bellman residual and use it to construct *global confidence sets* over value functions. Unlike tabular UCB or non-robust methods, which rely on per-state-action bonuses, RFL-$\phi$ optimizes a single least-squares objective on the dual residual that simultaneously (i) approximates the worst-case Bellman operator and (ii) acts as a global uncertainty quantifier for exploration. This dual-driven fitted learning mechanism is specific to online DR-RL and fundamentally differs from offline DR-RL, where the dual is analyzed under a fixed dataset and does not guide exploration.

**Sharp BE-dimension regret guarantees.** We establish the first regret guarantee for purely online DR-RL with general function approximation. Our bound takes the unified form
$$\mathcal{O}\left( H\, B_\phi(\sigma)\, \sqrt{\dim_{\text{BE}}^{\text{rob}}\; K/H\; \log(|\mathcal{F}||\mathcal{G}|KH/\delta)} + \varepsilon^{\text{dual}} \right),$$
where $B_\phi(\sigma)$, which is near-optimal when reduced to tabular or linear settings. The dependence on the problem instance is entirely through the intrinsic robust complexity

$\dim_{\text{BE}}^{\text{rob}}$, with *no coverage, density-ratio, or concentrability assumptions*, and the bound is independent of $|\mathcal{S}|$ and $|\mathcal{A}|$, demonstrating scalability to large or continuous domains.

## 2. Preliminaries and Problem Formulation

### 2.1. Distributionally Robust Markov Decision Process

DR-RL can be formulated as an episodic finite-horizon distributionally robust Markov decision process (RMDP) (Iyengar, 2005), represented by $\mathcal{M} := (\mathcal{S}, \mathcal{A}, H, \mathcal{P}, r)$, where the set $\mathcal{S} = \{1, \dots, S\}$ is the finite state space, $\mathcal{A} = \{1, \dots, A\}$ is the finite action space, $H$ is the horizon length, $r = \{r_h : \mathcal{S} \times \mathcal{A} \to [0,1]\}_{h=1}^H$ is the collection of reward functions, and $\mathcal{P} = \{\mathcal{P}_h\}_{h=1}^H$ is an uncertainty set of transition kernels. At step $h$, the agent is at state $s_h$ and takes an action $a_h$, receives the reward $r_h(s_h, a_h)$, and is transited to the next state $s_{h+1}$ following an arbitrary transition kernel $P_h(\cdot|s_h, a_h) \in \mathcal{P}_h$.

We consider the standard $(s, a)$-rectangular uncertainty set with divergence ball-structure (Wiesemann et al., 2013). Specifically, let $P^\star = \{P_h^\star\}_{h=1}^H$ be the *nominal* transition kernel, where each $P_h^\star : \mathcal{S} \times \mathcal{A} \to \Delta(\mathcal{S})$[1]. The uncertainty set, centered around $P^\star$, is defined as $\mathcal{P} = \mathcal{U}^{\phi,\sigma}(P^\star) = \bigotimes_{(h,s,a) \in [H] \times \mathcal{S} \times \mathcal{A}} \mathcal{U}_h^{\phi,\sigma}(s, a)$, and $\mathcal{U}_h^{\phi,\sigma}(s, a) \triangleq \{P \in \Delta(\mathcal{S}) : D_\phi(P, P_h^\star(\cdot|s, a)) \leq \sigma\}$, containing all the transition kernels that differ from $P^\star$ up to some uncertainty level $\sigma \geq 0$, under some probability divergence functions $D$ (Iyengar, 2005; Panaganti & Kalathil, 2022; Yang et al., 2022). Specifically, in this paper, we mainly focus on the standard $\phi$-divergence uncertainty set (Ghosh et al., 2026; Sason & Verdú, 2016).

**Definition 1** ($\phi$-Divergence Uncertainty Set). For each $(s, a)$ pair, the uncertainty set is defined as:

$$\mathcal{U}_h^{\phi,\sigma}(s, a) \triangleq \left\{ P \in \Delta(\mathcal{S}) : D_\phi\left(P, P_h^\star(\cdot|s, a)\right) \leq \sigma \right\},$$

where $D_\phi\left(P, P_h^\star(\cdot|s, a)\right) = \sum_{s' \in \mathcal{S}} \phi\left(\frac{P(s')}{P_h^\star(s'|s,a)}\right) P_h^\star(s'|s, a)$ is the $\phi$-divergence (Sason & Verdú, 2016).

Without loss of generality, we assume that $\text{Supp}(P) \subseteq \text{Supp}(P^\star)$ for all $P \in \mathcal{U}^{\phi,\sigma}$. Among the three uncertainty sets we consider later, the KL and $\chi^2$ divergence naturally satisfy this assumption, and the total variation also satisfies it under an additional standard assumption (Assumption 4). Notably, without this assumption, the sample complexity in the online setting can be exponentially large due to the information-bottleneck issue (Lu et al., 2024).

---

[1] $\Delta(\cdot)$ denotes the probability simplex over the space.

### 2.2. Policy and Robust Value Function

The agent's strategy of taking actions is captured by a Markov policy $\pi := \{\pi_h\}_{h=1}^H$, with $\pi_h : \mathcal{S} \to \Delta(\mathcal{A})$ for each step $h \in [H]$, where $\pi_h(\cdot|s)$ is the probability of taking actions at the state $s$ in step $h$. In RMDPs, the performance of a policy is captured by the worst-case performance, defined as the robust value functions. Specifically, given any policy $\pi$ and for each step $h \in [H]$, the *robust value function* and the *robust state-action value function* are defined as the expected accumulative reward under the worst possible transition kernel within the uncertainty set:

$$V_h^{\pi,\sigma}(s) = \inf_{P \in \mathcal{U}^{\phi,\sigma}} \mathbb{E}_{\pi,P}\left[\sum_{t=h}^H r_t(s_t, a_t)\Big| s_h = s\right], \quad (1)$$

$$Q_h^{\pi,\sigma}(s, a) = \inf_{P \in \mathcal{U}^{\phi,\sigma}} \mathbb{E}_{\pi,P}\left[\sum_{t=h}^H r_t(s_t, a_t)\Big| s_h = s, a_h = a\right],$$

where the expectation is taken with respect to the state-action trajectories induced by policy $\pi$ under transition $P$.

The goal of DR-RL is to find an optimal robust policy $\pi^\star := \{\pi_h^\star\}$ that maximizes the robust value function, for some initial state $s_1$:

$$\pi^\star \triangleq \arg\max_{\pi \in \Pi} V_1^{\pi,\sigma}(s_1), \quad (2)$$

where $\Pi$ is the set of policies. Such an optimal policy exists and can be a deterministic policy (Iyengar, 2005; Blanchet et al., 2023). Moreover, the optimal robust value functions (denoted by $Q_h^{\star,\sigma}, V_h^{\star,\sigma}$), which are the corresponding robust value functions of the optimal policy $\pi^\star$, are shown to be the unique solution to the robust Bellman equations:

$$Q_h^{\star,\sigma}(s, a) = r_h(s, a) + \mathbb{E}_{\mathcal{U}_h^{\phi,\sigma}(s,a)}\left[V_{h+1}^{\star,\sigma}\right],$$
$$V_h^{\star,\sigma}(s) = \max_{a \in \mathcal{A}} Q_h^{\star,\sigma}(s, a), \quad (3)$$

where $\mathbb{E}_{\mathcal{U}_h^{\phi,\sigma}(s,a)}[V] \triangleq \inf_{P_h \in \mathcal{U}_h^{\phi,\sigma}(s,a)} \mathbb{E}_{s' \sim P_h(\cdot|s,a)}[V(s')]$.

On the other hand, for any policy $\pi$, the corresponding robust value functions also satisfy the following robust Bellman equation for $\pi$ ((Blanchet et al., 2023)(Prop. 2.3)):

$$Q_h^{\pi,\sigma}(s, a) = r_h(s, a) + \mathbb{E}_{\mathcal{U}_h^{\phi,\sigma}(s,a)}\left[V_{h+1}^{\pi,\sigma}\right],$$
$$V_h^{\pi,\sigma}(s) = \mathbb{E}_{a \sim \pi_h(\cdot|s)}\left[Q_h^{\pi,\sigma}(s, a)\right]. \quad (4)$$

### 2.3. Online Distributionally Robust RL

In this work, we study distributionally robust RL in the online setting, where the agent aims to learn the robust-optimal policy $\pi^\star$ defined in eq. 2 through interaction with the nominal environment $P^\star$ over $K \in \mathbb{N}$ episodes. At the beginning of episode $k$, the agent observes the initial state

$s_1^k$, chooses a policy $\pi^k$ and executes it in $P^\star$ to generate a trajectory, and then updates its policy before the next episode. In the online setting, the agent cannot explore arbitrarily and must instead control the risk of worst-case outcomes during learning. Accordingly, the objective is to minimize the *cumulative robust regret* over $K$ episodes:

$$\text{Regret}(K) \triangleq \sum_{k=1}^{K} \left[ V_1^{\star,\sigma}(s_1^k) - V_1^{\pi^k,\sigma}(s_1^k) \right]. \quad (5)$$

We additionally assess performance via *sample complexity*, defined as the minimum number of samples $T = KH$ required to learn an $\varepsilon$-optimal robust policy $\hat{\pi}$ satisfying

$$V_1^{\star,\sigma}(s_1) - V_1^{\hat{\pi},\sigma}(s_1) \leq \varepsilon. \quad (6)$$

## 3. Robust Bellman Operator with Function Approximation

In this section, we describe the structural and computational challenges of *online* DR-RL with general function approximation under $\phi$-divergence uncertainty sets, and present our approach to overcoming them. Our presentation follows the Bellman–Eluder (BE) dimension framework for general RL with function approximation (Russo & Van Roy, 2013; Jiang et al., 2017; Sun et al., 2019; Jin et al., 2021), adapted to robust Bellman updates.

**Function approximation.** When the state–action space is large, learning robust policies from interaction alone is computationally challenging. To address this, we adopt the function approximation technique, where we use a general function class $\mathcal{F} = \{\mathcal{F}_h\}_{h=1}^{H}$, where $\mathcal{F}_h$ contains some functions $f : \mathcal{S} \times \mathcal{A} \to [0, H]$, to approximate the robust value function $Q_h^{\star,\sigma}$. This function class can be a parametric class with low-dimension parameters, e.g., neural network, to significantly reduce the computation and improve sample efficiency. To ensure effective learning with these function classes, prior work has identified structural conditions that they must satisfy (Russo & Van Roy, 2013; Jiang et al., 2017; Sun et al., 2019; Wang et al., 2020b; Jin et al., 2021; Panaganti et al., 2022). These conditions regulate how the functional class $\mathcal{F}$ interacts with the RMDP dynamics. The most commonly used assumptions are the ***representation conditions***, which require that $\mathcal{F}$ is expressive enough to capture the robust value functions of interest. More specifically, the optimal robust Q-function $Q^{\star,\sigma} \in \mathcal{F}$ (known as realizability) and closure under the robust Bellman operator, namely $\mathcal{T}_h^{\phi,\sigma}\mathcal{F}_{h+1} \subseteq \mathcal{F}_h$ (known as completeness). Following standard studies of function approximation in RL (Jin et al., 2021; Xie et al., 2022; Panaganti et al., 2022; Wang et al., 2019), we adopt the following completeness assumption.

**Assumption 1** (Completeness). *For all $h \in [H]$, we have $\mathcal{T}_h^{\phi,\sigma} f_{h+1} \in \mathcal{F}_h$ for all $f_{h+1} \in \mathcal{F}_{h+1}$.*

Per Assumption 1, $\mathcal{F}$ is closed under the robust Bellman operator $\mathcal{T}^{\phi,\sigma}$. Unlike standard function-approximation RL, we do not assume realizability ($Q^{\star,\sigma} \in \mathcal{F}$). Instead, realizability can be implied together with our assumption on the duality function class (Ma et al., 2022; Liu & Xu, 2024b;a; Liu et al., 2024; Wang et al., 2024a).

**Distribution families and robust BE dimension.** Let $\mathcal{X}$ be a domain and $\Phi \subseteq (\mathcal{X} \to \mathbb{R})$ be a function class. Let $\Pi$ be a family of probability distributions over $\mathcal{X}$.

**Definition 2** (Distributional $\varepsilon$-dependence (Jin et al., 2021)). A distribution $\mu \in \Pi$ is said to be $\varepsilon$-*dependent* on a set $\{\mu_1, \ldots, \mu_n\} \subseteq \Pi$ with respect to $\Xi$ if for all $\xi \in \Xi$,

$$\left| \mathbb{E}_\mu[\xi] \right| \leq \varepsilon + \left( \sum_{i=1}^{n} \mathbb{E}_{\mu_i}[\xi]^2 \right)^{1/2}. \quad (7)$$

Otherwise, $\mu$ is said to be $\varepsilon$-*independent* of $\{\mu_1, \ldots, \mu_n\}$.

**Definition 3** (Distributional Eluder (DE) dimension (Jin et al., 2021)). The *distributional Eluder dimension* of $\Xi$ with respect to $\Pi$ at scale $\varepsilon > 0$, denoted $\dim_{\text{DE}}(\Xi, \Pi, \varepsilon)$, is the length of the longest sequence $x_1, \ldots, x_m \in \Pi$ such that for each $i \in [m]$ there exists $\varepsilon' > \varepsilon$ such that $x_i$ is $\varepsilon'$-independent of $\{x_1, \ldots, x_{i-1}\}$.

**Remark 1.** *When $\Pi$ is taken to be the set of point masses $\{\delta_x(\cdot) := Dirac\ Distribution\ at\ x : x \in \mathcal{X}\}$, this definition reduces to the standard (pointwise) Eluder dimension.*

In the robust setting, we define the robust Bellman residual class at stage $h$ as

$$(\mathcal{I} - \mathcal{T}_h^{\phi,\sigma})\mathcal{F} := \left\{ f_h - \mathcal{T}_h^{\phi,\sigma} f_{h+1} : f \in \mathcal{F} \right\}, \quad (8)$$

where $\mathcal{T}_h^{\phi,\sigma} f(s,a) = r_h(s,a) + \mathbb{E}_{\mathcal{U}_h^{\phi,\sigma}(s,a)}[f_{\max}]$ is the robust Bellman operator, and $f_{\max}(s) = \max_a f(s,a)$. This class captures all possible robust Bellman errors that can arise from candidate value functions in $\mathcal{F}$. We then define the robust Bellman-Eluder (BE) dimension as follows.

**Definition 4.** Let $\dim_{\text{DE}}(\Xi, \Pi, \varepsilon)$ denote the DE dimension of a function class $\Xi$ with respect to a distribution family $\Pi$ at scale $\varepsilon$ (Jin et al., 2021). We then define the *robust Bellman–Eluder dimension* of $\mathcal{F}$ by

$$\dim_{\text{BE}}^{\text{rob}}(\mathcal{F}, \Pi, \varepsilon) := \max_{h \in [H]} \dim_{\text{DE}}\left( (\mathcal{I} - \mathcal{T}_h^{\phi,\sigma})\mathcal{F}, \Pi_h, \varepsilon \right),$$

where $\mathcal{I} - \mathcal{T}_h^{\phi,\sigma}$ is the robust Bellman residual class.

This quantity measures the intrinsic statistical complexity of learning robust value functions under function approxi-

mation. Smaller robust BE dimension implies fewer distinguishable robust Bellman errors along interaction trajectories, enabling tighter confidence sets and improved sample efficiency.

**Remark 2** (Distribution families $D_{\mathcal{F}}$ and $D_\Delta$)**.** *Following (Jin et al., 2021), we consider two distribution families:*

- $D_{\mathcal{F}} = \{D_{\mathcal{F},h}\}_{h=1}^H$, *where $D_{\mathcal{F},h}$ is the collection of step-$h$ state-action occupancy distributions induced by rolling in with $\pi_f$ for some $f \in \mathcal{F}$.*

- $D_\Delta = \{D_{\Delta,h}\}_{h=1}^H$, *where $D_{\Delta,h}$ is the set of point masses on $\mathcal{S} \times \mathcal{A}$.*

**Remark 3** (Robust Bellman Rank and Relations with known RL Problems)**.** *Many structured RL models—e.g., tabular, linear, reactive POMDPs—admit efficient learning guarantees. In the non-robust literature, two generic tractability notions—low Bellman rank and low Eluder dimension–cover many examples but are incomparable. For DR-RL, we instead use a single notion—robust BE dimension—defined via the robust Bellman residual class (eq. 8). As shown in Appendix D, low robust BE dimension subsumes the robust analogs of both low robust Bellman rank and low Eluder dimension, providing a unified structural condition for tractable robust RL with function approximation.*

**Empirical robust Bellman operator and functional optimization.** We first provide the dual formulation of $\phi$-divergence uncertainty set support functions. Given a nominal kernel $P^\star$, there is a dual formulation of $\mathbb{E}_{\mathcal{U}_h^{\phi,\sigma}(s,a)}[\cdot]$:

$$
\mathbb{E}_{\mathcal{U}_h^\sigma(s,a)}[V] = -\inf_{\eta>0,\ \nu\in\mathbb{R}} \Big\{ \eta\sigma - \nu + \\
\eta \mathbb{E}_{s'\sim P_h^\star(s'|s,a)} \left[ \phi^\star((\nu - V(s'))/\eta) \right] \Big\},
\tag{9}
$$

where $\phi^\star$ is the convex conjugate of $\phi$ restricted to $[0,\infty)$. We will use this representation as a unifying template for deriving robust bonuses. This formula, as given in eq. 9, follows from standard strong duality arguments; see, e.g., (Yang et al., 2022)[Lemma B.1].

We now denote $l_\phi(f; s, a, s'; \eta, \nu)$ as the pointwise dual integrand for a single next-state $s'$ and it is defined as

$$
l_\phi(f; s, a, s'; \eta, \nu) \triangleq \eta\sigma - \nu \tag{10} \\
+ \eta\phi^\star\left( -\frac{\max_{a'} f(s', a') + \nu}{\eta} \right).
$$

The robust Bellman operator is then

$$
(\mathcal{T}_h^{\phi,\sigma} f_{h+1})(s, a) := r(s, a) \\
- \inf_{\eta>0,\ \nu\in\mathbb{R}} \mathbb{E}_{s'\sim P_h^\star(s,a)} \left[ l_\phi(f_{h+1}; s, a, s'; \eta, \nu) \right]. \tag{11}
$$

From eq. 11 we can recall that the robust value function is the fixed point of the robust Bellman operator. Therefore, finding an optimal robust policy reduces to computing this fixed point. As a result, applying the operator exactly is generally impractical: for every $(s, a)$, the term $\mathbb{E}_{\mathcal{U}_h^{\phi,\sigma}(s,a)}[\cdot]$ entails solving an optimization problem over an $S$-dimensional $\phi$-divergence uncertainty set, which rapidly becomes computationally expensive.

To overcome this computational bottleneck, we develop an efficient empirical procedure, inspired by (Panaganti et al., 2022), which eliminates pointwise scalar optimizations by reformulating the computation as a *single* functional optimization problem. Specifically, consider the probability space $(\mathcal{S} \times \mathcal{A}, \Sigma(\mathcal{S} \times \mathcal{A}), \mu)$, where $\mu$ is a distribution over $\mathcal{S} \times \mathcal{A}$ (typically the stage-$h$ visitation distribution). Let $\mathcal{L}^1(\mu; \mathbb{R}^2) \triangleq \{g = (g_\eta, g_\nu) : \mathcal{S} \times \mathcal{A} \to \mathbb{R}^2 | g_\eta, g_\nu \in \mathcal{L}^1(\mu)\}$ denote the space of absolutely integrable dual functions and define the dual loss $\mathrm{DualLoss}(g; f)$ as

$$
\mathbb{E}_{(s,a)\sim\mu} \left[ \mathbb{E}_{s'\sim P_{s,a}^\star}[l_\phi(f; s, a, s'; g)] \right], \tag{12}
$$

where $g(s, a) = (g_\eta(s,a), g_\nu(s,a))$ and $l_\phi(f; s, a, s'; g) := g_\eta(s,a)\sigma - g_\nu(s,a) + g_\eta(s,a)\phi^\star\left( -\frac{\max_{a'} f(s',a') + g_\nu(s,a)}{g_\eta(s,a)} \right)$. We then extend the results in (Panaganti et al., 2022) (which considers the total variation set) and (Panaganti et al., 2024) (which considers regularized MDPs), and show that for general $\phi$-divergence RMDP, minimization of the dual loss is equivalent to the support function under some distribution.

**Lemma 1** (Dual loss minimization)**.** *Let $\mathrm{DualLoss}$ be defined as the dual loss function as in eq. 12. Then, for any function $f : \mathcal{S} \times \mathcal{A} \to [0, H]$, we have*

$$
\inf_{g \in \mathcal{L}^1(\mu;\mathbb{R}^2)} \mathrm{DualLoss}(g; f) = \mathbb{E}_{(s,a)\sim\mu} \left[ \mathbb{E}_{\mathcal{U}_h^{\phi,\sigma}(s,a)}[f] \right].
$$

This enables us to form an empirical counterpart of the dual objective, denoted by $\widehat{\mathrm{DualLoss}}(g; f)$, and compute an approximate dual minimizer via $\inf_{g \in \mathcal{L}^1(\mu;\mathbb{R}^2)} \widehat{\mathrm{DualLoss}}(g; f)$. We further introduce a function class $\mathcal{G} = \{g = (g_\eta, g_\nu) | g_\eta, g_\nu : \mathcal{S} \times \mathcal{A} \to \Theta_\phi\}$ to approximate the dual variables as well. We adopt the following realizability condition from (Panaganti et al., 2022; 2024).

**Assumption 2.** *For all $f \in \mathcal{F}$ and any policy $\pi$, there exists a uniform constant $\varepsilon^{\mathrm{dual}}$ such that*

$$
\inf_{g\in\mathcal{G}} \mathrm{DualLoss}(g; f) - \inf_{g\in\mathcal{L}^1(\mu^\pi;\mathbb{R}^2)} \mathrm{DualLoss}(g; f) \le \varepsilon^{\mathrm{dual}},
$$

*where $\mu^\pi$ is the distribution induced by $\pi$ under $P^\star$.*

This assumption is not restrictive. For instance, note that $\mathcal{L}^1$ can be approximated by deep/wide neural networks (Goodfellow et al., 2016), which ensures Assumption 2 with such

neural network classes. Accordingly, for a fixed $f$ and dataset $\mathcal{D}$, we approximate the robust Bellman operator by first computing $\underline{g}_f = \arg\min_{g \in \mathcal{G}} \widehat{\text{DualLoss}}(g; f)$, and then plugging it in the robust Bellman operator as $(\mathcal{T}_{\underline{g}_f}^{\phi,\sigma} f)(s,a)$:

$$r(s,a) - \mathbb{E}_{s' \sim P_{s,a}^{\star}}\left[l_\phi(f; s, a, s'; \underline{g}_f)\right]. \tag{13}$$

**Uniform approximation of the robust Bellman operator.** The following lemma quantifies how accurately the empirical operator approximates the true robust Bellman operator in a global $\mathcal{L}^1$ sense. We first adopt the following assumption, which holds for all uncertainty sets we considered.

**Assumption 3.** *There exist a compact set $\Theta_\phi \subseteq \mathbb{R}_+$ and a constant $B_\phi(\sigma) < \infty$ such that for all $h \in [H]$, all $f_{h+1} \in \mathcal{F}_{h+1}$, all $\eta, \nu \in \Theta_\phi$, and for all $(s, a, s')$, $\left|l_\phi(f_{h+1}; s, a, s'; \eta, \nu)\right| \leq B_\phi(\sigma)$.*

**Lemma 2** (Uniform approximation of robust Bellman operator). *Fix any policy $\pi$, let $\mu_h^\pi$ be the step-$h$ state-action distribution under $P^\star$, and let $\mathcal{D}$ be the dataset collected by executing $\pi$. Then, under Assumption 1-3, for any $\delta \in (0, 1)$, with probability at least $1 - \delta$, it holds that*

$$\sup_{f \in \mathcal{F}_{h+1}} \left\|\mathcal{T}_h^{\phi,\sigma} f - \mathcal{T}_{\underline{g}_f}^{\phi,\sigma} f\right\|_{1, \mu_h^\pi}$$

$$= \mathcal{O}\left(B_\phi(\sigma)\sqrt{\frac{\log\left(|\mathcal{F}_{h+1}||\mathcal{G}|/\delta\right)}{|\mathcal{D}|}} + \varepsilon^{\text{dual}}\right). \tag{14}$$

A similar result is derived for a fixed distribution (the offline dataset distribution) in (Panaganti et al., 2022; 2024), whereas we show it simultaneously holds for any policy and its induced distribution. Lemma 2 shows that our empirical functional optimization yields a uniformly accurate approximation to the robust Bellman operator under the $\mathcal{L}^1(\mu^\pi; \mathbb{R}^2)$ norm. Crucially, the error is controlled *globally* with respect to the distribution $\mu^\pi$, rather than pointwise in $(s, a)$. This global control is what we leverage later to define our robust confidence sets and the global error term that drives the design and analysis of our main algorithm.

**Remark 4** (Relation to $\varphi$-regularized RMDPs (Panaganti et al., 2024)). *Assumption 2 and Lemma 2 build on the dual functional machinery first developed by (Panaganti et al., 2022) and subsequently employed by (Panaganti et al., 2024) for $\varphi$-regularized RMDPs in a hybrid setting, where the policy value includes a Lagrangian penalty $\lambda$ with $\lambda > 0$ and the guarantees scale with $(\lambda + H)$. Although the $\varphi$-regularized RMDPs recovers the standard RMDPs with $\lambda = 0$, the studies in (Panaganti et al., 2024) are developed for $\lambda > 0$, hence our result cannot be obtained directly.*

## 4. Robust Fitting Learning Algorithm

We then utilize our previous constructions and propose our Robust Fitted Learning (RFL) algorithm.

---

**Algorithm 1** Robust Fitted Learning with $\phi$-Divergence Uncertainty Set (RFL-$\phi$)

---

1: **Input:** Function class $\mathcal{F}$, Dual Function class $\mathcal{G}$, $\beta > 0$, $\sigma > 0$.
2: **Initialize:** $\mathcal{F}^{(0)} \leftarrow \mathcal{F}$, $\mathcal{D}_h^{(0)} \leftarrow \emptyset \, \forall h \in [H]$
3: **for** episode $k = 1, 2, \ldots, K$ **do**
4:     $\pi^{(k)} \leftarrow \pi^{f^{(k)}}$, where $f^{(k)} \leftarrow \arg\max_{f \in \mathcal{F}^{(k-1)}} f(s_1, \pi_1^f(s_1))$
5:     Execute $\pi^{(k)}$ to collection trajectory $(s_1^{(k)}, a_1^{(k)}, r_1^{(k)}), \ldots, (s_H^{(k)}, a_H^{(k)}, r_H^{(k)})$
6:     $\mathcal{D}_h^{(k)} \leftarrow \mathcal{D}_h^{(k-1)} \cup \{(s_h^{(k)}, a_h^{(k)}, s_{h+1}^{(k)})\}, \forall h$
7:     $\mathcal{F}_H^{(k)} \leftarrow \{0\}$
8:     For all $f_{h+1} \in \mathcal{F}_{h+1}^{(k)}$, update the confidence set according to eq. 15:

$$\mathcal{F}_h^{(k)} \leftarrow \left\{ f \in \mathcal{F}_h : L_{\mathcal{D}_h^{(k)}}(f_h, f_{h+1}, \underline{g}_{f_{h+1}}) \right.$$
$$\left. - \min_{f_h' \in \mathcal{F}_h} L_{\mathcal{D}_h^{(k)}}(f_h', f_{h+1}, \underline{g}_{f_{h+1}}) \leq \beta, \forall h \in [H] \right\}$$

9: **end for**
10: **Output:** $\bar{\pi} = \text{unif}(\pi^{(1:K)})$.

---

Our algorithm follows the standard fitting learning structure. In each step $h$, we construct a confidence set $\mathcal{F}^{(k)}$ (Line 8) using the fitted error under the robust Bellman operator to ensure the inclusion of $Q^{\star,\sigma} \in \mathcal{F}^{(k)}$ (Lemma K.2). As discussed, we utilize our functional optimization based loss function and the error bound in Lemma 2 to construct the set. Namely, given a function $f$, we first obtain the dual-variable approximation $(\underline{g}_{\eta,f}, \underline{g}_{\nu,f})$ via minimizing empirical functional optimization loss, formulated as

$$\underline{g}_f \triangleq \arg\min_{g \in \mathcal{G}} \sum_{(s,a,s') \in \mathcal{D}_h^{(k)}} \left(l_\phi(f; s, a, s'; g)\right), \tag{15}$$

where we further capture the empirical robust Bellman error $L_{\mathcal{D}_h^{(k)}}(f', f, g)$ based on dataset $\mathcal{D}_h^{(k)}$ as $\sum_{(s,a,r,s') \in \mathcal{D}_h^{(k)}} \left\{f'(s, a) - r - l_\phi(f; s, a, s'; g)\right\}^2$.

Notably, to handle large-scale problems we build *global* confidence sets by optimizing over $\{f_h\}_{h=1}^H$ jointly (Zanette et al., 2020), rather than certifying errors state–action-wise as in tabular UCB. Concretely, $\mathcal{F}^{(k)}$ contains functions that (i) achieve small squared robust Bellman error on the collected data $\{\mathcal{D}_h^{(k)}\}_{h=1}^H$ via the dual plug-in residual, and (ii) include any $f$ whose empirical loss is within a tolerance of the best loss in $\mathcal{F}$. We later choose the threshold $\beta$ so that $Q^{\star,\sigma} \in \mathcal{F}^{(k)}$ holds with high probability. Given this valid set, we apply optimism and select $\pi^{(k)} = \pi^{f^{(k)}}$,

where $f^{(k)} \in \mathcal{F}^{(k)}$ maximizes the optimistic estimate $f_1^{(k)}\big(s_1, \pi_1^{(k)}(s_1)\big)$ of total return, thereby balancing exploration and performance.

**Algorithmic novelties.** While our algorithm follows the high-level paradigm of "optimism + fitted value iteration," similar in spirit to GOLF (Jin et al., 2021; Xie et al., 2022), its design is fundamentally tailored to the robust BE dimension rather than non-robust Bellman-error control or coverage-based arguments (Xie et al., 2022) and offline DR-RL (Panaganti et al., 2022; 2024). A central challenge is that the data $\mathcal{D}_h^{(k)}$ are collected under a sequence of evolving policies across episodes, so learning a single policy $\pi$ cannot rely on guarantees derived under a fixed sampling distribution $\mathcal{D}_h^{(k)} \sim \mu^\pi$. To address this, we develop an optimistic, dual-driven fitted learning procedure in which a value–dual pair $(f, g)$ is learned jointly online. The dual function $g$ serves a dual role: it provides a tractable functional approximation to the TV-robust Bellman operator, and it induces a global control of the robust Bellman residual class $(\mathcal{I} - \mathcal{T}_h^{\phi,\sigma})\mathcal{F}$, which is precisely the object governing the robust BE dimension (Definition 4). This allows us to construct confidence sets directly in function space and implement optimism in a manner that is tightly coupled to the intrinsic complexity of the problem. In contrast to GOLF, which controls squared non-robust Bellman errors, and to offline RFQI-style robust methods (Panaganti et al., 2022), where the dual is analyzed under a fixed data distribution and does not guide exploration, our approach leverages the dual to both approximate robustness and drive exploration. This alignment between algorithm design, statistical control, and complexity measure is what enables regret guarantees governed by the robust BE dimension.

## 5. Theoretical Guarantees

We then develop the theoretical guarantees of our algorithm.

**Theorem 1.** *Assume Assumption 1-3. For any $\delta \in (0, 1]$, we set $\beta = \mathcal{O}\Big(B_\phi(\sigma) \log\big(|\mathcal{F}||\mathcal{G}|KH/\delta\big)\Big)$ in Algorithm 1. Then, with probability at least $1 - \delta$,* [2]

$$\text{Regret}(K) \leq \tilde{\mathcal{O}}\Big(\sqrt{dH^2 B_\phi^2(\sigma) K} + \varepsilon^{\text{dual}}\Big), \quad (16)$$

*where $d := \dim_{\text{BE}}^{\text{rob}}\big(\mathcal{F}, D_{\mathcal{F}}, 1/\sqrt{K}\big)$ as in Definition 4.*

**Technical novelties and implications.** Our work provides the first regret guarantees for online DR-RL with general function approximation based on an intrinsic complexity measure—the robust Bellman–Eluder (BE) dimen-

sion—without relying on coverage, density-ratio, or concentrability assumptions. This brings online robust RL into the modern complexity-theoretic framework of general function approximation (Jin et al., 2021), while requiring new techniques to handle worst-case dynamics.

First, we introduce the robust BE dimension as the appropriate notion of intrinsic difficulty for online DR-RL. Our bounds depend only on the statistical complexity of the robust Bellman residual class $(\mathcal{I} - \mathcal{T}_h^{\phi,\sigma})\mathcal{F}$ evaluated along on-policy trajectories. This yields an exploration theory for robust MDPs analogous to BE-based non-robust RL, avoids external coverage assumptions, and recovers sharp rates in structured settings such as tabular and linear RMDPs. Second, unlike existing BE analyses that assume a known Bellman operator, our setting requires learning the robust Bellman operator from data. We address this via a dual ERM plug-in backup, which introduces a new operator approximation error. We develop a new uniform control argument based on bounded $\phi$-divergence multipliers and approximate dual realizability, a component absent from prior BE-based theory. Finally, our regret decomposition cleanly separates (i) an exploration term governed by the robust BE dimension and (ii) an operator-approximation term governed by dual learning error. This parallels the structure of non-robust BE analyses but requires fundamentally new arguments due to the learned robust operator.

In contrast to offline/hybrid robust RL, which assumes fixed datasets and strong global coverage assumptions (Panaganti et al., 2022; 2024), we study a fully online setting with evolving data distributions and provide the first intrinsic, complexity-theoretic characterization of exploration in online DR-RL with general function approximation.

As an immediate corollary, we obtain the sample complexity for learning an $\varepsilon$-optimal policy with RFL-$\phi$ by applying a standard online-to-batch conversion (Cesa-Bianchi et al., 2001) for each total variation (TV), $\chi^2$ and KL divergence uncertainty sets.

**Corollary 1** (Sample Complexity: TV, $\chi^2$, KL). *Under the same setup in Theorem 1, set $\varepsilon^{\text{dual}} = 0$ and additionally assume Assumption 4 for TV. Then, with probability at least $1-\delta$, the sample-complexity of RFL-$\phi$ to obtain an $\varepsilon$-optimal robust policy is $T = KH$, with $T$ given by*

$$
\begin{cases}
\mathcal{O}\left(\frac{H^5(\min\{H, 1/\sigma\})^2 d \log\left(|\mathcal{F}||\mathcal{G}|T/\delta\right)}{\varepsilon^2}\right), \text{TV-RMDP} \\
\mathcal{O}\left(\frac{H^5(1+\sqrt{\sigma})^2 d \log\left(|\mathcal{F}||\mathcal{G}|T/\delta\right)}{\varepsilon^2}\right), \quad \chi^2\text{-RMDP} \\
\mathcal{O}\left(\frac{H^5 \sigma^2 d \log\left(|\mathcal{F}||\mathcal{G}|T/\delta\right)}{\varepsilon^2}\right), \quad \text{KL-RMDP}.
\end{cases}
$$

Our bound is independent of $S$ and $A$, indicating scalability to large state and action spaces. Moreover, as we shall

---

[2]We assume for simplicity that $|\mathcal{F}|, |\mathcal{G}| < \infty$, but our result can be directly extended to the general infinite case with the standard finite coverage technique (Xie et al., 2022; Panaganti et al., 2022).

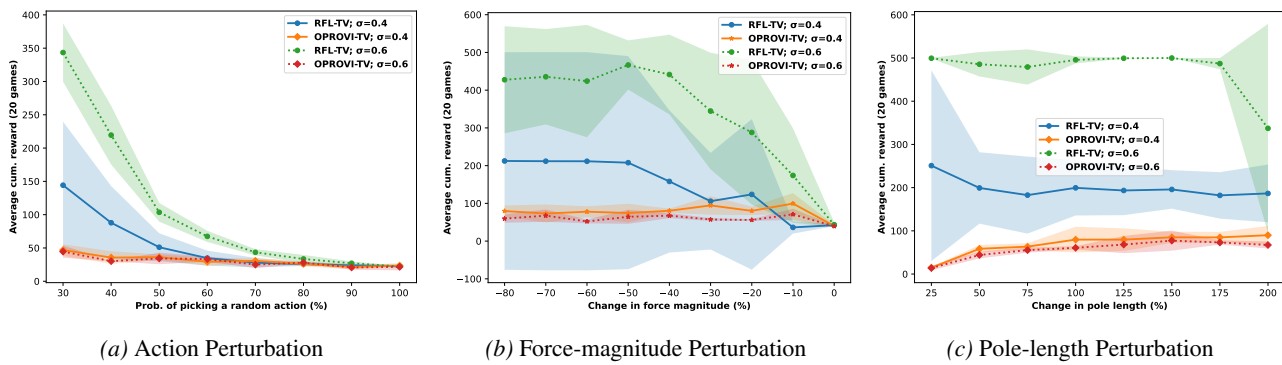

*(a)* Action Perturbation        *(b)* Force-magnitude Perturbation        *(c)* Pole-length Perturbation

*Figure 1.* RFL-TV vs. OPROVI-TV (Tabular).

discuss later, the dependences on other parameters, $H, \sigma, \varepsilon$, are also tight and near-optimal.

To validate the near optimality and sharpness of our results, we specialize them to two special cases: tabular case and linear RMDP (Ma et al., 2022; Liu et al., 2024) setting. A more detailed comparison is provided in Appendix B.

**Remark 5** (Tabular RMDPs). *In the finite tabular case, we take $\mathcal{F}$ and $\mathcal{G}$ as the full classes of bounded functions $\mathcal{S} \times \mathcal{A} \to [0, H]$ and $\mathcal{S} \times \mathcal{A} \to \Theta_\phi$, so that $\log(|\mathcal{F}||\mathcal{G}|) = \tilde{\mathcal{O}}(SA)$ (Jin et al., 2021) and $d = \mathcal{O}(SA)$. Plugging these into Theorem 1 yields a tight tabular regret bound. For instance, for $\chi^2$ set, our sample complexity for an $\varepsilon$-optimal policy is $\tilde{\mathcal{O}}\big(H^5(1 + \sqrt{\sigma})^2 S^2 A^2 \varepsilon^{-2}\big)$, which improves the prior results $\tilde{\mathcal{O}}\big(C_{\mathrm{vr}} H^5(1 + \sqrt{\sigma})^2 S^3 A \varepsilon^{-2}\big)$ in (He et al., 2025) (note that $C_{\mathrm{vr}}$ is polynomial in $S, H, A$).*

**Remark 6** (Linear RMDPs). *As another special case, we instantiate our framework to $d_{\mathrm{lin}}$-rectangular linear RMDPs (Ma et al., 2022; Liu et al., 2024). Note that for linear RMDPs, the uncertainty rectangularity is defined differently from our $(s, a)$-rectangularity. However, our method can be similarly extended (see Section F). When $\mathcal{F}$ and the dual class $\mathcal{G}$ are $d_{\mathrm{lin}}$-dimensional linear function classes, the robust BE complexity satisfies $\dim_{\mathrm{BE}}^{\mathrm{rob}}(\mathcal{F}, D_{\mathcal{F}}, 1/\sqrt{K}) = \tilde{\mathcal{O}}(d_{\mathrm{lin}})$ and $\log(|\mathcal{F}||\mathcal{G}|) = \tilde{\mathcal{O}}(d_{\mathrm{lin}})$ (cf. (Wang et al., 2020b; Jin et al., 2021)). Substituting into Theorem 1 with $\varepsilon^{\mathrm{dual}} = 0$ yields $\mathrm{Regret}(K) = \tilde{\mathcal{O}}\big(\sqrt{d_{\mathrm{lin}}^2 H^2 B_\phi^2(\sigma) K}\big)$. In particular, for TV uncertainty, we obtain $\mathrm{Regret}(K) = \tilde{\mathcal{O}}\big(\sqrt{d_{\mathrm{lin}}^2 H^4 \big(\min\{H, 1/\sigma\}\big)^2 K}\big)$, which matches the sharp dependencies on $(d_{\mathrm{lin}}, \sigma, K)$ in (Liu et al., 2024) except $H$. Hence, our general function-approximation theory recovers the near-optimal regret while strictly extending the scope beyond linear realizability.*

Our results hence are sharp and approximately match or improve the previous results under the two special cases, while strictly extending the generality.

## 6. Experiments

We then numerically verify the effectiveness of our algorithm. Additional experiments are presented in Section C.

We evaluate performances under the CartPole-v1 (Brockman et al., 2016) with discrete actions $\mathcal{A} = \{0, 1\}$ and horizon $H = 500$. We first learn policies by training under the *nominal environment*, and evaluate its performance under three types of environmental shifts: (i) **action perturbation**, where with probability $\rho$ the executed action is replaced by a uniformly random action; (ii) **force-magnitude scaling**, multiplying the applied force by $\eta_{\mathrm{force}}$; and (iii) **pole-length scaling**, multiplying the pole length by $\eta_{\mathrm{len}}$. Each reported point averages undiscounted return over 20 evaluation episodes, with 95% confidence intervals.

In our experiments, we implement RFL-TVwith two neural networks as approximation function classes (details in Appendix C.1; implementation in Algorithm 2) and compare against OPROVI-TV, an online tabular TV-RMDP baseline that discretizes the state space and applies the tabular method of (Lu et al., 2024). The tabular approach incurs complexity scaling with the number of bins/states $S$, whereas ours uses neural function approximation and is substantially more efficient. We test the average reward of learned policies under perturbations for different perturbation radii in Figure 1. As the results show, RFL-TV achieves consistently higher returns across all shift types, and is more robust (degrading more slowly) to large perturbations, indicating that function approximation can outperform tabular learning, especially at large scale.

We additionally provide a detailed analysis of the practical behavior of RFL-TV in Appendix C.2, including: (i) sensitivity to the robustness hyperparameters $(\sigma, \beta)$, where $\sigma$ controls the size of the uncertainty set and $\beta$ controls the strictness of the dual Bellman constraint enforcement, (ii) robustness under different perturbation regimes, and (iii) computational complexity and runtime overhead. The results show that the proposed method exhibits stable and

interpretable sensitivity trends, while introducing only a moderate runtime overhead ($\approx 1.5\times$ relative to DQN) despite the additional dual optimization step.

## 7. Conclusion

In this work, we introduced RFL-$\phi$, a DR-RL algorithm with general function approximation for online settings. The algorithm implements a fitted robust Bellman update via a functional optimization and replaces state-action bonuses with a global uncertainty quantifier that more effectively guides exploration. Our theoretical analysis is grounded in the robust BE dimension, which we proposed to capture the intrinsic statistical complexity of learning robust value functions under function approximation. Our result yields strong sample-efficiency guarantees for large-scale problems, achieving sub-linear regret, and is independent of state/action spaces. When reduced to both tabular and linear RMDP cases, our results are both near-optimal against existing works and minimax lower bounds, which implies the tightness and efficiency of our algorithms. Our algorithm thus constitutes the first purely online and sample-efficient algorithm for large-scale DR-RL, providing a theoretical foundation and a scalable algorithm for robust learning in high-dimensional environments.

## Acknowledgments

This work was supported by DARPA under Agreement No. HR0011-24-9-0427, NSF under Award CCF-2106339, and an Amazon Research Award, Fall 2025. Any opinions, findings, and conclusions or recommendations expressed in this material are those of the author(s) and do not reflect the views of DARPA, NSF or Amazon.

## Impact Statement

This paper presents work whose goal is to advance the field of machine learning. There are many potential societal consequences of our work, none of which we feel must be specifically highlighted here.

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

# A. Related Works

We discuss most related DR-RL works here.

**Tabular DR-RL:** DR-RL has been studied predominantly in the tabular regime. A substantial line of work establishes minimax-optimal or near-optimal guarantees under the generative-model setting (Clavier et al., 2023; Liu et al., 2022; Panaganti & Kalathil, 2022; Ramesh et al., 2024; Shi et al., 2023; Wang et al., 2023a;b; 2024b; Xu et al., 2023; Yang et al., 2022; 2023; Badrinath & Kalathil, 2021; Li et al., 2022b; Liang et al., 2023), where the agent can query a simulator or relies on a large offline dataset (Blanchet et al., 2023; Shi & Chi, 2024; Zhang et al., 2023; Liu & Xu, 2024a; Wang et al., 2024c;a). More recently, a limited number of works have addressed the online DR-RL setting (Dong et al., 2022; Wang & Zou, 2021; Lu et al., 2024; He et al., 2025; Ghosh et al., 2026), where the information bottleneck is handled via additional technical assumptions to obtain sample-efficient guarantees. However, these studies are primarily model-based or value-based, and consequently do not scale well to large state–action spaces.

**DR-RL with Function Approximation:** Existing theoretical DR-RL with function approximation has largely focused on linear classes. Yet linear classes are typically not closed under the robust Bellman operator, so one cannot generally guarantee approximation quality. To sidestep this mismatch, prior work often assumes strong, hard-to-verify structure in the underlying RMDP, such as a small discount factor (Xu & Mannor, 2010; Tamar et al., 2014; Zhou et al., 2024) or an explicitly linear RMDP model (Ma et al., 2022; Liu & Xu, 2024b;a; Liu et al., 2024; Wang et al., 2024a). In contrast, we allow a general function class, avoiding these restrictive assumptions. General function approximation for DR-RL has so far been developed mainly in (Panaganti et al., 2022; 2024), which adopt functional-optimization-based robust backups but operate in offline or hybrid regimes under global coverage conditions, thereby bypassing the exploration difficulties intrinsic to our fully online setting; moreover, (Panaganti et al., 2024) analyzes regularized RMDPs, which differs from the DR-RL objective studied here.

## A.1. Other Related Works: Non-Robust RL with Functional Approximation

Function approximation has been extensively studied in non-robust reinforcement learning. While a large body of work focuses on *offline RL with general function approximation* (e.g., (Zhan et al., 2022; Jiang & Xie, 2024; Wang et al., 2020a)), our work operates in the *online* setting, where the agent must actively explore while learning from interaction with the environment.

A central theme in online RL is identifying *structural complexity measures* that characterize when learning with function approximation is tractable. The *Eluder dimension* (Li et al., 2022a; Russo & Van Roy, 2013) was introduced to quantify the sequential complexity of a function class and has been used to design optimistic algorithms whose confidence sets and exploration bonuses scale with this measure (Wang et al., 2020b). However, the Eluder dimension captures only the complexity of the function class in isolation, without accounting for its interaction with the environment dynamics.

To address this limitation, more refined notions were proposed to capture the interplay between the hypothesis class and the MDP. The *Bellman rank* (Jiang et al., 2017) and *Witness rank* (Sun et al., 2019) formalize this interaction, and were later unified under the *Bellman–Eluder (BE) dimension* framework (Jin et al., 2021). The BE dimension directly measures the complexity relevant to value-based learning—namely, the difficulty of approximating Bellman residuals—and provides a unified lens for understanding tractability across a wide range of structured RL models.

More recently, a complementary line of work has emphasized *coverage-type conditions* as characterizations of learnability. In particular, (Xie et al., 2022) introduced the notion of coverability and showed that it is both necessary and sufficient for efficient exploration with function approximation, subsuming earlier assumptions such as concentrability and bounded Bellman rank. At the same time, hardness results (Foster et al., 2021; Du et al., 2021) demonstrate that without such structural or complexity assumptions, online RL with rich observations can be exponentially hard.

Our work situates itself firmly in this online learning regime but departs from these approaches in a fundamental way: we use the (robust) Bellman–Eluder dimension as the primary complexity measure, rather than coverage coefficients. This choice is deliberate. BE dimension provides a model-based notion of complexity tied directly to Bellman residuals and value approximation, which is better aligned with robust Bellman operators and admits a natural extension to distributionally robust settings.

Crucially, the guarantees developed for non-robust online RL do *not* transfer directly to the robust case. Distributionally robust RL replaces a single transition kernel with an *uncertainty set* and optimizes against a *worst-case Bellman operator*,

*Table 1.* Comparison of sample complexity in general-function, tabular and linear settings. Our contributions are highlighted.

| Setting | Method | Robust | Divergence | Sample complexity |
|---|---|---|---|---|
| **General** | Online, (Xie et al., 2022) | No | – | $\tilde{\mathcal{O}}(C_{\text{cov}}H^3 \log(|\mathcal{F}|/\delta)\varepsilon^{-2})$ |
| | Online, (Jin et al., 2021) | No | – | $\tilde{\mathcal{O}}(H^3 \dim_{\text{BE}} \log(|\mathcal{F}|/\delta)\varepsilon^{-2})$ |
| | **Online, RFL-$\phi$ (Ours)** | Yes | TV | $\tilde{\mathcal{O}}(H^5(\min\{H,1/\sigma\})^2 \dim_{\text{BE}}^{\text{rob}} \log(|\mathcal{F}||\mathcal{G}|/\delta)\varepsilon^{-2})$ |
| | | | $\chi^2$ | $\tilde{\mathcal{O}}(H^5(1+\sqrt{\sigma})^2 \dim_{\text{BE}}^{\text{rob}} \log(|\mathcal{F}||\mathcal{G}|/\delta)\varepsilon^{-2})$ |
| | | | KL | $\tilde{\mathcal{O}}(H^5\sigma^2 \dim_{\text{BE}}^{\text{rob}} \log(|\mathcal{F}||\mathcal{G}|/\delta)\varepsilon^{-2})$ |
| | Lower Bound | – | – | N/A |
| **Tabular** | Online, (Azar et al., 2017) | No | – | $\tilde{\mathcal{O}}(SAH^4\varepsilon^{-2})$ |
| | Online, (Lu et al., 2024) | Yes | TV | $\tilde{\mathcal{O}}(H^3 \min\{H,\sigma^{-1}\}SA\varepsilon^{-2})$ |
| | Online, (He et al., 2025) | Yes | TV | $\tilde{\mathcal{O}}(C_{\text{vr}}S^3AH^5\varepsilon^{-2})$ |
| | | | $\chi^2$ | $\tilde{\mathcal{O}}(C_{\text{vr}}(1+\sqrt{\sigma})^2S^3AH^5\varepsilon^{-2})$ |
| | | | KL | $\tilde{\mathcal{O}}(C_{\text{vr}}(1+(H\sqrt{S}/\sigma P_{\min}^\star))^2 SAH^3\varepsilon^{-2})$ |
| | **Online, RFL-$\phi$ (Ours)** | Yes | TV | $\tilde{\mathcal{O}}(H^5(\min\{H,1/\sigma\})^2S^2A^2\varepsilon^{-2})$ |
| | | | $\chi^2$ | $\tilde{\mathcal{O}}(H^5(1+\sqrt{\sigma})^2S^2A^2\varepsilon^{-2})$ |
| | | | KL | $\tilde{\mathcal{O}}(H^5\sigma^2S^2A^2\varepsilon^{-2})$ |
| | Lower Bound, (Lu et al., 2024) | Yes | TV | $\tilde{\Omega}(H^3 \min\{H,1/\sigma\}SA\varepsilon^{-2})$ |
| | Lower Bound, (Ghosh et al., 2026) | Yes | $\chi^2$ | $\tilde{\Omega}(H^5(1+\sigma)SA\varepsilon^{-2})$ |
| | | | KL | $\tilde{\Omega}(H^5SA\varepsilon^{-2}/(P_{\min}^\star\sigma^2))$ |
| **Linear** | Online, (He et al., 2023) | No | – | $\tilde{\mathcal{O}}(d_{\text{lin}}^2 H^4\varepsilon^{-2})$ |
| | Online, (Liu et al., 2024) | Yes | TV | $\tilde{\mathcal{O}}(d_{\text{lin}}^2 H^3(\min\{H,1/\sigma\})^2\varepsilon^{-2})$ |
| | Hybrid, (Panaganti et al., 2024) | Yes | TV | $\tilde{\mathcal{O}}(\max\{C^2(\pi^\star),1\} d_{\text{lin}}^3 H^3(\lambda+H)^2\varepsilon^{-2})$ |
| | **Online, RFL-$\phi$ (Ours)** | Yes | TV | $\tilde{\mathcal{O}}(d_{\text{lin}}^2 H^5(\min\{H,1/\sigma\})^2\varepsilon^{-2})$ |
| | | | $\chi^2$ | $\tilde{\mathcal{O}}(d_{\text{lin}}^2 H^5(1+\sqrt{\sigma})^2\varepsilon^{-2})$ |
| | | | KL | $\tilde{\mathcal{O}}(d_{\text{lin}}^2 H^5\sigma^2\varepsilon^{-2})$ |
| | Lower Bound (Liu et al., 2024) | Yes | TV | $\tilde{\Omega}(d^2 d_{\text{lin}}^2 H^2(\min\{H,1/\sigma\})^2\varepsilon^{-2})$ |

introducing several new technical challenges. (i) The robust Bellman operator is inherently *non-linear*, so Bellman errors can no longer be analyzed via standard variance-style concentration arguments. Instead, error propagation requires controlling a form of *functional transfer* between value functions and dual variables. (ii) Confidence sets and exploration bonuses must simultaneously control *statistical estimation error* and *adversarial model shift* induced by the worst-case kernel. In contrast, non-robust analyses only account for sampling noise. (iii) The mismatch between nominal and worst-case dynamics necessitates new structural control beyond classical realizability or completeness, even when the underlying function classes are well-behaved.

To address these challenges, we develop new concentration arguments that commute with the supremum over models, introduce new optimism–pessimism couplings to control duality gaps, and formalize a *robust BE dimension* that captures the intrinsic difficulty of learning under model uncertainty. These ingredients lead to an analysis and algorithmic framework that is genuinely distinct from—and not derivable from—existing non-robust online RL theory.

## B. Comparison

Table 1 summarizes our theoretical guarantees of RFL-$\phi$ of $\phi$-divergence uncertainty set, and positions them relative to existing results across three regimes. First, in the *general-function, purely online* setting, our bound is governed by an intrinsic BE-style complexity measure and makes the TV/$\chi^2$/KL uncertainty dependence explicit. Second, in *tabular* RMDPs, our specialization recovers the same TV uncertainty scaling achieved by near-optimal tabular analyses, while providing parallel $\chi^2$/KL scalings in the same uncertainty-multiplier format. Third, in *linear* RMDPs, the table should be read as a *reduction*: under small robust BE dimension, our general theorem yields the stated linear rates, consistent with BE-dimension-based comparisons and without implying universal learnability of arbitrary online linear RMDPs (Liu et al.,

2024; Jin et al., 2021).

**General function approximation: robust extension of BE-style characterizations.** In non-robust online RL with general function approximation, the sharpest comparisons are typically phrased in terms of intrinsic complexity (e.g., BE/DE dimension) rather than explicit $S, A$ dependence. In particular, the BE-dimension framework yields sample complexity of order $\widetilde{O}\big(H^3 \dim_{\mathrm{BE}} \log(|\mathcal{F}|/\delta)\,\varepsilon^{-2}\big)$ for learning near-optimal policies in low-BE problems (Jin et al., 2021). Motivated by BE-dimension analyses, we characterize exploration through the complexity of Bellman errors: we work with the *robust* Bellman residual class and derive a regret bound that scales with the resulting robust BE dimension $\dim_{\mathrm{BE}}^{\mathrm{rob}}$ (Definition 4).

A second natural comparator in the general-function online literature is work that incurs explicit *coverage/concentrability* factors (often denoted $C_{\mathrm{cov}}$) in sample complexity. Table 1 includes such a representative non-robust baseline with complexity $\widetilde{O}\big(C_{\mathrm{cov}}H^3 \log(|\mathcal{F}|/\delta)\varepsilon^{-2}\big)$. Relative to these baselines, our contribution is not to "beat" non-robust learning in the worst case, but to show that robust learning remains sample-efficient under *the same type of intrinsic characterization* (robust BE dimension), with an additional and interpretable $\sigma$-dependent multiplier that reflects the uncertainty set geometry (TV/$\chi^2$/KL).

**Interpreting the $\sigma$-dependence: matching the qualitative "robustness helps" effect in tabular RMDPs.** A central insight from recent tabular RMDP analyses under TV uncertainty is that robustness can fundamentally alter the statistical difficulty of learning: the minimax sample complexity depends explicitly on the uncertainty radius, and for constant uncertainty the problem can be strictly easier than standard (non-robust) MDP learning (Shi et al., 2023; Lu et al., 2024; Liu et al., 2024). This behavior is reflected in our TV specialization through the factor $\min\{H, 1/\sigma\}$ in Table 1, where larger uncertainty induces contraction of the robust Bellman operator and leads to improved statistical rates. Recent work further establishes near-optimal rates for online tabular RMDPs with explicit uncertainty dependence (Lu et al., 2024; Liu et al., 2024), and our tabular reduction recovers the same qualitative scaling. The distinction lies in the analytical structure rather than the rate: existing results derive tabular guarantees through tabular-specific arguments, whereas our bound arises as a specialization of a unified robust BE-dimension framework by upper bounding $\dim_{\mathrm{BE}}^{\mathrm{rob}} = \mathcal{O}(S^2 A^2)$, while preserving a formulation that extends naturally beyond the tabular setting.

For $\chi^2$ and KL uncertainty sets, existing tabular results often involve additional distribution-dependent quantities (e.g., variance or minimum transition mass). In contrast, we present bounds in a uniform and comparable form across divergences, isolating the uncertainty contribution as $(1 + \sqrt{\sigma})$ for $\chi^2$ and $\sigma$ for KL. This representation is deliberate: it highlights how different $\phi$-divergences induce different degrees of smoothing in the dual formulation, and hence lead to quantitatively distinct statistical regimes, consistent with the divergence-sensitive comparisons emphasized in recent $\phi$-robust RL analyses (Panaganti et al., 2024).

**Comparison with non-robust linear MDPs.** In the standard (non-robust) online linear MDP setting, algorithms such as UCRL-VTR and its refinements (He et al., 2023) achieve the minimax regret rate $\widetilde{\mathcal{O}}(\sqrt{d_{\mathrm{lin}}^2 H^3 K})$. In contrast, when specializing our general bound to linear RMDP-$\phi$ for $\phi$-divergence set, we obtain the regret bound $\widetilde{\mathcal{O}}\big(d_{\mathrm{lin}}^2 H^2 B_\phi^2(\sigma)K\big)$, (see Theorem 2). For instance, the regret bound o TV-divergence is $\widetilde{\mathcal{O}}\big(d_{\mathrm{lin}}^2 H^4 (\min\{H, 1/\sigma\})^2 K\big)$ which matches the optimal dependence on the feature dimension $d_{\mathrm{lin}}$ and the uncertainty-dependent factor $\min\{H, 1/\sigma\}$ that appears in recent robust linear analyses, while incurring a higher-order dependence on the horizon $H$ due to the need to control robust Bellman errors under function approximation. Accordingly, we do not claim minimax optimality in $H$, but emphasize that the dimension and uncertainty scalings are consistent with known robust lower bounds.

**Comparison with linear RMDPs.** Several recent works study DR-RL under linear structure but under settings that are not directly comparable to ours. Offline DR-RL with linear function approximation is studied in (Ma et al., 2022) and (Wang et al., 2024a) where value-estimation rates of order $\widetilde{\mathcal{O}}(\sqrt{d_{\mathrm{lin}}/N})$ or $\widetilde{\mathcal{O}}(\sqrt{d_{\mathrm{lin}}^3/N})$ are obtained depending on coverage assumptions and where $N$ denotes the number of trajectories. These results operate in an offline regime and do not address online exploration. In the *online* setting, (Liu & Xu, 2024b) and (Liu et al., 2024) study $d_{\mathrm{lin}}$-rectangular linear RMDPs for TV-divergence set where the agent interacts online with a nominal (source) environment but the performance criterion is the worst-case value over a perturbed (target) environment, and attains regret rate $\widetilde{\mathcal{O}}\big(\sqrt{d_{\mathrm{lin}}^2 H^2 (\min\{H, 1/\sigma\})^2 K}\big)$ together with an information-theoretic lower bound that is optimal in $(d_{\mathrm{lin}}, K, \sigma)$ up to a $\sqrt{H}$ factor (Liu et al., 2024). When specialized to the same $d_{\mathrm{lin}}$-rectangular linear setting, our framework yields the regret bound $\widetilde{\mathcal{O}}\big(\sqrt{d_{\mathrm{lin}}^2 H^4 (\min\{H, 1/\sigma\})^2 K}\big)$, for the TV-divergence set (see Table 1). While this introduces an additional polynomial dependence on the horizon $H$

compared to the most refined linear-specific analyses, it recovers the same dependence on the feature dimension $d_{\text{lin}}$ and the uncertainty-dependent contraction factor $\min\{H, 1/\sigma\}$.

**Remark 7.** *(Panaganti et al., 2024) studies a hybrid $\varphi$-regularized RMDP under TV-divergence set that combines an offline dataset with online interactions. Under approximate value and dual realizability, a bilinear model of dimension $d_{\text{lin}}$, and an offline concentratability coefficient $C(\pi^\star)$, they obtain a suboptimality bound of order $\mathcal{O}\Big(\max\{C(\pi^\star), 1\}(\lambda + H)\sqrt{d_{\text{lin}}^3 H^2 K}\Big)$. By contrast, for TV-divergences set we specialize our general theorem to a $d_{\text{lin}}$-rectangular linear TV-RMDP and show that RFL-TV achieves robust regret $\widetilde{\mathcal{O}}\Big(\sqrt{H^4(\min\{H, 1/\sigma\})^2 d_{\text{lin}}^2 K}\Big)$, which is better than the one in (Panaganti et al., 2024). This comparison implies our algorithm achieves a tighter sample complexity, even without any prior collected offline dataset.*

*We also highlight that, conceptually, the two setups address different questions and are not directly comparable. (Panaganti et al., 2024) analyze a regularized robust objective in a hybrid offline–online regime, where the parameter $\lambda$ controls a trade-off induced by a $\varphi$-regularizer and the guarantees depend on offline coverage through $C(\pi^\star)$. In contrast, we study a constrained RMDP-$\phi$ in a purely online (off-dynamics) setting, where robustness is enforced via an explicit divergence ball of radius $\sigma$ around the nominal model and performance is measured by cumulative regret with respect to the unconstrained $\phi$-divergence robust value. Our general Theorem 1 applies to arbitrary parametric function classes.*

## C. Numerical Experiments

### C.1. Experimental Setup on CartPole

**Environment.** We consider the standard `CartPole-v1` benchmark with a discrete action space. The state $s \in \mathbb{R}^4$ contains the cart position, cart velocity, pole angle, and pole angular velocity, and the action space is $\mathcal{A} = \{0, 1\}$, corresponding to applying a fixed horizontal force to the left or right. Episodes terminate either when the pole falls beyond the allowed angle or when the time limit is reached (maximum horizon $H = 500$). Rewards are the standard per-step rewards from the environment, and the agent aims to maximize the undiscounted return over each episode.

**Robustness evaluation.** We are interested in how the learned policies behave under several kinds of mismatch between training and test conditions. Policies are always *trained on the nominal environment* and are evaluated under the following perturbation families, applied only at evaluation time:

- **Action perturbation.** At each time step, with probability $\rho \in [0, 1]$ the environment ignores the agent's action and instead executes a uniformly random action in $\mathcal{A}$. We evaluate over a grid of perturbation levels and, for the plots in the main text, we focus on
$$\Gamma_{\text{act}} = \{0.3, 0.4, 0.5, 0.6, 0.7, 0.8, 0.9, 1.0\},$$
where $\rho = 0$ corresponds to the nominal case (used internally for sanity checks but not always displayed in the figures).

- **Force-magnitude perturbation.** The horizontal push force applied in the dynamics is multiplied by a scalar factor $\eta_{\text{force}}$. We evaluate the learned policies on a finite set of scale values $\eta_{\text{force}} \in \Gamma_{\text{force}}$ that includes values
$$\Gamma_{\text{force}} = \{0.8, 0.7, 0.6, 0.5, 0.4, 0.3, 0.2, 0.1, 0.0\},$$
where smaller values correspond to progressively weaker control inputs, and $\eta_{\text{force}} = 1.0$ is the nominal strength (used for training but not repeated in this sweep).

- **Pole-length perturbation.** The physical pole length is multiplied by a scalar factor $\eta_{\text{len}}$. At the configuration level, we specify the effective evaluation grid as
$$\Gamma_{\text{len}} = \{0.25, 0.5, 0.75, 1.0, 1.25, 1.5, 1.75, 2.0\},$$
covering shorter and longer poles relative to the nominal length.

In all settings, training is performed on the nominal environment ($\rho = 0, \eta_{\text{force}} = 1, \eta_{\text{len}} = 1$), while robustness is measured by evaluating the final policy on perturbed environments from the above families. Unless otherwise stated, each reported return corresponds to the average over 20 evaluation episodes and 3 independent random seeds $\{0, 1, 2\}$; we also plot $95\%$ confidence intervals computed across seeds and episodes.

*Table 2.* Training hyper-parameters for RFL-TV on `CartPole-v1`.

| Parameter | Symbol | Value |
|---|---|---|
| Discount factor | $\gamma$ | 0.99 |
| Target update rate | $\tau$ | 0.005 |
| Replay buffer size | $|\mathcal{D}|$ | $2 \times 10^5$ transitions |
| Mini-batch size | $B$ | 256 |
| Q-network learning rate | $lr_Q$ | $3 \times 10^{-4}$ |
| Dual-network learning rate | $lr_g$ | $3 \times 10^{-4}$ |
| Exploration start | $\varepsilon_{\text{start}}$ | 1.0 |
| Exploration end | $\varepsilon_{\text{end}}$ | 0.05 |
| Epsilon decay horizon | $T_\varepsilon$ | 200 episodes |
| Gradient updates per step | – | 1 |
| Training episodes | $K$ | 500 |
| Evaluation episodes per configuration | – | 20 |
| Random seeds | – | $\{0, 1, 2\}$ |
| TV-robustness radii | $\sigma$ | $\{0.0, 0.2, 0.3, 0.4, 0.5, 0.6\}$ |
| Slack parameter | $\beta$ | $\{0.0, 0.5, 1.0\}$ |

**Practical RFL-$\phi$ agent.** For CartPole we use a purely value-based implementation of RFL-$\phi$ with a discrete action space. The agent maintains two Q-networks $Q_1, Q_2$ (for Double Q-learning) and their target copies $\bar{Q}_1, \bar{Q}_2$, together with a dual network $g$ and its target copy $\bar{g}$. All networks are multilayer perceptrons with ReLU activations:

- **Q-networks.** We maintain two Q-networks $Q_1$ and $Q_2$. Each network takes the state $s \in \mathbb{R}^4$ as input and outputs a vector in $\mathbb{R}^{|\mathcal{A}|}$, one value per discrete action ($|\mathcal{A}| = 2$ for CartPole). The architecture is a two-layer fully connected MLP with hidden sizes $(128, 128)$ and ReLU activations, followed by a linear output layer. The scalar value $Q_i(s, a)$ is obtained by indexing the corresponding component of this output vector.

- **Dual network.** The dual function $g(s, a)$ is parameterized by a network with the same backbone as the Q-networks: it takes $s \in \mathbb{R}^4$ as input, passes it through two fully connected ReLU layers with $(128, 128)$ units, and produces a vector in $\mathbb{R}^{|\mathcal{A}|}$, one value per action. The output is passed through a sigmoid and scaled so that $g(s, a) \in [0, 10]$ for all $(s, a)$, enforcing non-negativity and preventing numerical blow-up in the dual updates.

**Training protocol and robustness hyper-parameters.** We have considered **RFL-TV** to apply for the experiment. RFL-TV is trained off-policy on `CartPole-v1` using a replay buffer and an $\varepsilon$-greedy exploration strategy. Unless otherwise specified, we fix the discount factor to $\gamma = 0.99$ and use soft target updates with rate $\tau = 0.005$ for all target networks. Transitions are stored in a replay buffer of size $2 \times 10^5$, from which we sample mini-batches of size 256 and perform one gradient update per environment step. The Q-networks and dual network are optimized with Adam at a learning rate of $3 \times 10^{-4}$. Exploration uses $\varepsilon$-greedy action selection, where the exploration rate is initialized at $\varepsilon_{\text{start}} = 1.0$ and decayed linearly to $\varepsilon_{\text{end}} = 0.05$ over the first 200 episodes, and then held fixed at 0.05 for the remainder of training. Each configuration is trained for $K = 500$ episodes, and we report performance statistics over three random seeds $\{0, 1, 2\}$. The robust RFL-TV backup is parameterized by a TV-radius $\sigma \in [0, 1]$ and a slack parameter $\beta \geq 0$ that controls how strictly the dual constraint is enforced. On CartPole, we conduct an ablation study over the robustness radius $\sigma \in \{0.0, 0.2, 0.3, 0.4, 0.5, 0.6\}$ and the slack parameter $\beta \in \{0.0, 0.5, 1.0\}$. The parameter $\sigma$ controls the size of the TV uncertainty set, while $\beta$ controls how strictly the empirical dual Bellman constraint is enforced in the confidence-set construction. After normalizing rewards and values so that the dual residual has typical scale $\mathcal{O}(1)$, the choices $\beta = 0, 0.5, 1.0$ correspond to hard, moderately relaxed, and more relaxed constraint enforcement, respectively. The numerical values of all optimization hyper-parameters and network architectures are summarized in Tables 2 and 3.

**Practical RFL-TV update (CartPole, discrete).** For completeness, Algorithm 2 summarizes the training loop for the discrete practical RFL-TV agent used in the CartPole experiments. The pseudocode follows our implementation: we use Double Q-learning with a dual network that approximates the robust inner optimization under total variation, and we incorporate the slack parameter $\beta$ by clipping the dual residual inside a quadratic penalty.

*Table 3.* Network architectures for RFL-TV on CartPole.

| Network | Hidden layers |
| --- | --- |
| Q-network $Q_1, Q_2$ | $(128, 128)$ (ReLU) |
| Dual network $g$ (default) | $(128, 128)$ (ReLU) |
| Dual network $g$ (capacity sweep) | $(64, 64)$/ $(128, 128)$/ $(256, 256)$ (ReLU) |

## C.2. Practicality, Runtime, and Sensitivity Analysis

To further evaluate the practical behaviour of RFL-TV beyond the main learning curves reported in Section 6, we provide additional analyses on: (i) sensitivity to the robustness hyperparameters, (ii) robustness under different perturbation regimes, and (iii) computational complexity and runtime overhead. Together, these experiments help clarify the stability, scalability, and practical implementability of the proposed robust fitted-learning framework.

**Sensitivity analysis and robustness behaviour.** Table 4 summarizes the sensitivity of RFL-TV to $(\sigma, \beta)$ under three perturbation families: action perturbation, force-magnitude scaling, and pole-length scaling. For each perturbation family, the weak/moderate/severe settings correspond to increasing perturbation strengths, as specified in Table 4. Overall, the results show that the dependence on $(\sigma, \beta)$ is structured and interpretable rather than brittle.

The robustness radius $\sigma$ is the dominant factor controlling robustness. Across all perturbation families, increasing $\sigma$ generally improves performance under stronger distribution shifts, with the effect being particularly pronounced for force-magnitude and pole-length perturbations. This behavior is consistent with the role of $\sigma$, since larger uncertainty sets allow the robust Bellman update to better hedge against environment mismatch.

In contrast, the slack parameter $\beta$ has a more nuanced effect. Under stochastic action perturbations, stricter constraint enforcement ($\beta = 0$) consistently achieves the best performance. Intuitively, when the perturbation mainly introduces random action noise, enforcing the robust Bellman constraint more aggressively improves stability and prevents excessive optimism. However, under structured dynamics shifts such as force-magnitude and pole-length perturbations, a moderate relaxation ($\beta = 0.5$) yields the strongest and most stable performance. In these cases, allowing a small amount of flexibility in the dual constraint improves adaptability to systematic dynamic mismatches while still maintaining robustness.

Based on this sensitivity analysis, we use $\beta = 0$ for action perturbations and $\beta = 0.5$ for force-magnitude and pole-length perturbations in the main experiments. Overall, the results indicate that the proposed framework remains stable across a broad range of hyperparameter choices while exhibiting clear and interpretable robustness trends.

**Computational complexity and practical scalability.** We further discuss the computational complexity and practical scalability of RFL-TV. Relative to standard fitted RL methods, our robust formulation introduces two additional components: (i) the dual-function optimization used to approximate the robust Bellman backup, and (ii) the confidence-set update based on the dual robust Bellman residual. When the value class $\mathcal{F}$ and dual class $\mathcal{G}$ are parameterized by neural networks, both components reduce to standard empirical risk minimization problems and can be implemented efficiently using gradient-based optimization. Consequently, the per-update computational cost scales primarily with the number of collected samples and the neural network size, rather than explicitly with the number of states or state-action pairs.

This provides a significant computational advantage over tabular robust RL methods. In tabular approaches, robust Bellman updates typically require solving separate optimization problems for each $(s, a)$, leading to computational complexity that scales polynomially with the discretized state-action space. In contrast, RFL-TV replaces these local robust optimizations with a single global dual regression problem over the function class $\mathcal{G}$, while confidence sets are constructed directly in function space. As a result, the algorithm scales with the complexity of the chosen function approximators rather than with $|\mathcal{S}|$ or $|\mathcal{A}|$, making it more suitable for large-scale or continuous-state settings.

To empirically evaluate this computational overhead, we compare the wall-clock training time of RFL-TV with DQN under identical neural architectures, optimization settings, hardware, and training budgets. The results are reported in Table 5. We observe that RFL-TV incurs only a moderate runtime overhead relative to DQN, with an average runtime ratio of approximately $1.5\times$. This additional cost mainly arises from maintaining the extra dual network and performing one additional optimization step per update. Importantly, despite this additional computation, the method remains practically implementable and computationally efficient in neural-network-based settings, while simultaneously providing robustness to

*Table 4.* Sensitivity of RFL-TV to robustness radius $\sigma$ and slack parameter $\beta$ across perturbation families. Each entry reports the average episode return after training for 500 episodes, averaged over multiple seeds. Performance is evaluated under three perturbation levels: *weak*, *moderate*, and *severe*. For action perturbations, these correspond to random action probabilities $\rho = \{0.3, 0.6, 0.9\}$; for force-magnitude perturbations, to actuation reductions of $\{20\%, 50\%, 80\%\}$; and for pole-length perturbations, to scaling factors $\{0.25, 0.75, 1.25\}$. Bold entries denote the best $(\sigma, \beta)$ configuration within each perturbation family.

| Perturbation | Uncertainty Radius ($\sigma$) | Slack Parameter ($\beta$) | Weak | Moderate | Severe |
|---|---|---|---|---|---|
| Action | 0.0 | 0.0 | 26.45 | 21.30 | 20.05 |
| | | 0.5 | 20.08 | 17.27 | 15.33 |
| | | 1.0 | 23.90 | 18.00 | 16.07 |
| | 0.4 | 0.0 | 148.56 | 47.34 | 26.51 |
| | | 0.5 | 217.41 | 42.28 | 27.44 |
| | | 1.0 | 238.95 | 50.39 | 27.13 |
| | **0.6** | **0.0** | **335.91** | **62.53** | **24.11** |
| | | 0.5 | 42.76 | 26.76 | 22.14 |
| | | 1.0 | 275.86 | 52.69 | 25.66 |
| Force-magnitude | 0.0 | 0.0 | 35.75 | 30.20 | 26.58 |
| | | 0.5 | 34.75 | 22.12 | 17.30 |
| | | 1.0 | 21.18 | 15.65 | 12.57 |
| | 0.3 | 0.0 | 193.45 | 181.90 | 108.63 |
| | | 0.5 | 37.48 | 27.30 | 26.50 |
| | | 1.0 | 85.35 | 84.88 | 57.80 |
| | **0.5** | 0.0 | 300.00 | 297.85 | 124.20 |
| | | **0.5** | **428.75** | **382.80** | **170.22** |
| | | 1.0 | 280.00 | 166.05 | 101.20 |
| Pole-length | 0.0 | 0.0 | 8.28 | 18.00 | 29.68 |
| | | 0.5 | 12.13 | 18.13 | 21.65 |
| | | 1.0 | 7.93 | 9.82 | 13.12 |
| | 0.3 | 0.0 | 350.77 | 289.77 | 286.98 |
| | | 0.5 | 42.73 | 54.85 | 31.32 |
| | | 1.0 | 133.05 | 84.48 | 88.08 |
| | **0.5** | 0.0 | 463.80 | 439.55 | 422.23 |
| | | **0.5** | **498.37** | **500.00** | **496.00** |
| | | 1.0 | 401.72 | 386.23 | 300.00 |

environment mismatch that standard non-robust methods lack.

## C.3. RFL-TV vs. Functional Approximation Benchmarks: Gains Under Shift

Figure 2 compares RFL-TV to three function-approximation baselines: DQN, the value-function method GOLF (Xie et al., 2022), and a dual-augmented variant GOLF-DUAL, which shares the same dual architecture as RFL-TV but is run with $\sigma = 0$. All three baselines are trained without explicit distributional robustness and thus correspond to the non-robust ($\sigma = 0$) setting. For RFL-TV, we fix the uncertainty radius to the value that achieves the best nominal CartPole performance on our $\sigma$ grid, selecting $\sigma = 0.6$ for action perturbations, $\sigma = 0.5$ for force-magnitude perturbations, and $\sigma = 0.5$ for pole-length perturbations.

Across all three perturbation families, RFL-TV (with its best-performing $\sigma > 0$) consistently dominates the non-robust functional approximation baselines. Under *action perturbations*, for moderate noise levels $\rho \in [0.2, 0.5]$, RFL-TV achieves roughly 30–60% higher average return than DQN and about 15–30% higher than the best non-robust value-based baseline, with performance at $\rho \approx 0.3$ nearly twice that of DQN. For *force-magnitude shifts* of 40–80% from nominal, RFL-TV maintains average returns of roughly 150–400, while DQN stays below about 260 and GOLF/GOLF-DUAL lie mostly in

*Table 5.* Runtime comparison between RFL-TV and DQN measured in training wall-clock time (seconds) over a fixed budget of 500 episodes under identical architectures, optimization settings, and hardware. Results are reported as mean ± standard deviation across multiple seeds. The runtime ratio is defined as Ratio = $\frac{\mathbb{E}[\text{RFL-TV time}]}{\mathbb{E}[\text{DQN time}]}$. Experiments are conducted under three perturbation settings: action noise ($\rho = 0.0$), force-magnitude scaling (1.0), and pole-length scaling (1.0).

| Perturbation | RFL-TV | DQN | Ratio ($\times$) |
|---|---|---|---|
| Action | $247.21 \pm 164.49$ | $110.39 \pm 5.56$ | 2.24 |
| Force | $210.11 \pm 140.06$ | $220.47 \pm 138.41$ | 0.95 |
| Pole-Length | $106.77 \pm 55.15$ | $72.70 \pm 5.47$ | 1.47 |
| **Average Ratio** | — | — | **1.55** |

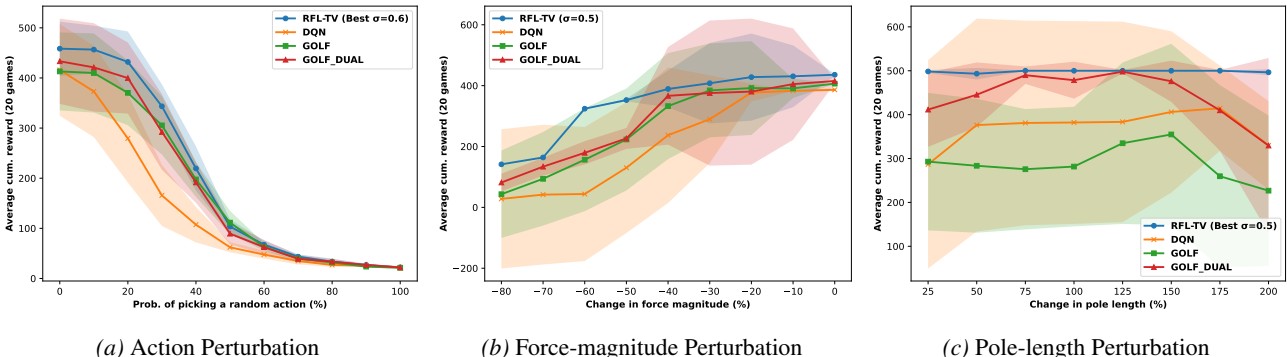

*(a)* Action Perturbation      *(b)* Force-magnitude Perturbation      *(c)* Pole-length Perturbation

*Figure 2.* RFL-TV vs. Functional Approximation Algorithms

the 60–380 range, corresponding to roughly $\approx 1.5$–$3\times$ higher returns than DQN at severe shifts ($\geq 60\%$) and typically a 5–15% gain over the GOLF baselines around the 40–50% shift region. For *pole-length changes* between 25% and 200% of nominal, RFL-TV stays near 500 reward throughout, while the best non-robust baseline ranges between $\approx 330$ and 480, yielding about 5–50% higher return depending on the shift. Overall, for a fixed function class, turning on robustness in the Bellman update (via $\sigma > 0$ and the dual term) yields substantially better robustness to both action noise and dynamics misspecification than any of the non-robust functional approximation baselines. These trends also highlight that robustness is inherently $\sigma$-dependent: for a fixed training robustness level, performance eventually degrades as the test-time perturbation grows, so maintaining high returns under stronger shifts typically requires training with a larger $\sigma$ and, in practice, possibly a more expressive function class.

**C.4. RFL-TV vs. Online Tabular TV-RMDP**

Figure 1 evaluates how closely our practical RFL-TV implementation matches an ideal TV-robust planner by comparing it to OPROVI-TV (Lu et al., 2024), a tabular algorithm that exactly solves the TV-robust Bellman equations for a given radius $\sigma$. Although OPROVI-TV is restricted to small state spaces such as CartPole, it serves as a strong oracle-style baseline for TV-robust planning. In contrast, our practical RFL-TV implementation operates with neural function classes and sample-based updates, so its per-iteration computational cost depends on the network sizes, batch size, and action-space cardinality $A$, but *not* on the number of states $S$, making it applicable to large-scale problems where typically $S \gg A$. Across action perturbations and dynamics perturbations (force magnitude and pole length), RFL-TV with $\sigma \in \{0.4, 0.6\}$ consistently matches, and often exceeds the returns of OPROVI-TV at the same $\sigma$.

For action perturbations (random-action probability $\rho \in [0.3, 0.7]$), RFL-TV with $\sigma = 0.6$ achieves between roughly 100% and 400% higher average return than OPROVI-TV, while $\sigma = 0.4$ yields gains on the order of 30%–200% depending on the noise level; the two methods converge to similar near-random performance only as $\rho$ approaches 1. Under force-magnitude perturbations, RFL-TV with $\sigma = 0.6$ improves over OPROVI-TV by about 100%–300% at large changes (40%–80% deviation from nominal), and $\sigma = 0.4$ still offers roughly 30%–150% gains. For pole-length perturbations, RFL-TV with $\sigma = 0.6$ maintains returns that are typically 150%–300% higher than the tabular baseline over most of the tested range, whereas $\sigma = 0.4$ yields about 30%–150% improvements. Overall, these trends indicate that a simple two-layer

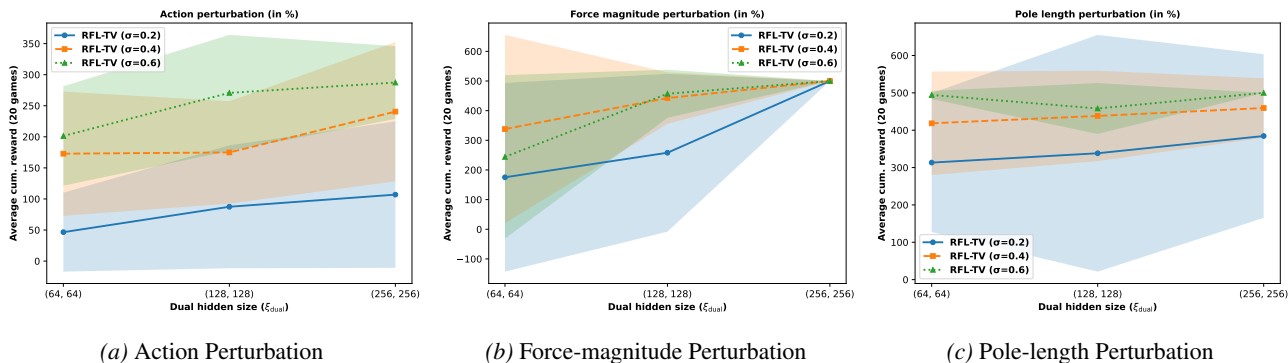

*(a)* Action Perturbation        *(b)* Force-magnitude Perturbation        *(c)* Pole-length Perturbation

*Figure 3.* RFL-TV: uncertainty level $\sigma$ vs. Uniform dual-approximation error $\xi_{\mathrm{dual}}$.

ReLU MLP (with 128–256 hidden units for both Q and dual networks) can closely track—and often outperform—the robust value structure computed by an exact tabular TV-RMDP solver, while enjoying computational complexity that scales with network size and $A$ rather than $S$, which is particularly advantageous in regimes where $S \gg A$.

### C.5. Balancing Robustness Radius and Dual-Network Capacity

Figure 3 examines how the TV robustness radius $\sigma$ and the dual-network width $\xi_{\mathrm{dual}}$ jointly shape the performance of RFL-TV. For each perturbation family (action noise, force–magnitude scaling, and pole–length scaling), we vary $\xi_{\mathrm{dual}}$ over two-layer MLPs with hidden sizes $(64, 64)$, $(128, 128)$, and $(256, 256)$ and evaluate RFL-TV for $\sigma \in \{0.2, 0.4, 0.6\}$ at a representative perturbation level. Note that enlarging the dual hidden size can only decrease the approximation gap $\xi_{\mathrm{dual}}$ to the ideal dual optimizer; in other words, we can view the dual width as a structural knob that monotonically reduces the realizability constant $\xi_{\mathrm{dual}}$. Across all three families, increasing the dual capacity markedly improves robustness: moving from $(64, 64)$ to $(256, 256)$ yields roughly $40\%$–$120\%$ higher average return under action perturbations, about $50\%$–$180\%$ gains for force–magnitude shifts, and roughly $100\%$–$250\%$ gains for pole–length perturbations. At any fixed $\xi_{\mathrm{dual}}$, larger robustness radii clearly help: compared to $\sigma = 0.2$, using $\sigma = 0.6$ improves returns by about $60\%$–$160\%$ under action noise, $30\%$–$80\%$ under force–magnitude changes, and $50\%$–$150\%$ under pole–length changes, with $\sigma = 0.4$ typically lying in between. This behaviour is natural: when $\sigma$ is too small, the uncertainty set remains close to the nominal dynamics and the dual term contributes less, so the policy tends to overfit to the unperturbed environment and degrades sharply under shift. Larger radii ($\sigma \approx 0.4$–$0.6$), together with a sufficiently expressive dual network, force the optimizer to hedge against adversarial transitions, leading to policies that are more conservative around failure modes yet still high-reward under the moderately perturbed environments we evaluate on. In practice, these results suggest a simple tuning recipe: increase $\xi_{\mathrm{dual}}$ until the robust return curve flattens, and select $\sigma$ in a moderate range where performance gains saturate (here around $0.4$–$0.6$), thereby jointly controlling approximation quality and the strength of robustness.

---

**Algorithm 2** Practical RFL-TV for CartPole

---

1: **Inputs:** TV radius $\sigma$, slack $\beta$, discount $\gamma$, target rate $\tau$, batch size $B$, episodes $K$, horizon $H$, exploration schedule $(\varepsilon_{\text{start}}, \varepsilon_{\text{end}}, K_{\text{decay}})$.

2: Initialize replay buffer $\mathcal{D} \leftarrow \emptyset$.

3: Initialize Q-networks $Q_1, Q_2$ and dual network $g$; set target networks $\bar{Q}_i \leftarrow Q_i$ for $i = 1, 2$ (and optionally $\bar{g} \leftarrow g$).

4: **for** $k = 1, \dots, K$ **do**

5:     Set $\varepsilon_k$ by linearly decaying from $\varepsilon_{\text{start}}$ to $\varepsilon_{\text{end}}$ over $K_{\text{decay}}$ episodes, then clamping.

6:     Reset environment and observe $s_0$.

7:     **for** $t = 0, \dots, H - 1$ **do**

8:         With prob. $\varepsilon_k$ sample $a_t$ uniformly; otherwise

$$a_t = \arg\max_a \min\{Q_1(s_t, a),\, Q_2(s_t, a)\}.$$

9:         Execute $a_t$, observe $(r_t, s_{t+1}, d_t)$, and store $(s_t, a_t, r_t, s_{t+1}, d_t)$ in $\mathcal{D}$.

10:         **if** $|\mathcal{D}| \geq B$ **then**

11:           Sample minibatch $\{(s, a, r, s', d)\}_{j=1}^{B}$ from $\mathcal{D}$.

          **\*\*\* Target value (Double Q) \*\*\***

12:           Compute $\bar{Q}_i(s', \cdot)$, $i = 1, 2$, and update

$$v_{\text{next}}(s') = \max_{a'} \min\{\bar{Q}_1(s', a'),\, \bar{Q}_2(s', a')\}.$$

          **\*\*\* Dual update with slack $\beta$ \*\*\***

13:           Evaluate $g(s, a)$ and define

$$\text{dual\_term}(s, a) = \big(g(s, a) - v_{\text{next}}(s')\big)_+ - (1 - \sigma)\, g(s, a).$$

14:           Compute residual

$$r_{\text{dual}}(s, a) = \big|\text{dual\_term}(s, a)\big| - \beta, \quad \tilde{r}_{\text{dual}}(s, a) = \max\{r_{\text{dual}}(s, a), 0\},$$

          and minimize

$$L_g = \mathbb{E}\big[\tilde{r}_{\text{dual}}(s, a)^2\big]$$

          w.r.t. the parameters of $g$ (one gradient step).

          **\*\*\* Q-update using updated $g$ \*\*\***

15:           Recompute

$$\text{dual\_term}^{\text{new}}(s, a) = \big(g(s, a) - v_{\text{next}}(s')\big)_+ - (1 - \sigma)\, g(s, a),$$

          and form targets

$$y = r + (1 - d)\, \gamma \big(v_{\text{next}}(s') + \text{dual\_term}^{\text{new}}(s, a)\big).$$

16:           Compute $Q_i(s, a)$, $i = 1, 2$, and minimize

$$L_Q = \mathbb{E}\big[(Q_1(s, a) - y)^2 + (Q_2(s, a) - y)^2\big]$$

          w.r.t. the parameters of $Q_1, Q_2$ (one gradient step).

17:           Soft-update targets: $\bar{Q}_i \leftarrow (1 - \tau)\, \bar{Q}_i + \tau\, Q_i, \quad i = 1, 2,$
          and optionally: $\bar{g} \leftarrow (1 - \tau)\, \bar{g} + \tau\, g.$

18:         **end if**

19:         **if** $d_t = 1$ **then**

20:           **break**

21:         **end if**

22:     **end for**

23: **end for**

24: **Return** greedy policy: $\pi(s) = \arg\max_a \min\{Q_1(s, a),\, Q_2(s, a)\}.$

---

## D. Robust Bellman Rank and Relations with known tractable classes of Robust RL problems

We define a robust analogue of Bellman rank (Q-type) to connect to known "low-rank" regimes.

**Definition 5** (Robust Bellman rank (Q-type)). We say $\mathcal{F}$ has robust Bellman rank $d$ with normalization parameter $\gamma$ if for each $h$ there exist mappings $\varphi_h : \mathcal{F} \to \mathbb{R}^d$ and $\psi_h : \mathcal{F} \to \mathbb{R}^d$ such that for all $f, f' \in \mathcal{F}$,

$$\varepsilon_h^{\phi,\sigma}(f, f') := \mathbb{E}_{\pi_{f'}} \left[ (f_h - \mathcal{T}_h^{\phi,\sigma} f_{h+1})(s_h, a_h) \right] = \langle \varphi_h(f), \, \psi_h(f') \rangle, \tag{17}$$

and $\|\varphi_h(f)\|_2 \|\psi_h(f')\|_2 \leq \gamma$. The definition is similar to (Jin et al., 2021)[Definition 10] where we replace $\mathcal{T}_h$ by $\mathcal{T}_h^{\phi,\sigma}$.

**Proposition 1** (Low Robust Bellman Rank $\subset$ Low Robust BE Dimension). *If an RMDP-$\phi$ with function class $\mathcal{F}$ has robust Bellman rank $d$ with normalization parameter $\gamma$, then for all $\varepsilon > 0$,*

$$\dim_{\mathrm{BE}}^{\mathrm{rob}}(\mathcal{F}, D_{\mathcal{F}}, \varepsilon) \leq O\big(1 + d \log\big(1 + \gamma/\varepsilon\big)\big). \tag{18}$$

**Justification.** Proposition 1 claims that the problems with low robust Bellman rank have low robust BE dimension to a multiplicative logarithmic factor in $\gamma$ and $\varepsilon^{-1}$. This proposition is the robust analogue of (Jin et al., 2021)[Proposition 11].

**Proof.** The proof of (Jin et al., 2021)[Proposition 11] uses only: (i) the bilinear factorization of the average Bellman error and (ii) norm bounds, and never uses any special structure of the nominal operator beyond appearing inside the residual. Thus, the same determinant-growth proof applies verbatim after substituting $\mathcal{T}_h \mapsto \mathcal{T}_h^{\phi,\sigma}$ and using Definition 5. See (Jin et al., 2021)[Supp., App. D.1] for the full proof.

**Proposition 2** (Low Eluder Dimension $\subset$ Low Robust BE Dimension). *Under Assumption 1 for all $h$, for all $\varepsilon > 0$,*

$$\dim_{\mathrm{BE}}^{\mathrm{rob}}(\mathcal{F}, D_{\Delta}, \varepsilon) \leq \max_{h \in [H]} \dim_{\mathrm{E}}(\mathcal{F}_h, \varepsilon), \tag{19}$$

*where $\dim_{\mathrm{E}}(\mathcal{G}, \varepsilon)$ is the Eluder dimension which is the length of the longest sequence $\{x_1, \cdots, x_n\} \subset \mathcal{X}$ such that there exists $\varepsilon' > \varepsilon$ where $x_i$ is $\varepsilon'$-independent of $\{x_1, \cdots, x_{i-1}\}$ with respect to $\mathcal{G}$ for all $i \in [n]$.*

**Justification.** Proposition 2 shows that any problem class with low Eluder dimension necessarily also has low robust BE dimension. This follows naturally from the completeness property and from the fact that the Eluder dimension is a special case of the DE dimension. This proposition is the robust analogue of (Jin et al., 2021)[Proposition 12].

**Proof.** The proof of (Jin et al., 2021)[Proposition 12] reduces distributional independence for the residual class to Eluder independence within $\mathcal{F}_h$ by defining $g_h^i = \mathcal{T}_h f_{h+1}^i$ and invoking completeness to ensure $g_h^i \in \mathcal{F}_h$. In the robust case, we define the residual class by eq. 8, and define $g_h^i = \mathcal{T}_h^{\phi,\sigma} f_{h+1}^i$ and invoke robust completeness (Assumption 1). No other step changes. See (Jin et al., 2021)[Supp., App. D.2].

**Proposition 3** (Low Robust BE Dimension $\nsubseteq$ Low Eluder Dimension $\cup$ Low Robust BEllman Rank). *For any $m \in \mathbb{N}^+$, there exists an RMDP-$\phi$ with function class $\mathcal{F}$ with $\phi$-divergence uncertainty set such that for all $\varepsilon \in (0, 1]$,*

$$\dim_{\mathrm{BE}}^{\mathrm{rob}}(\mathcal{F}, D_{\mathcal{F}}, \varepsilon) = \dim_{\mathrm{BE}}^{\mathrm{rob}}(\mathcal{F}, D_{\Delta}, \varepsilon) \leq 5, \; but \; \min\left\{ \min_h \dim_{\mathrm{E}}(\mathcal{F}_h, \varepsilon), \; Robust \; Bellman \; Rank \right\} \geq m.$$

**Justification.** Proposition 3 states that the class of problems with low robust Bellman–Eluder (BE) dimension is strictly broader than the union of (i) problems with low Eluder dimension and (ii) problems with low robust Bellman rank. In the non-robust setting, low BE dimension already captures additional models—such as kernel reactive POMDPs—that fall outside both the Bellman rank and Eluder dimension frameworks (Jin et al., 2021)[Appendix C]. The same inclusion continues to hold in the robust setting. This proposition is the robust analogue of (Jin et al., 2021)[Proposition 13].

**Proof.** The separation construction in (Jin et al., 2021)[Proposition 13] is a horizon-1 linear bandit instance. Since, $H = 1$ removes any dependence on transition dynamics in the Bellman operator, adding a $\phi$-divergence ambiguity set does not affect the residual class or the BE-dimension calculation. Thus, the same construction applies unchanged. See (Jin et al., 2021)[Supp., App. D.3].

# E. Proof of the main results

Recall the robust Bellman operator as in eq. 11 as follows:

$$[\mathcal{T}^{\phi,\sigma}f](s,a) = r(s,a) - \inf_{\eta \geq 0,\ \nu \in \mathbb{R}} \mathbb{E}_{s' \sim P_h^\star(s,a)} \left[ \eta\sigma - \nu + \eta\,\phi^\star\left( -\frac{\max_{a'} f(s',a') + \nu}{\eta} \right) \right] \qquad (20)$$

And we define the empirical duality loss as:

$$\widehat{\mathrm{DualLoss}}(g;f) = \sum_{(s,a,s') \sim \mathcal{D}} \left( g_\eta(s,a)\sigma - g_\nu(s,a) + g_\eta(s,a)\,\phi^\star\left( -\frac{\max_{a'} f(s',a') + g_\nu(s,a)}{g_\eta(s,a)} \right) \right), \qquad (21)$$

**Special cases: TV, $\chi^2$, and KL** For concreteness, we recall the resulting one-dimensional variational forms for three choices frequently used in robust RL; detailed derivations can be found in (Yang et al., 2022; He et al., 2025). Under the $\mathcal{S} \times \mathcal{A}$-rectangularity assumption and eq. 20, the robust expectation for any $V : \mathcal{S} \to [0, H]$ and $P_h^\star$ admits the following equivalent forms:

- **TV-divergence** ($\phi(t) = |t - 1|$). Under Assumption 4, in this case, eq. 20 simplifies to

$$\mathbb{E}_{\mathcal{U}_h^{TV,\sigma}(s,a)}[f] = -\inf_{\nu \in [0, 2H/\sigma]} \left\{ \mathbb{E}_{s' \sim \mathbb{P}_h^\star(\cdot|s,a)}\left[ \nu - \max_{a'} f(s',a') \right]_+ + (1 - \sigma)\nu \right\}. \qquad (22)$$

- **$\chi^2$-divergence** ($\phi(t) = (t - 1)^2$). One obtains a variance-sensitive form:

$$\mathbb{E}_{\mathcal{U}_h^{\chi^2,\sigma}(s,a)}[f] = -\inf_{\nu \in [0, H]} \left\{ \sqrt{\sigma\,\mathrm{Var}_{P_h^\star(\cdot|s,a)}\left(\nu - \max_{a'} f(s',a')\right)_+} + \mathbb{E}_{s' \sim \mathbb{P}_h^\star(\cdot|s,a)}\left[ \max_{a'} f(s',a') - \nu \right]_+ \right\}. \qquad (23)$$

- **KL-divergence** ($\phi(t) = t \log t$). The robust expectation can be written as

$$\mathbb{E}_{\mathcal{U}_h^\sigma(s,a)}[V] = -\inf_{\nu \in [\underline{\nu}, H]} \left\{ \nu \log\left( \mathbb{E}_{s' \sim \mathbb{P}_h^\star(\cdot|s,a)}\left[ \exp\{ -\max_{a'} f(s',a')/\nu \} \right] \right) + \nu\sigma \right\}, \qquad (24)$$

where $\underline{\nu} > 0$ is a regularity bound on the optimal dual variable, as commonly assumed in (Blanchet et al., 2023; He et al., 2025).

## E.1. Proof of Lemma 1

*Proof.* Fix $f$. For notational convenience, define $\omega := (s, a)$ and $\Omega := \mathcal{S} \times \mathcal{A}$. Define the pointwise integrand

$$F(\omega; \eta, \nu) := \mathbb{E}_{s' \sim P^\star(\cdot|\omega)}\left[ \eta\sigma - \nu + \eta\,\phi^\star\left( -\frac{\psi_f(s') + \nu}{\eta} \right) \right], \qquad \eta > 0, \nu \in \mathbb{R},$$

and set $F(\omega; \eta, \nu) = +\infty$ whenever $\eta \leq 0$. Then we can rewrite

$$\mathrm{DualLoss}(g;f) = \mathbb{E}_{\omega \sim \mu}\left[ F(\omega; g_\eta(\omega), g_\nu(\omega)) \right]. \qquad (25)$$

**Step 1: Applicability of Rockafellar–Wets Theorem.** We apply Theorem T.2 with $\mathcal{X} = \mathcal{L}^1(\mu; \mathbb{R}^2)$ and integrand $f_{\mathrm{RW}}(\omega, x) = F(\omega; x)$ for $x = (\eta, \nu) \in \mathbb{R}^2$.

*(i) Decomposability.* By Remark 8, $\mathcal{L}^1(\mu)$ is decomposable. Since $\mathcal{L}^1(\mu; \mathbb{R}^2) = \mathcal{L}^1(\mu) \times \mathcal{L}^1(\mu)$, it is decomposable relative to $(\Omega, \Sigma(\Omega), \mu)$ in the sense of Definition 6.

*(ii) Normal integrand.* For each fixed $(\eta, \nu)$ with $\eta > 0$, the map

$$(s', \omega) \mapsto \eta\sigma - \nu + \eta\,\phi^\star\left( -\frac{\psi_f(s') + \nu}{\eta} \right)$$

is measurable since $\psi_f$ is measurable and $\phi^\star$ is lower semicontinuous. Thus, $\omega \mapsto F(\omega; \eta, \nu)$ is measurable.

For each fixed $\omega$, continuity of $(\eta, \nu) \mapsto F(\omega; \eta, \nu)$ on $\eta > 0$ follows from continuity of the perspective transform of $\phi^\star$ and dominated convergence, since $\psi_f \in [0, H]$ and the integrand is finite under standard boundedness assumptions on multipliers. Hence, $F$ is a normal integrand in the sense of Definition 6. Therefore, Theorem T.2 applies and yields

$$\inf_{g \in \mathcal{L}^1(\mu; \mathbb{R}^2)} \mathrm{DualLoss}(g; f) = \inf_g \mathbb{E}_{\omega \sim \mu}[F(\omega; g(\omega))] = \mathbb{E}_{\omega \sim \mu}\left[\inf_{(\eta, \nu) \in \mathbb{R}^2} F(\omega; \eta, \nu)\right]. \tag{26}$$

**Step 2: Identification of the pointwise infimum.** Fix $\omega = (s, a)$. By classical $\phi$-divergence duality (e.g., (Shapiro, 2017; Duchi & Namkoong, 2021)), for any bounded measurable $V$,

$$\inf_{P: D_\phi(P \| P^\star) \leq \sigma} \mathbb{E}_P[V] = \inf_{\eta > 0, \nu \in \mathbb{R}} \left\{\eta\sigma - \nu + \eta \mathbb{E}_{P^\star}\left[\phi^\star\left(-\frac{V + \nu}{\eta}\right)\right]\right\}.$$

Applying this identity with $V = \psi_f$ and $P^\star = P^\star(\cdot | s, a)$ gives

$$\inf_{(\eta, \nu) \in \mathbb{R}^2} F((s, a); \eta, \nu) = \inf_{P \in \mathcal{U}_h^{\phi, \sigma}(s, a)} \mathbb{E}_{s' \sim P(\cdot | s, a)}[\psi_f(s')].$$

**Step 3: Conclusion.** Substituting the above into equation 26, we conclude

$$\inf_{g \in \mathcal{L}^1(\mu; \mathbb{R}^2)} \mathrm{DualLoss}(g; f) = \mathbb{E}_{(s,a) \sim \mu}\left[\inf_{P \in \mathcal{U}_h^{\phi, \sigma}(s, a)} \mathbb{E}_{s' \sim P(\cdot | s, a)}[\psi_f(s')]\right],$$

as claimed. $\qquad\square$

### E.2. Proof of Theorem 1

*Proof.* We prove Theorem 1 in the robust BE dimension (Definition 4) framework. Throughout, all expectations are taken under the distributions induced by the algorithm's policies in *nominal* $P^\star$, consistent with the definition of $\Pi_h$ and the DE dimension (Jin et al., 2021) as given in Definition 3.

**Step 1: Reduce robust regret to robust Bellman residuals under nominal transition kernel.** According to Assumption 1 and Lemma K.1, we can guarantee $f^{(k)}$ is optimistic. Based on this optimistic algorithm, we will now relate the regret to the robust average Bellman error under the learner's sequence of policies. By Lemma K.1, under Assumption 1 and the confidence-set optimism (which ensures $f^{(k)}$ is optimistic in each episode), we have

$$\mathrm{Regret}(K) \leq \sum_{k=1}^K \sum_{h=1}^H \varepsilon_h^{\phi, \sigma}(f^{(k)}, \pi^{(k)}), \qquad \pi^{(k)} := \pi^{f^{(k)}}. \tag{27}$$

Recalling the definition of $\varepsilon_h^{\phi, \sigma}(f^{(k)}, \pi^{(k)})$ as in eq. 67,

$$\varepsilon_h^{\phi, \sigma}(f^{(k)}, \pi^{(k)}) = \mathbb{E}_{\pi^{(k)}}\left[f_h^{(k)}(s_h, a_h) - (\mathcal{T}_h^{\phi, \sigma} f_{h+1}^{(k)})(s_h, a_h)\right],$$

where state–actions at the step-$h$ is induced by executing $\pi^{(k)}$ in the nominal MDP $P^\star$.

**Step 2: Add and subtract the dual-based operator.** For each $(k, h)$, add and subtract the dual-based empirical robust Bellman backup $\mathcal{T}_{h, \underline{g}_{f_{h+1}^{(k)}}}^{\phi, \sigma}$:

$$f_h^{(k)}(s, a) - (\mathcal{T}_h^{\phi, \sigma} f_{h+1}^{(k)})(s, a) = \underbrace{\left(f_h^{(k)}(s, a) - (\mathcal{T}_{h, \underline{g}_{f_{h+1}^{(k)}}}^{\phi, \sigma} f_{h+1}^{(k)})(s, a)\right)}_{\text{dual-based Bellman residual}}$$

$$+ \underbrace{\left((\mathcal{T}_{h, \underline{g}_{f_{h+1}^{(k)}}}^{\phi, \sigma} f_{h+1}^{(k)})(s, a) - (\mathcal{T}_h^{\phi, \sigma} f_{h+1}^{(k)})(s, a)\right)}_{\text{dual approximation error}}. \tag{28}$$

Plugging eq. 28 into eq. 27 yields

$$\text{Regret}(K) \;\le\; \text{I} + \text{II}, \tag{29}$$

where

$$\text{I} := \sum_{k=1}^{K} \sum_{h=1}^{H} \mathbb{E}_{\pi^{(k)}} \Big[ f_h^{(k)}(s_h, a_h) - (\mathcal{T}_{h, \underline{g}_{f_{h+1}^{(k)}}}^{\phi, \sigma} f_{h+1}^{(k)})(s_h, a_h) \Big], \tag{30}$$

$$\text{II} := \sum_{k=1}^{K} \sum_{h=1}^{H} \mathbb{E}_{\pi^{(k)}} \Big[ (\mathcal{T}_{h, \underline{g}_{f_{h+1}^{(k)}}}^{\phi, \sigma} f_{h+1}^{(k)})(s_h, a_h) - (\mathcal{T}_h^{\phi, \sigma} f_{h+1}^{(k)})(s_h, a_h) \Big]. \tag{31}$$

**Step 3: Bound** I **via BE dimension (distributional Eluder dimension).** Fix any step $h \in [H]$ and define the robust Bellman residual class

$$\Xi_h^{\xi} := (\mathcal{I} - \mathcal{T}_h^{\phi, \sigma}) \mathcal{F} = \{ f_h - \mathcal{T}_h^{\phi, \sigma} f_{h+1} : f \in \mathcal{F} \},$$

consistent with eq. 8. Let $\mu_k$ denote the distribution undr policy $\pi^{(k)}$ step-$h$ roll-in distribution under $P^\star$.

Define for each episode $k$ the robust residual function

$$\xi_k(\cdot, \cdot) := f_h^{(k)}(\cdot, \cdot) - \mathcal{T}_{h, \underline{g}_{f_{h+1}^{(k)}}}^{\phi, \sigma} f_{h+1}(\cdot, \cdot) \in \Xi_h^{\xi}.$$

Then the step-$h$ contribution to I can be written as

$$\text{I}_h := \sum_{k=1}^{K} \mathbb{E}_{\mu_k} [\xi_k] \quad \text{and} \quad \text{I} = \sum_{h=1}^{H} \text{I}_h. \tag{32}$$

We now invoke Lemma K.3, which is a direct application of the distributional Eluder dimension summation argument of (Jin et al., 2021)[Lemma 17]. To apply Lemma K.3, we need a squared-control condition of the form $\sum_{t<k} \mathbb{E}_{\mu_t} [\xi_k^2] \le \beta$. This is exactly ensured by Lemma K.2(b) (with the same $\beta$ choice as in the theorem statement), which provides a uniform control on the cumulative squared (dual-based) Bellman residuals; combined with Assumption 1 and the confidence-set validity, this yields the required $\beta$-type bound at each stage (see Lemma K.2 and its proof template following (Jin et al., 2021)).

Therefore, applying Lemma K.3 at each fixed $h$ gives: for all $k \in [K]$,

$$\sum_{i=1}^{k} \big| \mathbb{E}_{\mu_i} [\xi_i] \big| \;\le\; \mathcal{O}\Big( \sqrt{\dim_{\text{DE}}(\Xi_h^{\xi}, \Pi_h, 1/k) \, \beta \, k} \Big).$$

In particular, taking $k = K$ and summing over $h \in [H]$ yields

$$\text{I} \;\le\; \sum_{h=1}^{H} \mathcal{O}\Big( \sqrt{\dim_{\text{DE}}(\Xi_h^{\xi}, \Pi_h, 1/K) \, \beta \, K} \Big). \tag{33}$$

Equivalently, in terms of the robust BE dimension $\dim_{\text{BE}}^{\text{rob}}(\mathcal{F}, \Pi, 1/K) := \max_{h \in [H]} \dim_{\text{DE}}(\Xi_h^{\xi}, \Pi_h, 1/K)$, we may write

$$\text{I} \;=\; \mathcal{O}\Big( H \sqrt{\dim_{\text{BE}}^{\text{rob}}(\mathcal{F}, \Pi, 1/K) \, \beta \, K} \Big). \tag{34}$$

**Step 4: Bound** II **via the TV dual optimization error lemma.** For II we control, for each $(k, h)$,

$$\Delta_{k,h}(s, a) := (\mathcal{T}_{h, \underline{g}_{f_{h+1}^{(k)}}}^{\phi, \sigma} f_{h+1}^{(k)})(s, a) - (\mathcal{T}_h^{\phi, \sigma} f_{h+1}^{(k)})(s, a).$$

Then by definition eq. 31,

$$\text{II} = \sum_{k=1}^{K} \sum_{h=1}^{H} \mathbb{E}_{\pi^{(k)}} \big[ \Delta_{k,h}(s_h^k, a_h^k) \big] \;\le\; \sum_{k=1}^{K} \sum_{h=1}^{H} \| \Delta_{k,h} \|_{1, \mu_h^{\pi^{(k)}}},$$

where $\|\cdot\|_{1,\mu}$ denotes the $\ell_1(\mu)$ seminorm.

Now apply Lemma K.4 with $\pi = \pi^{(k)}$, $\mu_h^\pi$ as the distribution induced by $\pi^{(k)}$, $f_{h+1} = f_{h+1}^{(k)}$, and dataset size $|\mathcal{D}_h^{(k)}| \geq kH$ (one transition per episode per stage). Using a union bound over $(k,h)$ and the standard choice of failure probabilities, we obtain that with probability at least $1 - \delta$,

$$\left\|\Delta_{k,h}\right\|_{1,\mu_h^{\pi^{(k)}}} = \mathcal{O}\left(B_\phi(\sigma)\sqrt{\frac{\log\left(|\mathcal{F}_{h+1}||\mathcal{G}|KH/\delta\right)}{kH}} + \frac{\varepsilon^{\mathrm{dual}}}{KH}\right). \tag{35}$$

Summing eq. 35 over $k \in [K]$ and $h \in [H]$, and using $\sum_{k=1}^K k^{-1/2} \leq 2\sqrt{K}$, gives

$$\mathrm{II} = \mathcal{O}\left(B_\phi(\sigma)\sqrt{KH\log\left(\frac{|\mathcal{F}||\mathcal{G}|KH}{\delta}\right)} + \varepsilon^{\mathrm{dual}}\right) = \mathcal{O}\left(\sqrt{HB_\phi(\sigma)\beta K} + \varepsilon^{\mathrm{dual}}\right). \tag{36}$$

**Step 5: Combine the bounds.** Combining eq. 29, eq. 34, and eq. 36 yields that with probability at least $1 - \delta$,

$$\mathrm{Regret}(K) \leq \mathcal{O}\left(H\sqrt{\dim_{\mathrm{BE}}^{\mathrm{rob}}(\mathcal{F}, \Pi, 1/K)\,\beta\,K} + \sqrt{HB_\phi(\sigma)\beta K} + \varepsilon^{\mathrm{dual}}\right).$$

Finally, by setting $\beta = \mathcal{O}\left(B_\phi(\sigma)\log\left(|\mathcal{F}||\mathcal{G}|KH/\delta\right)\right)$ completes the proof. $\qquad\square$

### E.3. Proof of Corollary 1

For TV, we adopt the standard assumption.

**Assumption 4** (Failure States). *For a TV-RMDP, there exists a set of failure states $\mathcal{S}_F \subseteq \mathcal{S}$, such that $r_h(s,a) = 0$, and $P_h^\star(s'|s,a) = 0$, $\forall a \in \mathcal{A}, \forall s \in \mathcal{S}_F, \forall s' \notin \mathcal{S}_F$.*

*Proof.* Under Assumption 3, we will first find out the value of $B_\phi(\sigma)$ for each divergence. Then, applying the value of $B_\phi(\sigma)$ and $\varepsilon^{dual} = 0$ in Theorem 1, we will find the sample-complexity bound for each case, as follows:

- **TV-Divergence Case:** According to eq. 22, we have

$$l_{\mathrm{TV}}(f; s, a, s'; \nu) \leq B_{\mathrm{TV}} := \mathcal{O}\left(H\min\{H, 1/\sigma\}\right). \tag{37}$$

  Applying eq. 37, $\varepsilon^{dual} = 0$ and $\dim_{\mathrm{BE}}^{\mathrm{rob}} \geq 1$ in Theorem 1, we get the sample-complexity bound as

$$T = KH = \mathcal{O}\left(\frac{H^5\left(\min\{H, 1/\sigma\}\right)^2 d\log\left(|\mathcal{F}||\mathcal{G}|T/\delta\right)}{\varepsilon^2}\right). \tag{38}$$

- **$\chi^2$-Divergence Case:** According to eq. 23, we have

$$l_{\chi^2}(f; s, a, s'; \nu) \leq B_{\chi^2} := \mathcal{O}\left(H(1 + \sqrt{\sigma})\right). \tag{39}$$

  Applying eq. 39, $\varepsilon^{dual} = 0$ and $\dim_{\mathrm{BE}}^{\mathrm{rob}} \geq 1$ in Theorem 1, we get the sample-complexity bound as

$$T = KH = \mathcal{O}\left(\frac{H^5(1 + \sqrt{\sigma})^2 d\log\left(|\mathcal{F}||\mathcal{G}|T/\delta\right)}{\varepsilon^2}\right). \tag{40}$$

- **KL-Divergence Case:** According to eq. 24, we have

$$l_{\mathrm{KL}}(f; s, a, s'; \nu) \leq B_{\mathrm{KL}} := \mathcal{O}\left(H\sigma\right). \tag{41}$$

  Applying eq. 41, $\varepsilon^{dual} = 0$ and $\dim_{\mathrm{BE}}^{\mathrm{rob}} \geq 1$ in Theorem 1, we get the sample-complexity bound as

$$T = KH = \mathcal{O}\left(\frac{H^5\sigma^2 d\log\left(|\mathcal{F}||\mathcal{G}|T/\delta\right)}{\varepsilon^2}\right). \tag{42}$$

$\square$

## F. Specialization to Linear RMDP-$\phi$

We now show that our regret bound for general functional approximation specializes to a near–dimension-optimal bound when the robust value function admits a linear representation, in the spirit of the $d_{\mathrm{lin}}$-rectangular linear RMDP framework of (Ma et al., 2022) and (Liu et al., 2024).

**Assumption 5** ($d_{\mathrm{lin}}$-Rectangular Linear RMDP-$\phi$). *There exists a known feature map $\boldsymbol{\kappa}_h : \mathcal{S} \times \mathcal{A} \to \mathbb{R}^d$ for each $h \in [H]$ with $\sum_{i=1}^{d} \kappa_{h,i}(s,a) = 1$ and $\kappa_{h,i}(s,a) \geq 0$ for any $(i,s,a) \in [d] \times \mathcal{S} \times \mathcal{A}$ such that:*

1. *(Linear nominal model.) The reward and nominal kernel are linear:*

$$r_h(s,a) = \boldsymbol{\kappa}_h(s,a)^\top \boldsymbol{\Omega}_h, \qquad P_h^\star(\cdot \mid s,a) = \boldsymbol{\kappa}_h(s,a)^\top \boldsymbol{\lambda}_h^\star(\cdot),$$

   *for some unknown probability measures $\{\boldsymbol{\lambda}_h^\star\}_{h=1}^H$ over $\mathcal{S}$ and known vectors $\{\boldsymbol{\Omega}_h\}_{h=1}^H$ with $\|\boldsymbol{\Omega}_h\|_2 \leq \sqrt{d_{\mathrm{lin}}}$.*

2. *($d_{\mathrm{lin}}$-rectangular $\phi$-divergence uncertainty set.) For each step $h$ and feature index $i \in [d_{\mathrm{lin}}]$ we can parameterize our uncertainty set $\mathcal{P}$ by $\{\boldsymbol{\lambda}_h^\star\}_{h=1}^H$, and thereby, can be defined as $\mathcal{P} = \mathcal{U}^{\phi,\sigma}(P^\star) = \bigotimes_{(h,s,a) \in [H] \times \mathcal{S} \times \mathcal{A}} \mathcal{U}_h^{\phi,\sigma}(s,a; \boldsymbol{\lambda}_h^\star)$, where $\mathcal{U}_h^{\phi,\sigma}(s,a; \boldsymbol{\lambda}_h^\star)$ is defined as*

$$\mathcal{U}_h^{\phi,\sigma}(s,a; \boldsymbol{\lambda}_h^\star) \triangleq \left\{ \sum_{i=1}^{d} \kappa_{h,i}(s,a) \lambda_{h,i}(\cdot) : \nu_{h,i} \in \Delta(\mathcal{S}) \text{ and } D_\phi(\lambda_{h,i}, \lambda_{h,i}^\star(\cdot|s,a)) \leq \sigma \right\}.$$

This is the $\phi$-divergence analogue of the $d_{\mathrm{lin}}$-rectangular linear RMDP of (Liu et al., 2024)[Sec. 3.2].

**Linear function classes induced by the $d$-Rectangular linear RMDP-$\phi$.** Under the linear RMDP-$\phi$ structure in Assumption 5, we specialize our general functional class $\mathcal{F}$ and dual functional class $\mathcal{G}$ used by RFL-$\phi$ as linear function classes with a common feature map $\boldsymbol{\kappa}_h : \mathcal{S} \times \mathcal{A} \to \mathbb{R}^{d_{\mathrm{lin}}}$, and denote them as follows:

$$\mathcal{F}^{lin} := \{\mathcal{F}_h^{lin}\}_{h=1}^H, \text{ where } \mathcal{F}_h^{lin} := \left\{ f_h : f_h(s,a) = \boldsymbol{\kappa}_h(s,a)^\top \boldsymbol{w}_h, \; \boldsymbol{w}_h \in \mathbb{R}^{d_{\mathrm{lin}}} \right\}, \tag{43}$$

$$\mathcal{G}^{lin} := \{\mathcal{G}_h^{lin}\}_{h=1}^H, \text{ where } \mathcal{G}_h^{lin} := \left\{ g_h = (g_{\eta,h}, g_{\nu,h}) : g_{\eta,h}(s,a) = \boldsymbol{\kappa}_h(s,a)^\top \boldsymbol{u}_{\eta,h}, \right.$$

$$\left. g_{\nu,h}(s,a) = \boldsymbol{\kappa}_h(s,a)^\top \boldsymbol{u}_{\nu,h}, \; \boldsymbol{u}_{\eta,h}, \boldsymbol{u}_{\nu,h} \in \mathbb{R}^{d_{\mathrm{lin}}} \right\}. \tag{44}$$

The class $\mathcal{F}^{lin}$ is used to approximate robust $Q$–functions, while $\mathcal{G}^{lin}$ parameterizes the dual variables $(\eta, \nu)$ appearing in the $\phi$–robust Bellman operator (via the functional dual loss in Eq. 12)[See Sec. 3 for the definition of the dual loss and its empirical counterpart].

**Lemma 3** (Linear realizability and completeness). *Suppose the linear RMDP satisfies Assumption 5. Then:*

1. **Linear realizability of $Q^{\pi,\sigma}$ and $Q^{\star,\sigma}$.** *For any Markov policy $\pi$ and any $\sigma \geq 0$, there exist vectors $\boldsymbol{w}_1^{\pi,\sigma}, \ldots, \boldsymbol{w}_H^{\pi,\sigma} \in \mathbb{R}^{d_{\mathrm{lin}}}$ such that for all $h \in [H]$,*

$$Q_h^{\pi,\sigma}(s,a) = \boldsymbol{\kappa}_h(s,a)^\top \boldsymbol{w}_h^{\pi,\sigma}, \qquad \forall (s,a) \in \mathcal{S} \times \mathcal{A}, \tag{45}$$

   *and, in particular, for the robust-optimal policy $\pi^\star$ there exist $\boldsymbol{w}_1^{\star,\sigma}, \ldots, \boldsymbol{w}_H^{\star,\sigma}$ with*

$$Q_h^{\star,\sigma}(s,a) = \boldsymbol{\kappa}_h(s,a)^\top \boldsymbol{w}_h^{\star,\sigma}, \qquad \forall (s,a), \; h \in [H]. \tag{46}$$

   *Hence $Q^{\pi,\sigma}, Q^{\star,\sigma} \in \mathcal{F}^{lin}$.*

2. **Closure under the robust Bellman operator.** *Let $f \in \mathcal{F}^{lin}$ with component functions $f_h(s,a) = \boldsymbol{\kappa}_h(s,a)^\top \boldsymbol{w}_h$. Then, for each $h \in [H]$ there exists $\boldsymbol{w}_h' \in \mathbb{R}^{d_{\mathrm{lin}}}$ such that the robust Bellman backup satisfies*

$$[\mathcal{T}_h^{\phi,\sigma} f_{h+1}](s,a) = r_h(s,a) + \mathbb{E}_{P \in \mathcal{U}_h^{\phi,\sigma}(s,a)} \left[ V_{h+1}(s') \right]$$

$$= \boldsymbol{\kappa}_h(s,a)^\top \boldsymbol{w}_h', \qquad \forall (s,a), \tag{47}$$

   *so that $\mathcal{T}_h^{\phi,\sigma} f_{h+1} \subseteq \mathcal{F}_h^{lin}$ for all $h$.*

3. **Linear dual representation.** *Fix any $f \in \mathcal{F}^{\mathrm{lin}}$. The dual minimizer $g_f^* = (g_\eta, g_\nu(s,a)) \in \mathbb{R}^2$ that attains the pointwise $\phi$-divergence dual can be cosen. in $\mathcal{G}^{\mathrm{lin}}$, i.e., there exist $\boldsymbol{u}_{\eta,h}^f, \boldsymbol{u}_{\nu,h}^f \in \mathbb{R}^{d_{\mathrm{lin}}}, \forall h \in [H]$ such that for all $(s,a)$ and $h \in [H]$,*

$$g_{f,h}^\star(s,a) := \left(g_{\eta,f,h}^\star(s,a),\, g_{\nu,f,h}^\star(s,a)\right) = \left(\boldsymbol{\kappa}_h(s,a)^\top \boldsymbol{u}_{\eta,h}^f,\ \boldsymbol{\kappa}_h(s,a)^\top \boldsymbol{u}_{\nu,h}^f\right), \qquad \forall(s,a),\ h \in [H]. \tag{48}$$

*Consequently, the dual realizability error $\varepsilon^{\mathrm{dual}}$ in Assumption 2 is zero whenever the dual class used by the algorithm satisfies $\mathcal{G}^{\mathrm{lin}} \equiv \mathcal{L}^1(\mu^\pi; \mathbb{R}^2)$.*

*Proof.* We will now proof Lemma 3. *(i) Linear realizability of $Q^{\pi,\sigma}$ and $Q^{\star,\sigma}$.* The linear RMDP literature (e.g., (Ma et al., 2022)[Prop. 3.2 and Lem. 4.1] and (Liu et al., 2024)[Sec. 3.2]) implies that both the robust Bellman operator and the robust value functions preserve linearity in $\boldsymbol{\kappa}_h$, yielding eq. 45–47, the nominal kernel and all kernels in the $\phi$-divergence uncertainty set are linear mixtures of the base measures $\{\boldsymbol{\lambda}_h\}_{h=1}^H$, and the reward is linear in $\boldsymbol{\kappa}_h$.

*(ii) Closure under $\mathcal{T}_h^{\phi,\sigma}$.* Let $f \in \mathcal{F}^{lin}$ with $f_{h+1}(s,a) = \boldsymbol{\kappa}_{h+1}(s,a)^\top \boldsymbol{w}_{h+1}$. Define the value $V_{h+1}(s) = \max_{a \in \mathcal{A}} f_{h+1}(s,a)$. By the $d_{\mathrm{lin}}$-rectangular structure, any $P \in \mathcal{U}_h^{\phi,\sigma}(s,a)$ can be written as $P(\cdot \mid s,a) = \sum_{i=1}^d \kappa_{h,i}(s,a)\lambda_{h,i}(\cdot)$ with $\lambda_{h,i} \in \mathcal{U}_h^\sigma(s,a; \boldsymbol{\lambda}_h^\star)$. Thus,

$$\inf_{P_h \in \mathcal{U}_h^{\phi,\sigma}(s,a)} \mathbb{E}_{s' \sim P_h(\cdot|s,a)}\left[V_{h+1}^{\pi,\sigma}(s')\right] = \inf_{\lambda_{h,1},\ldots,\lambda_{h,d}} \sum_{i=1}^d \kappa_{h,i}(s,a)\, \mathbb{E}_{s' \sim \mu_{h,i}}[V_{h+1}(s')] \tag{49}$$

$$= \sum_{i=1}^d \kappa_{h,i}(s,a) \inf_{\lambda_{h,i} \in \mathcal{U}_h^{\phi,\sigma}(s,a;\boldsymbol{\lambda}_h^\star)} \mathbb{E}_{s' \sim \nu_{h,i}}[V_{h+1}(s')] \tag{50}$$

$$= \sum_{i=1}^d \kappa_{h,i}(s,a)\, \zeta_{h,i}(\boldsymbol{w}_{h+1}), \tag{51}$$

where each scalar $\zeta_{h,i}(\boldsymbol{w}_{h+1})$ depends only on $V_{h+1}$ (and hence on $\boldsymbol{w}_{h+1}$) and the local $\phi$-divergence ball at index $i$. We therefore obtain

$$[\mathcal{T}_h^{\phi,\sigma} f_{h+1}](s,a) = \boldsymbol{\kappa}_h(s,a)^\top \boldsymbol{\Omega}_h + \boldsymbol{\kappa}_h(s,a)^\top \boldsymbol{\zeta}_h(\boldsymbol{w}_{h+1}) = \boldsymbol{\kappa}_h(s,a)^\top \boldsymbol{w}_h', \tag{52}$$

with $\boldsymbol{w}_h' := \boldsymbol{\Omega}_h + \boldsymbol{\zeta}_h(\boldsymbol{w}_{h+1})$. This yields eq. 47 and shows that $\mathcal{T}_h^{\phi,\sigma} f_{h+1} \subseteq \mathcal{F}_h^{lin}$.

*(iii) Linear dual representation.* Fix any $f \in \mathcal{F}^{\mathrm{lin}}$ and $(s,a,h)$. The $\phi$-divergence dual form of the robust Bellman operator (Eq. 9) expresses the inner worst-case expectation as a two-dimensional convex optimization problem over scalar dual variables $(\eta, \nu)$. In our functional formulation, this corresponds to optimizing over dual functions $g_{f,h}(s,a) = (g_{\eta,h}(s,a), g_{\nu,h}(s,a)) \in \mathbb{R}^2$.

Under the linear RMDP structure, the nominal transition kernel admits the representation

$$P_h^\star(\cdot \mid s,a) = \sum_{i=1}^{d_{\mathrm{lin}}} \kappa_{h,i}(s,a)\lambda_{h,i}^\star,$$

so the dual objective decomposes as a weighted combination over the base measures $\{\lambda_{h,i}^\star\}_{i=1}^{d_{\mathrm{lin}}}$. As a consequence, the pointwise dual minimizer can be chosen to decompose coordinate-wise: there exist scalars $\{\eta_{h,i}^\star, \nu_{h,i}^\star\}_{i=1}^{d_{\mathrm{lin}}}$ such that the optimal dual function admits the representation

$$g_{f,h}^\star(s,a) = \left(g_{\eta,f,h}^\star(s,a),\, g_{\nu,f,h}^\star(s,a)\right) = \left(\boldsymbol{\kappa}_h(s,a)^\top \boldsymbol{u}_{\eta,h}^f,\ \boldsymbol{\kappa}_h(s,a)^\top \boldsymbol{u}_{\nu,h}^f\right),$$

for some vectors $\boldsymbol{u}_{\eta,h}^f, \boldsymbol{u}_{\nu,h}^f \in \mathbb{R}^{d_{\mathrm{lin}}}$ (see, e.g., the TV dual derivation in (Liu et al., 2024)[Sec. 3.2]). Equivalently, $g_f^\star \in \mathcal{G}^{\mathrm{lin}}$ as defined in eq. 48.

Therefore, the infimum in the dual representation is attained within the class $\mathcal{G}^{\mathrm{lin}}$, and the dual realizability error $\varepsilon^{\mathrm{dual}}$ in Assumption 2 is zero whenever the dual class used by the algorithm satisfies $\mathcal{G}^{\mathrm{lin}} \equiv \mathcal{L}^1(\mu^\pi; \mathbb{R}^2))$. $\qquad\square$

**Assumption 6** (Finite linear covering). *For $\varepsilon_0 = 1/K$, the union class $\mathcal{H} = \mathcal{F}^{lin} \cup \mathcal{G}^{lin}$ admits a finite $\varepsilon_0$-cover in $\|\cdot\|_\infty$ such that*

$$\log N_\mathcal{H}(\varepsilon_0) \leq c_o d_{\lin} H \log(c_o K) \tag{53}$$

*for some absolute constant $c_o > 0$.*

This bound follows from standard metric-entropy results for linear predictors on a bounded domain (see, e.g., (Shalev-Shwartz & Ben-David, 2014)[Thm. 14.5]). In our setting, the feature vectors satisfy the simplex constraints $\sum_i \kappa_{h,i}(s,a) = 1$ and $\kappa_{h,i}(s,a) \geq 0$ for all $(s,a,h)$, which immediately implies $|\kappa_h(s,a)|2 \leq 1$. Together with the fact that the parameter vectors of $\mathcal{F}^{lin}$ and $\mathcal{G}^{lin}$ are restricted to a bounded ball, this ensures that every function in the union class $\mathcal{H} = \mathcal{F}^{lin} \cup \mathcal{G}^{lin}$ behaves as a linear predictor in an ambient space of dimension $d^{lin} = dH$, yielding a covering-number bound of the form $\log N_\mathcal{H}(\varepsilon_0) \leq c_o d_{\lin} H \log(c_o/\varepsilon_0)$ for some absolute constant $c_o$.

**Theorem 2** (Regret of RFL-$\phi$ in linear RMDP-$\phi$). *For any $\delta \in (0,1]$ and all linear classes, $\mathcal{F}^{lin}, \mathcal{G}^{lin}$ as in eq. 43–44, we set $\beta = \mathcal{O}\big(B_\phi(\sigma) d_{\lin} H \log(KH/\delta))\big)$ in RFL-$\phi$. Then, under Assumption 1–6 and setting $\varepsilon^{\mathrm{dual}} = 0$, with probability at least $1 - \delta$, it holds that*

$$\mathrm{Regret}(K) \leq \widetilde{\mathcal{O}}\Big(\sqrt{H^2 d_{\lin}^2 B_\phi^2(\sigma) K}\Big).$$

*In particular, plugging in the divergence-dependent envelope constants yields:*

   **(TV)**     $\mathrm{Regret}(K) \leq \widetilde{\mathcal{O}}\Big(\sqrt{H^4 d_{\lin}^2 (\min\{H, \sigma^{-1}\})^2 K}\Big),$     *where* $B_{\mathrm{TV}}(\sigma) = \mathcal{O}\big(H \min\{H, \sigma^{-1}\}\big),$    (54)

   **($\chi^2$)**     $\mathrm{Regret}(K) \leq \widetilde{\mathcal{O}}\Big(\sqrt{H^4 d_{\lin}^2 (1 + \sqrt{\sigma})^2 K}\Big),$     *where* $B_{\chi^2}(\sigma) = \mathcal{O}\big(H(1 + \sqrt{\sigma})\big),$    (55)

   **(KL)**     $\mathrm{Regret}(K) \leq \widetilde{\mathcal{O}}\Big(\sqrt{H^4 d_{\lin}^2 \sigma^2 K}\Big),$     *where* $B_{\mathrm{KL}}(\sigma) = \mathcal{O}\big(H\sigma\big).$    (56)

*Proof.* We adapt the proof of Theorem 1 to the linear $\phi$–RMDP setting under Assumption 5, with linear function classes $\mathcal{F}^{lin}, \mathcal{G}^{lin}$ as defined in eq. 43–44. In this work, the exploration is controlled intrinsically by the robust BE dimension (Definition 4).

**Step 1: Starting point from the general regret proof (I-II decomposition).** By the definition of robust regret eq. 5 and the same decomposition steps as in the proof eq. 27–29 of Theorem 1, we have

$$\mathrm{Regret}(K) \leq I + II, \tag{57}$$

where: (i) $I$ is the *exploration / Bellman-residual term* controlled by the robust BE dimension (eq. 30); (ii) $II$ is the *robust-operator approximation term* due to learning the robust Bellman operator via the dual ERM plug-in (eq. 31).

By eq. 34 and eq. 36, we can bound from I and I$I$, respectively, as

$$I \leq \mathcal{O}\Big(H\sqrt{\dim_{\mathrm{BE}}^{\mathrm{rob}}(\mathcal{F}, \mathcal{D}_\mathcal{F}, 1/\sqrt{K}) B_\phi(\sigma) K \log(|\mathcal{F}^{lin}||\mathcal{G}^{lin}|KH/\delta)}\Big), \tag{58}$$

$$II \leq \mathcal{O}\Big(\sqrt{HB_\phi^2(\sigma) \log(|\mathcal{F}^{lin}||\mathcal{G}^{lin}|KH/\delta)K} + \varepsilon^{\mathrm{dual}}\Big), \tag{59}$$

where $B_\phi(\sigma)$ is the uniform envelope constant from Assumption 3, and $\varepsilon^{\mathrm{dual}}$ is the dual realizability bias (Assumption 2).

**Step 2: Linear $\phi$–RMDP consequences (structural assumptions and complexity terms).** For better clarity, we work under the exact dual realizability condition, and we set $\varepsilon^{\mathrm{dual}} = 0$ for simplicity of proof [3]. Under Assumption 5, the linear

---

[3]By Lemma 3(iii), when we instantiate RFL-$\phi$ with the linear dual class $\mathcal{G}^{lin}$, the dual minimizer of the TV robust Bellman operator is exactly realizable, so the dual approximation error in Assumption 2 vanishes and we have $\varepsilon^{\mathrm{dual}} = 0$. For clarity, we therefore focus on this exact-realizability case in the sequel. If one instead works with a dual class that only approximately realizes the optimal dual (so $\varepsilon^{\mathrm{dual}} > 0$), the same proof strategy goes through with an additional additive term of order $\varepsilon^{\mathrm{dual}}$ propagating from the bound on $II$ (cf. 36) into the final regret bound; no other part of the argument needs to be modified, and the dependence on $(K, d, H, \sigma)$ remains unchanged.

classes $\mathcal{F}^{lin}, \mathcal{G}^{lin}$ together with Lemma 3 guarantee that all structural assumptions used in Theorem 1 remain valid when we instantiate the analysis with the linear RMDP-$\phi$; the only resulting changes are as follows:

- The complexity term $\log(|\mathcal{F}^{lin}||\mathcal{G}^{lin}|)$ is replaced by a covering-number bound for the union class $\mathcal{H} \triangleq \mathcal{F}^{lin} \cup \mathcal{G}^{lin}$. By Assumption 6, for $\varepsilon_0 = 1/KH$, the union class $\mathcal{H} = \mathcal{F}^{lin} \cup \mathcal{G}^{lin}$ admits an $\varepsilon_0$-cover in $\|\cdot\|_\infty$ with $\log N_{\mathcal{H}}(\varepsilon_0) \leq c_0\, d_{\lin} H\, \log(c_0 K)$, for some absolute constant $c_0 > 0$ (Shalev-Shwartz & Ben-David, 2014). Therefore, using thsi fact we have $\log\big(|\mathcal{F}^{lin}||\mathcal{G}^{lin}|KH/\delta\big) \leq c_0\, d_{\lin} H\, \log(c_0 K) + \log(c_1 KH/\delta) = \mathcal{O}\Big(d_{\lin} H \log(KH/\delta)\Big)$

- The dual bias term $\varepsilon^{\mathrm{dual}}$ drops out.

**Step 3: Bounding** II **(robust operator approximation) in the linear case.**    The derivation of the general bound eq. 59 for II (Lemma K.4) uses ERM generalization bound Lemma T.1 and a union bound over all episodes, time steps, and function pairs $(f, g) \in \mathcal{F}^{lin} \times \mathcal{G}^{lin}$. In the linear case, we instead apply the same argument to a finite $\varepsilon_0$-net of $\mathcal{H}$.

More precisely, fix $\varepsilon_0 = 1/KH$ and let $\mathcal{H}_0 \subset \mathcal{H}$ be a minimal $\varepsilon_0$-net under $\|\cdot\|_\infty$, such that $|\mathcal{H}_0| = N_{\mathcal{H}}(\varepsilon_0)$. We then repeat the concentration analysis of Lemma T.1, but take the union bound over the finite set $(k, h, \varphi) \in [K] \times [H] \times \mathcal{H}_0$ instead of $(k, h, f, g) \in [K] \times [H] \times \mathcal{F} \times \mathcal{G}$. The approximation error between any $f \in \mathcal{H}$ and its nearest neighbor $f' \in \mathcal{H}_0$ is at most $\varepsilon_0$ in $\|\cdot\|_\infty$ and hence contributes only an $o(1)$ term in $K$ to the final regret bound, which we absorb into the big-$\mathcal{O}$ notation. Therefore, following the same steps of the proof of Lemma K.4 and setting $\varepsilon^{\mathrm{dual}} = 0$, we conclude that in the linear case eq. 59 becomes

$$II \leq \mathcal{O}\Big(\sqrt{HB_\phi^2(\sigma)d_{\lin}K \log\big(KH/\delta\big)}\Big). \tag{60}$$

**Step 4: Bounding** I **via robust BE dimension (linear case).**    By Definition 4, the exploration term is controlled by the DE dimension of the robust residual class $(I - \mathcal{T}_h^{\phi,\sigma})\mathcal{F}$ under the on-policy family $\mathcal{D}_{\mathcal{F}}$. The proof of Theorem 1 shows that, on the event that all confidence sets are valid, eq. 58 holds with $\dim_{\mathrm{BE}}^{\mathrm{rob}}(\mathcal{F}, \mathcal{D}_{\mathcal{F}}, 1/\sqrt{K})$. In the linear case, the robust BE dimension is finite and satisfies

$$\dim_{\mathrm{BE}}^{\mathrm{rob}}(\mathcal{F}^{lin}, \mathcal{D}_{\mathcal{F}^{lin}}, 1/\sqrt{K}) = \widetilde{O}(d_{\lin}), \tag{61}$$

by the same linear-DE/BE arguments as in the non-robust setting (Jin et al., 2021; Wang et al., 2020b) and as summarized in Remark 6. Hence,

$$I \leq \mathcal{O}\Big(H\sqrt{d_{\lin}^2 B_\phi(\sigma)K \log\big(KH/\delta\big)}\Big). \tag{62}$$

**Step 5: Combine** I **and** II**.**    Combining eq. 57, eq. 60, and eq. 62, we obtain

$$\begin{aligned}
\mathrm{Regret}(K) &\leq \mathcal{O}\Big(H\sqrt{d_{\lin}^2 B_\phi(\sigma)K \log\big(KH/\delta\big)}\Big) + \mathcal{O}\Big(\sqrt{HB_\phi^2(\sigma)d_{\lin}K \log\big(KH/\delta\big)}\Big) \\
&\leq \widetilde{\mathcal{O}}\Big(\sqrt{H^2 d_{\lin}^2 B_\phi^2(\sigma)K}\Big).
\end{aligned} \tag{63}$$

**Step 6: Plugging in divergence-specific envelopes.**    We now obtain the regret bound for each divergences as follows:

- **TV-Divergence Case:** According to eq. 37, we have $B_{\mathrm{TV}} := \mathcal{O}\big(H \min\{H, 1/\sigma\}\big)$. Putting this in eq. 63, we get

$$\mathrm{Regret}(K) \leq \widetilde{\mathcal{O}}\Big(\sqrt{H^4 d_{\lin}^2 (\min\{H, 1/\sigma\})^2 K}\Big). \tag{64}$$

- $\chi^2$**-Divergence Case:** According to eq. 39, we have $B_{\chi^2} := \mathcal{O}\big(H(1 + \sqrt{\sigma})\big)$. Putting this in eq. 63, we get

$$\mathrm{Regret}(K) \leq \widetilde{\mathcal{O}}\Big(\sqrt{H^4 d_{\lin}^2 (1 + \sqrt{\sigma})^2 K}\Big). \tag{65}$$

- **KL-Divergence Case:** According to eq. 41, we have $B_{\mathrm{KL}} := \mathcal{O}\big(H\sigma\big)$. Putting this in eq. 63, we get

$$\mathrm{Regret}(K) \leq \widetilde{\mathcal{O}}\Big(\sqrt{H^4 d_{\lin}^2 \sigma^2 K}\Big). \tag{66}$$

This concludes the proof. $\qquad\square$

## F.1. Key Lemmas

**Lemma K.1** (Robust value function error decomposition)**.** *Consider an RMDP with a $\Xi$-divergence uncertainty set. For any $f = \{f_h\}_{h=1}^H \in \mathcal{F}$, let $\pi^f$ be the greedy policy induced by $f$, and we define $V^f := \mathbb{E}\Big[f_1(s_1, \pi_1^f(s_1))\Big]$, and $V^{\pi,Q} := \mathbb{E}_{a_{1:H} \sim \pi, \, s_{h+1} \sim Q_h}\Big[\sum_{h=1}^H r_h(s_h, a_h)\Big]$. Let $\psi_{h+1}^f(s') := \max_{a' \in \mathcal{A}} f_{h+1}(s', a')$. For any policy $\pi$ and stage $h$, define the robust average Bellman error*

$$\varepsilon_h^{\phi,\sigma}(f, \pi) := \mathbb{E}\Big[f_h(s_h, a_h) - (\mathcal{T}_h^{\phi,\sigma} f_{h+1})(s_h, a_h) \,\Big|\, a_h \sim \pi_h(\cdot|s_h), \; s_{h+1} \sim P_h^\star(\cdot|s_h, a_h)\Big], \tag{67}$$

*where the expectation is taken over the trajectory distribution induced by executing $\pi$ in the nominal environment $P^\star$. Then, under Assumption 1 and the optimism property of $\{f^{(k)}\}_{k=1}^K$ ensured by the confidence sets, the cumulative robust regret in eq. 5 satisfies*

$$\text{Regret}(K) \leq \sum_{k=1}^K \sum_{h=1}^H \varepsilon_h^{\phi,\sigma}(f^{(k)}, \pi^{(k)}), \tag{68}$$

*where $\pi^{(k)} := \pi^{f^{(k)}}$.*

*Proof.* Fix any kernel $Q \in \mathcal{P}$ and any $f \in \mathcal{F}$. Recall by eq. 11 we have

$$(\mathcal{T}_h^{\phi,\sigma} f_{h+1})(s, a) = r_h(s, a) + \inf_{P \in \mathcal{U}_h^{\phi,\sigma}(s,a)} \mathbb{E}_{s' \sim P(\cdot|s,a)}\Big[V_{h+1}^f(s')\Big].$$

Therefore, for every $(s, a)$,

$$(\mathcal{T}_h^{\phi,\sigma} f_{h+1})(s, a) \leq r_h(s, a) + \mathbb{E}_{s' \sim Q_h(\cdot|s,a)}\Big[V_{h+1}^f(s')\Big], \tag{69}$$

which implies

$$f_h(s, a) - (\mathcal{T}_h^{\phi,\sigma} f_{h+1})(s, a) \geq f_h(s, a) - r_h(s, a) - \mathbb{E}_{s' \sim Q_h(\cdot|s,a)}\Big[V_{h+1}^f(s')\Big]. \tag{70}$$

Now take expectation along a trajectory generated by executing $\pi^f$ in the environment with transition kernel $Q$:

$$\sum_{h=1}^H \mathbb{E}^{\pi^f, Q}\Big[f_h(s_h, a_h) - (\mathcal{T}_h^{\phi,\sigma} f_{h+1})(s_h, a_h)\Big] \geq \sum_{h=1}^H \mathbb{E}^{\pi^f, Q}\Big[f_h(s_h, a_h) - r_h(s_h, a_h) - \mathbb{E}_{Q_h}[V_{h+1}^f]\Big], \tag{71}$$

where $\mathbb{E}^{\pi^f, Q}$ denotes expectation over trajectories with $a_h \sim \pi_h^f(\cdot|s_h)$ and $s_{h+1} \sim Q_h(\cdot|s_h, a_h)$.

The right-hand side of eq. 71 admits a standard telescoping argument (cf. (Jiang et al., 2017)[Lemma 1]):

$$\sum_{h=1}^H \mathbb{E}^{\pi^f, Q}\Big[f_h(s_h, a_h) - r_h(s_h, a_h) - \mathbb{E}_{Q_h}[V_{h+1}^f]\Big] = V^f - V^{\pi^f, Q}. \tag{72}$$

Combining eq. 71 and eq. 72 yields

$$\sum_{h=1}^H \mathbb{E}^{\pi^f, Q}\Big[f_h(s_h, a_h) - (\mathcal{T}_h^{\phi,\sigma} f_{h+1})(s_h, a_h)\Big] \geq V^f - V^{\pi^f, Q}. \tag{73}$$

Now specialize to the *worst-case* kernel $Q = P^\omega(\pi^f)$ for policy $\pi^f$, i.e., for each $(s, a, h)$,

$$\mathbb{E}_{s' \sim P_h^\omega(\cdot|s,a)}\Big[V_{h+1}^f(s')\Big] = \inf_{P \in \mathcal{U}_h^{\phi,\sigma}(s,a)} \mathbb{E}_{s' \sim P(\cdot|s,a)}\Big[V_{h+1}^f(s')\Big].$$

In this case, eq. 69 holds with equality pointwise, and hence eq. 73 tightens to

$$V^f - V^{\pi^f, P^\omega} = \sum_{h=1}^H \mathbb{E}^{\pi^f, P^\omega}\left[ f_h(s_h, a_h) - (\mathcal{T}_h^{\phi, \sigma} f_{h+1})(s_h, a_h) \right]. \tag{74}$$

Finally, apply eq. 74 episode-by-episode with $f = f^{(k)}$ and $\pi^{(k)} = \pi^{f^{(k)}}$. By optimism of $f^{(k)}$ (guaranteed by the validity of the confidence sets under Assumption 1), we have $V_1^{\star, \sigma}(s_1^k) \leq V_1^{f^{(k)}}(s_1^k)$. Moreover, by definition of the robust value, $V_1^{\pi^{(k)}, \sigma}(s_1^k)$ is the value of $\pi^{(k)}$ under the robust Bellman recursion induced by $\mathcal{T}^\sigma$. Therefore,

$$\begin{aligned}
\text{Regret}(K) &= \sum_{k=1}^K \left( V_1^{\star, \sigma}(s_1^k) - V_1^{\pi^{(k)}, \sigma}(s_1^k) \right) \\
&\leq \sum_{k=1}^K \left( V_1^{f^{(k)}}(s_1^k) - V_1^{\pi^{(k)}, \sigma}(s_1^k) \right).
\end{aligned}$$

It remains to relate $V_1^{f^{(k)}}(s_1^k) - V_1^{\pi^{(k)}, \sigma}(s_1^k)$ to the robust Bellman residuals of $f^{(k)}$. Using the identity eq. 74 (which holds for the robust Bellman operator $\mathcal{T}^{\phi, \sigma}$) and a standard telescoping argument along the trajectory generated by executing $\pi^{(k)}$, we obtain

$$V_1^{f^{(k)}}(s_1^k) - V_1^{\pi^{(k)}, \sigma}(s_1^k) = \sum_{h=1}^H \mathbb{E}\left[ f_h^{(k)}(s_h^k, a_h^k) - (\mathcal{T}_h^{\phi, \sigma} f_{h+1}^{(k)})(s_h^k, a_h^k) \;\Big|\; a_h^k \sim \pi_h^{(k)}(\cdot | s_h^k) \right],$$

and hence,

$$\begin{aligned}
\text{Regret}(K) &\leq \sum_{k=1}^K \sum_{h=1}^H \mathbb{E}\left[ f_h^{(k)}(s_h^k, a_h^k) - (\mathcal{T}_h^{\phi, \sigma} f_{h+1}^{(k)})(s_h^k, a_h^k) \;\Big|\; a_h^k \sim \pi_h^{(k)}(\cdot | s_h^k) \right] \\
&= \sum_{k=1}^K \sum_{h=1}^H \varepsilon_h^{\phi, \sigma}(f^{(k)}, \pi^{(k)}),
\end{aligned}$$

where $\varepsilon_h^{\phi, \sigma}(f, \pi)$ is defined in eq. 67. This completes the proof. $\qquad\square$

**Lemma K.2.** *Suppose Assumption 1 holds. Then if $\beta > 0$ is selected as in Theorem 1, then with probability at least $1 - \delta$, for all $k \in [K]$, RFL-$\phi$ satisfies*

*(a)* $Q^{\star, \sigma} \in \mathcal{F}^{(k)}$.

*(b)* $\sum_{t=1}^{k-1} \mathbb{E}_{(s,a) \sim \pi^t}\left[ \left( f_h^{(k)}(s, a) - \left[ \mathcal{T}_{h, g_{f_{h+1}^{(k)}}}^{\phi, \sigma} f_{h+1}^{(k)} \right](s, a) \right)^2 \right] \leq \mathcal{O}(\beta)$.

*Proof.* The proof follows the same structure as the non-robust argument (Jin et al., 2021)[Lemma 39 and 40] and (Xie et al., 2022)[Lemma 15] (martingale concentration via Freedman's inequality plus a finite cover of the functional class), with two robust-specific ingredients: (i) the dual scalar representation of the TV worst-case expectation and (ii) the use of the dual pointwise integrand as a sample target. We derive the complete proof as follows.

☞ *Proof of ineq. (b)* To show ineq. (b), we will focus on the proof-lines of (Jin et al., 2021)[Lemma 39] and (Xie et al., 2022)[Lemma 15 (2)]. We first fix $(k, h, f)$ tuple, where an episode $k$ we consider a function $f^{(k)} = \{f_1^{(k)}, \ldots, f^{(k)})_H\} \in \mathcal{F}$. Let us denote $\psi_{h+1}^{f^{(k)}}(s) := \psi_{f_{h+1}^{(k)}}^f(s)$ such that $\psi_{h+1}^{f^{(k)}}(s_{h+1}) := f_{h+1}^{(k)}(s_{h+1}, \pi_{h+1}^{(k)}(s_{h+1}))$, and we assume $\|f\|_\infty, \|\psi^f\|_\infty \leq H$ (this is the boundedness assumption used throughout). We consider the filtration induced as

$$\mathcal{H}_h^{(k)} = \{s_1^i, a_1^i, r_1^i, \ldots, s_H^i\}_{i=1}^{k-1} \bigcup \{s_1^k, a_1^k, r_1^k, \ldots, s_h^k, a_h^k\}$$

as the filtration containing the history up to the episode $k$ at step $h$ and $\mathcal{H}_h^{(k)}$ is sampled by following $\pi^{(k)}$ in the $k^{th}$ episode.

We obtain $\underline{g}_{f^{(k)}} := (\underline{g}_{\eta, f^{(k)}}, \underline{g}_{\nu, f^{(k)}})$ such that $\underline{g}_{\eta, f^{(k)}}, \underline{g}_{\nu, f^{(k)}} \in [0, 2H/\sigma]$ as a measurable minimizer of eq. 21 that satisfies Assumption 2. For the trajectory of episode $k$, we define

$$
l_\phi\left(\psi_{h+1}^{f^{(k)}}; s_h^k, a_h^k, \pi_{h+1}^{(k)}; \underline{g}_{f_{h+1}^{(k)}}\right) := \underline{g}_{\eta, f_{h+1}^{(k)}}(s_h^k, a_h^k)\sigma - \underline{g}_{\nu, f_{h+1}^{(k)}}(s_h^k, a_h^k)
$$
$$
+ \underline{g}_{\eta, f_{h+1}^{(k)}}(s_h^k, a_h^k)\, \phi^\star\left(-\frac{\psi_{h+1}^{f^{(k)}}(s_h^k, a_h^k) + \underline{g}_{\nu, f_{h+1}^{(k)}}(s_h^k, a_h^k)}{\underline{g}_{\eta, f_{h+1}^{(k)}}(s_h^k, a_h^k)}\right), \qquad (75)
$$

such that $\left| l_\phi\left(\psi_{h+1}^{f^{(k)}}; s_h^k, a_h^k, \pi_{h+1}^{(k)}; \underline{g}_{f_{h+1}^{(k)}}\right)\right| \le B_\phi(\sigma)$, where $C_1 > 0$ is an absolute constant, and

$$
\mathbb{E}\left[l_\phi\left(\psi_{h+1}^{f^{(k)}}; s_h^k, a_h^k, \pi_{h+1}^{(k)}; \underline{g}_{f_{h+1}^{(k)}}\right)\Big| \mathcal{H}_h^{(k)}\right] = \left[\mathcal{T}_{h, \underline{g}_{f_{h+1}^{(k)}}}^{\phi, \sigma} f_{h+1}^{(k)}\right](s_h^k, a_h^k) - r_h^{(k)}(s_h^k, a_h^k). \qquad (76)
$$

For each episode $k$ and step $h$, we define the martingale difference as

$$
X_h^{(k)}(f, \underline{g}_f) := \left(f_h^{(k)}(s_h^k, a_h^k) - r_h^{(k)}(s_h^k, a_h^k) - l_\phi\left(\psi_{h+1}^{f^{(k)}}; s_h^k, a_h^k, \pi_{h+1}^{(k)}; \underline{g}_{f_{h+1}^{(k)}}\right)\right)^2
$$
$$
- \left(\left[\mathcal{T}_{h, \underline{g}_{f_{h+1}^{(k)}}}^{\phi, \sigma} f_{h+1}^{(k)}\right](s_h^k, a_h^k) - r_h^{(k)}(s_h^k, a_h^k) + l_\phi\left(\psi_{h+1}^{f^{(k)}}; s_h^k, a_h^k, \pi_{h+1}^{(k)}; \underline{g}_{f_{h+1}^{(k)}}\right)\right)^2, \qquad (77)
$$

such that we have $\left|X_h^{(k)}(f, \underline{g}_f)\right| \le C_2 B_\phi(\sigma)^2$, where $C_2 > 0$ is an absolute constant. Moreover,

$$
\mathbb{E}\left[X_h^{(k)}(f, \underline{g}_f)\Big| \mathcal{H}_h^{(k)}\right] = \left(f_h^{(k)}(s, a) - \left[\mathcal{T}_{h, \underline{g}_{f_{h+1}^{(k)}}}^{\phi, \sigma} f_{h+1}^{(k)}\right](s, a)\right)^2
$$
$$
\mathrm{Var}\left[X_h^{(k)}(f, \underline{g}_f)\Big| \mathcal{H}_h^{(k)}\right] \le C_3 B_\phi(\sigma)^2 \mathbb{E}\left[X_h^{(k)}(f, \underline{g}_f)\Big| \mathcal{H}_h^{(k)}\right], \qquad (78)
$$

where $C_2, C_3 > 0$ are absolute constants.

Therefore, by Freedman's inequality as given Lemma T.3, we can write

$$
\left|\sum_{k=1}^K \left(X_h^{(k)}(f, \underline{g}_f) - \mathbb{E}\left[X_h^{(k)}(f, \underline{g}_f)\right]\right)|\mathcal{H}_h^{(k)}\right| \le \mathcal{O}\left(\sqrt{\log(1/\delta)\sum_{k=1}^K \mathbb{E}\left[X_h^{(k)}(f, \underline{g}_f)\Big|\mathcal{H}_h^{(k)}\right]} + \log(1/\delta)\right). \qquad (79)
$$

Now, let us consider $\mathcal{X}_\rho$ be the $\rho$-cover of $\mathcal{F}\bigcup\mathcal{G}$. Now taking a union bound for all $(k, h, \Xi) \in [K] \times [H] \times \mathcal{X}_\rho$, and following the same proof-lines as in (Jin et al., 2021)[Lemma 39], we get

$$
\sum_{t<k}\mathbb{E}\left[\left(f_h^{(k)}(s, a) - \left[\mathcal{T}_{h, \underline{g}_{f_{h+1}^{(k)}}}^{\phi, \sigma} f_{h+1}^{(k)}\right](s, a)\right)^2\Bigg|\mathcal{H}_h^{(t)}\right] \le \mathcal{O}(\beta), \qquad (80)
$$

where $\beta = \mathcal{O}\left(B_\phi(\sigma)\log(\mathcal{N}_{F\cup G}(\rho) \cdot KH/\delta))\right)$. Now, we set $\rho = 1/K$. In addition, as $\mathcal{F}\cup\mathcal{G}$ is finite, then $\log\mathcal{N}_{F\cup G}(1/K) \le \log|\mathcal{F}\cup\mathcal{G}|$, therefore, we consider $\beta$ as $\beta = \mathcal{O}\left(B_\phi(\sigma)\log(|\mathcal{F}||\mathcal{G}| \cdot KH/\delta))\right)$.

Therefore, eq. 80 concludes that $\sum_{t<k}\mathbb{E}_{(s,a)\sim\pi^t}\left[\left(f_h^{(k)}(s, a) - \left[\mathcal{T}_{h, \underline{g}_{f_{h+1}^{(k)}}}^{\phi, \sigma} f_{h+1}^{(k)}\right](s, a)\right)^2\right] \le \mathcal{O}(\beta)$.

☞ *Proof of ineq. (a)* To show ineq. (a), we will focus on the proof-lines of (Jin et al., 2021)[Lemma 40] and (Xie et al., 2022)[Lemma 15 (1)]. Fix $(k, h, f)$ and follow the same notation as mentioned in the proof lines of the inequality (b), we define

$$
W_h^{(t)}(f, \underline{g}_f) := \left( f_h^{(t)}(s_h^t, a_h^t) - r_h^{(t)}(s_h^t, a_h^t) - l_\phi\left( \psi_{h+1}^{f^{(k)}}; s_h^k, a_h^k, \pi_{h+1}^{(k)}; \underline{g}_{f_{h+1}^{(k)}} \right) \right)^2
$$
$$
- \left( Q_h^{\star,\sigma}(s_h^t, a_h^t) - r_h^{(t)}(s_h^t, a_h^t) + l_\phi\left( \psi_{h+1}^{f^{(k)}}; s_h^k, a_h^k, \pi_{h+1}^{(k)}; \underline{g}_{f_{h+1}^{(k)}} \right) \right)^2, \quad \text{for } 1 \le t \le k.
$$

As in eq. 78, $\mathbb{E}\left[ W_h^{(t)}(f, \underline{g}_f) \mid \mathcal{H}_h^{(t)} \right] = \left( f_h^{(t)}(s_h^t, a_h^t) - Q_h^{\star,\sigma}(s_h^t, a_h^t) \right)^2$ where $\mathcal{H}_h^{(t)}$ be the filtration induced by $\{s_1^i, a_1^i, r_1^i, \ldots, s_H^i\}_{i=1}^{t-1} \bigcup \{s_1^t, a_1^t, r_1^t, \ldots, s_h^t, a_h^t\}$. Similarly, we can verify that $|W_h^{(t)}(f, \underline{g}_f)| \le C_4 \left( B_\phi(\sigma)(\sigma) \right)^2$ and $\mathrm{Var}\left[ W_h^{(t)}(f, \underline{g}_f) \mid \mathcal{H}_h^{(t)} \right] \le C_5 \left( B_\phi(\sigma)(\sigma) \right)^2 E\left[ W_h^{(t)}(f, \underline{g}_f) \mid \mathcal{H}_h^{(t)} \right]$. Now, following the proof-lines of (Jin et al., 2021)[Lemma 40], and applying Freedman's ineq. (Lemma T.3 and a cover of $\mathcal{G}$ yields, w.p. $1 - \delta$, we get

$$
\sum_{t=1}^{k-1} \left[ Q_h^{\star,\sigma}(s_h^t, a_h^t) - r_h^t(s_h^t, a_h^t) - Q_{h+1}^{\star,\sigma}(s_{h+1}^t, \pi_{h+1}^{Q^{\star,\sigma}}(s_{h+1}^t)) \right]^2
$$
$$
\le \sum_{t=1}^{k-1} \left[ f_h^{(t)}(s_h^t, a_h^t) - r_h^t(s_h^t, a_h^t) - Q_{h+1}^{\star,\sigma}(s_{h+1}^t, \pi_{h+1}^{Q^{\star,\sigma}}(s_{h+1}^t)) \right]^2 + \mathcal{O}(\beta).
$$

Finally, by recalling the definition of $\mathcal{F}^{(k)}$, we conclude that with probability at least $1 - \delta$, $Q^{\star,\sigma} \in \mathcal{F}^{(k)}$ for all $k \in [K]$.

This concludes the proof of Lemma K.2. □

**Lemma K.3** (Robust Bellman–Eluder Dimension Bound (Jin et al., 2021)). *Fix $h \in [H]$ and let $\Xi_h^\xi = (\mathcal{I} - \mathcal{T}_h^{\phi,\sigma})\mathcal{F}$. Let $\Pi$ be the family of distributions under $P^\star$. Assume $\sup_{\xi \in \Xi_h^\xi} \|\xi\|_\infty \le C_\phi$. Let $\{(f^{(k)}, \pi^{(k)})\}_{k=1}^K$ be the sequence produced by the algorithm and define*
$$
\xi_k := f_h^{(k)} - (\mathcal{T}_h^{\phi,\sigma} f_{h+1}^{(k)}) \in \Xi_h^\xi, \qquad \mu := \{\mu_k\}_{k=1}^K \in \Pi.
$$
*Suppose there exists $\beta > 0$ such that for all $k \in [K]$,*
$$
\sum_{t=1}^{k-1} \mathbb{E}_{\mu_t}\left[ \xi_t^2 \right] \le \beta.
$$

*Then for all $k \in [K]$,*
$$
\sum_{t=1}^{k} \left| \mathbb{E}_{\mu_t}[\xi_t] \right| \le \mathcal{O}\left( \sqrt{\dim_{\mathrm{DE}}(\Xi_h^\xi, \Pi_h, 1/k) \beta k} \right).
$$

*Proof.* The proof is the same as in Jin et al. (2021)[Lemma 41]. Fix $h \in [H]$ and let $\Xi = \Xi_h^\xi$ and $\Pi = \Pi_h$. Assume $|\xi(x)| \le C_\phi$ for all $(\xi, x) \in \Xi \times \mathcal{X}$. Suppose a sequence $\{\xi_k\}_{k=1}^K \subseteq \Xi$ and $\{\mu_k\}_{k=1}^K \subseteq \Pi$ satisfies for all $k \in [K]$,
$$
\sum_{t=1}^{k-1} \left( \mathbb{E}_{\mu_t}[\xi_t] \right)^2 \le \beta.
$$

Let $e_t := |\mathbb{E}_{\mu_t}[\xi_t]|$ and sort $(e_1, \ldots, e_K)$ in non-increasing order: $e_{(1)} \ge e_{(2)} \ge \cdots \ge e_{(K)}$. Then
$$
\sum_{t=1}^{K} e_t = \sum_{t=1}^{K} e_{(t)} \le \sum_{t=1}^{K} e_{(t)} \mathbf{1}\{e_{(t)} \le \omega\} + \sum_{t=1}^{K} e_{(t)} \mathbf{1}\{e_{(t)} > \omega\} \le K\omega + \sum_{t=1}^{K} e_{(t)} \mathbf{1}\{e_{(t)} > \omega\}.
$$

Let $d := \dim_{\mathrm{DE}}(\Xi, \Pi, \omega)$. Fix any index $t$ with $e_{(t)} > \omega$. Then there exists $\alpha$ such that $e_{(t)} > \alpha \geq \omega$. By Proposition T.4 applied at threshold $\alpha$,

$$t \leq \sum_{i=1}^{K} \mathbf{1}\{e_i > \alpha\} \leq \left(\frac{\beta}{\alpha^2} + 1\right) \dim_{\mathrm{DE}}(\Xi, \Pi, \alpha) \leq \left(\frac{\beta}{\alpha^2} + 1\right)d,$$

where the last inequality uses that $\dim_{\mathrm{DE}}(\Xi, \Pi, \alpha)$ is non-increasing in $\alpha$ and $\alpha \geq \omega$. Rearranging gives $\alpha^2 \leq d\beta/(t-d)$ (for $t > d$), hence

$$e_{(t)} \leq \min\left\{C_\phi, \sqrt{\frac{d\beta}{t-d}}\right\}.$$

Therefore,

$$\sum_{t=1}^{K} e_{(t)}\mathbf{1}\{e_{(t)} > \omega\} \leq \min\{d,k\}C_\phi + \sum_{t=d+1}^{K} \sqrt{\frac{d\beta}{t-d}} \leq \min\{d,k\}C_\phi + \sqrt{d\beta}\int_0^K \frac{1}{\sqrt{t}}\,dt \leq \min\{d,k\}C_\phi + 2\sqrt{d\beta k}.$$

Combining with the earlier decomposition yields

$$\sum_{t=1}^{K} |\mathbb{E}_{\mu_t}[\xi_t]| \leq K\omega + \min\{d,K\}C_\phi + 2\sqrt{d\beta K}, \tag{81}$$

which is the claimed bound up to absolute constants.

Let $\xi_k$ and $\mu_k$ be defined as:
$$\xi_k := f_h^{(k)} - (\mathcal{T}_h^{\phi,\sigma} f_{h+1}^{(k)}) \in \Xi_h^{\xi_k}, \qquad \mu_k \in \Pi.$$

We set $\Xi = \Xi_h^\xi$, and $\Pi = \Pi_h$. By Jensen's inequality, for each $k$ and each $t < k$,

$$\left(\mathbb{E}_{\mu_t}[\xi_k]\right)^2 \leq \mathbb{E}_{\mu_t}[\xi_k^2].$$

Hence, the assumed condition $\sum_{t<k} \mathbb{E}_{\mu_t}[\xi_k^2] \leq \beta$ implies $\sum_{t<k}(\mathbb{E}_{\mu_t}[\xi_k])^2 \leq \beta$ for all $k$. Applying eq. 81 with $\omega = 1/k$ then yields

$$\sum_{i=1}^{k} |\mathbb{E}_{\mu_i}[\xi_i]| \leq \mathcal{O}\left(\sqrt{\dim_{\mathrm{DE}}(\Xi, \Pi, 1/k)\,\beta\,k} + \min\{k, \dim_{\mathrm{DE}}(\Xi, \Pi, 1/k)\}C_\phi + 1\right).$$

Dropping lower-order terms gives the claimed bound. The key inequality eq. 81 itself follows from Proposition T.4, whose proof is a direct adaptation of Jin et al. (2021)[Proposition 43]. This completes the proof.

$\square$

**Lemma K.4** ($\phi$-Dual optimization error bound (Lemma 1))**.** *Fix $h \in [H]$ and a policy $\pi$. Let $\mu_h^\pi$ denote the step-$h$ state–action distribution induced by $\pi$ under the nominal kernel $P_h^\star$, and let $\mathcal{D}_h$ be a dataset of transitions $(s,a,s')$ collected by executing $\pi$ at step $h$. For any $f_{h+1} \in \mathcal{F}_{h+1}$, let $\underline{g}_{f_{h+1}}$ denote the dual parameter obtained from the empirical optimization in eq. 21 for a given state–action value function $f$ as given in eq. 15, and let $\mathcal{T}_g^{\phi,\sigma}$ be as defined in eq. 13. Then for any $\delta \in (0,1)$, with probability at least $1 - \delta$,*

$$\sup_{f_{h+1} \in \mathcal{F}_{h+1}} \left\|\mathcal{T}_h^{\phi,\sigma} f_{h+1} - \mathcal{T}_{h,\underline{g}_{f_{h+1}}}^{\phi,\sigma} f_{h+1}\right\|_{1,\mu_h^\pi} = \mathcal{O}\left(B_\phi(\sigma)\sqrt{\frac{\log(|\mathcal{F}_{h+1}||\mathcal{G}|/\delta)}{|\mathcal{D}_h|}} + \varepsilon^{\mathrm{dual}}.\right). \tag{82}$$

*Proof.* For a fixed $h \in [H]$, fix an arbitrary $f \in \mathcal{F}$ and recall that $\underline{g}_f$ as defined in eq. 21, where $\widehat{\mathrm{DualLoss}}$ is given in eq. 21. For notational convenience, define the dual objective

$$\zeta_f(g) := \mathbb{E}_{(s,a)\sim\mu_h^\pi,\,s'\sim P_h^\star(\cdot|s,a)}\left[l_\phi(f;s,a,s';g)\right].$$

where $l_\phi(f;s,a,s';g) := g_\eta(s,a)\sigma - g_\nu(s,a) + g_\eta(s,a)\,\phi^\star\left(-\frac{\max_{a'} f(s',a')+g_\nu(s,a)}{g_\eta(s,a)}\right)$ by eq. 10.

Using the dual representation in eq. 20, the difference between the true robust Bellman operator and its empirical counterpart can be written as

$$\left\| \mathcal{T}^{\phi,\sigma} f - \mathcal{T}^{\phi,\sigma}_{\underline{g}_f} f \right\|_{1,\mu^\pi} = \zeta_f(\underline{g}_f) - \mathbb{E}_{(s,a)\sim\mu^\pi}\left[ \inf_{\eta\geq 0,\nu\in\mathbb{R}} l_\phi\big(f; s, a, s'; \eta, \nu\big) \right]. \tag{83}$$

Next, we use the functional reformulation, which (by the interchange rule for integral functionals (Rockafellar & Wets, 1998)[Theorem 14.60]) (as given in Lemma T.2) states that

$$\mathbb{E}_{(s,a)\sim\mu^\pi}\left[ \inf_{\eta\geq 0,\nu\in\mathbb{R}} l_\phi\big(f; s, a, s'; \eta, \nu\big) \right] = \inf_{g\in\mathcal{L}^1(\mu^\pi;\mathbb{R}^2)} \zeta_f(g).$$

Substituting this into eq. 83 gives

$$\begin{aligned}
\left\| \mathcal{T}^{\phi,\sigma} f - \mathcal{T}^{\phi,\sigma}_{\underline{g}_f} f \right\|_{1,\mu^\pi} &= \zeta_f(\underline{g}_f) - \inf_{g\in\mathcal{L}^1(\mu^\pi;\mathbb{R}^2)} \zeta_f(g) \\
&= \big[ \zeta_f(\underline{g}_f) - \inf_{g\in\mathcal{G}} \zeta_f(g) \big] + \big[ \inf_{g\in\mathcal{G}} \zeta_f(g) - \inf_{g\in\mathcal{L}^1(\mu^\pi;\mathbb{R}^2)} \zeta_f(g) \big].
\end{aligned}$$

The second bracket is controlled by the approximate dual realizability assumption (Assumption 2), which gives

$$\inf_{g\in\mathcal{G}} \zeta_f(g) - \inf_{g\in\mathcal{L}^1(\mu^\pi;\mathbb{R}^2)} \zeta_f(g) \leq \varepsilon^{\mathrm{dual}}.$$

Hence,

$$\left\| \mathcal{T}^{\phi,\sigma} f - \mathcal{T}^{\phi,\sigma}_{\underline{g}_f} f \right\|_{1,\mu^\pi} \leq \zeta_f(\underline{g}_f) - \inf_{g\in\mathcal{G}} \zeta_f(g) + \varepsilon^{\mathrm{dual}}. \tag{84}$$

We now bound the optimization error term $\zeta_f(\underline{g}_f) - \inf_{g\in\mathcal{G}} \zeta_f(g)$. Consider the loss function as given in eq. 10 as

$$l_\phi(f; s, a, s'; g) := g_\eta(s,a)\sigma - g_\nu(s,a) + g_\eta(s,a)\, \phi^\star\left( -\frac{\max_{a'} f(s',a') + g_\nu(s,a)}{g_\eta(s,a)} \right),$$

so that $\zeta_f(g) = \mathbb{E}_{(s,a,s')}\big[l_\phi(f; s, a, s'; g)\big]$ and $\widehat{\mathrm{DualLoss}}(g; f)$ in eq. 21 is the empirical average of $\ell_\phi$ over $\mathcal{D}$. Since $f \in \mathcal{F}$ and $g_\eta, g_\nu \in \mathcal{G}$ take values in $[0, H]$ and $\Theta_\phi$, respectively, and by Assumption 3 we have $|l_\phi(f; s, a, s'; g)| \leq B_\phi(\sigma)$.

By applying the empirical risk minimization generalization bound ((Panaganti et al., 2022)[Lemma 3]) together with the Lipschitz-based bound in eq. 87 of Lemma T.1, we obtain that, with probability at least $1 - \delta$,

$$\zeta_f(\underline{g}_f) - \inf_{g\in\mathcal{G}} \zeta_f(g) \leq C_6 B_\phi(\sigma)\sqrt{\frac{2\log|\mathcal{G}|}{|\mathcal{D}|}} + C_7 B_\phi(\sigma)\sqrt{\frac{2\log(8/\delta)}{|\mathcal{D}|}}, \tag{85}$$

where $C_6$ and $C_7$ are absolute constants. Combining eq. 84 and eq. 85, and then taking a union bound over $f \in \mathcal{F}$, we conclude that, with probability at least $1 - \delta$,

$$\sup_{f\in\mathcal{F}} \left\| \mathcal{T}^{\phi,\sigma} f - \mathcal{T}^{\phi,\sigma}_{\underline{g}_f} f \right\|_{1,\mu^\pi} \leq C\, B_\phi(\sigma)\sqrt{\frac{2\log\big(8|\mathcal{G}||\mathcal{F}|/\delta\big)}{|\mathcal{D}|}} + \varepsilon^{\mathrm{dual}},$$

for some absolute constant $C > 0$, which proves the claimed big-$\mathcal{O}$ bound. $\square$

### F.2. Technical Lemmas

We now state a result for the generalization bounds on empirical risk minimization (ERM) problems. This result is adapted from (Shalev-Shwartz & Ben-David, 2014)[Theorem 26.5, Lemma 26.8, Lemma 26.9].

**Lemma T.1** (ERM generalization bound ([Panaganti et al.](), [2022](), Lemma 3)). *Let $P$ be a distribution on $\mathcal{X}$ and let $\mathcal{H}$ be a hypothesis class of real-valued functions on $\mathcal{X}$. Assume the loss* $\mathrm{Loss} : \mathcal{H} \times \mathcal{X} \to \mathbb{R}$ *satisfies*

$$|\mathrm{Loss}(h, x)| \le c_0, \quad \forall\, h \in \mathcal{H},\ x \in \mathcal{X}, \quad \text{for some constant } c_0 > 0.$$

*Given an i.i.d. sample $\mathcal{D} = \{X_i\}_{i=1}^N$ from $P$, define the empirical risk minimizer $\widetilde{h} \in \arg\min_{h \in \mathcal{H}} \frac{1}{N} \sum_{i=1}^N \mathrm{Loss}(h, X_i)$. For any $\delta \in (0, 1)$ and any population risk minimizer $h^\star \in \arg\min_{h \in \mathcal{H}} \mathbb{E}_{X \sim P}[\mathrm{Loss}(h, X)]$, the following holds with probability at least $1 - \delta$:*

$$\mathbb{E}_{X \sim P}[\mathrm{Loss}(\widetilde{h}, X)] - \mathbb{E}_{X \sim P}[\mathrm{Loss}(h^\star, X)] \le 2R(\mathrm{Loss} \circ \mathcal{H} \circ \mathcal{D}) + 5c_0 \sqrt{\frac{2\log(8/\delta)}{N}}, \tag{86}$$

*where $R(loss \circ \mathcal{H} \circ \mathcal{D})$ is the empirical Rademacher complexity of the loss-composed class $loss \circ \mathcal{H}$, defined by*

$$R(\mathrm{Loss} \circ \mathcal{H} \circ \mathcal{D}) = \frac{1}{N} \mathbb{E}_{\{\sigma_i\}_{i=1}^N} \left[ \sup_{g \in \mathrm{Loss} \circ \mathcal{H}} \sum_{i=1}^N \sigma_i g(X_i) \right],$$

*with $\{\sigma_i\}_{i=1}^N$ independent of $\{X_i\}_{i=1}^N$ and i.i.d. according to a Rademacher random variable $\sigma$ (i.e., $\mathbb{P}(\sigma = 1) = \mathbb{P}(\sigma = -1) = 0.5$). Moreover, if $\mathcal{H}$ is finite, $|\mathcal{H}| < \infty$, and there exist constants $c_1, c_2 > 0$ such that*

$$|h(x)| \le c_0 \quad \forall\, h \in \mathcal{H},\ x \in \mathcal{X}, \qquad \text{and} \qquad \mathrm{Loss}(h, x) \text{ is } c_1\text{-Lipschitz in } h,$$

*then with probability at least $1 - \delta$ we further have*

$$\mathbb{E}_{X \sim P}[\mathrm{Loss}(\widetilde{h}, X)] - \mathbb{E}_{X \sim P}[\mathrm{Loss}(h^\star, X)] \le 2c_1 c_2 \sqrt{\frac{2\log(|\mathcal{H}|)}{N}} + 5c_0 \sqrt{\frac{2\log(8/\delta)}{N}}. \tag{87}$$

We now mention two important concepts from variational analysis ([Rockafellar & Wets](), [1998]()) literature that is useful to relate minimization of integrals and the integrals of pointwise minimization under special class of functions.

**Definition 6** (Decomposable spaces and Normal integrands ([Rockafellar & Wets](), [1998]())(Definition 14.59, Example 14.29)). A space $\mathcal{X}$ of measurable functions is a decomposable space relative to an underlying measure space $(\Omega, \mathcal{A}, \mu)$, if for every function $x_0 \in \mathcal{X}$, every set $A \in \mathcal{A}$ with $\mu(A) < \infty$, and any bounded measurable function $x_1 : A \to \mathbb{R}$, the function

$$x(\omega) = x_0(\omega)\mathbf{1}(\omega \notin A) + x_1(\omega)\mathbf{1}(\omega \in A)$$

belongs to $\mathcal{X}$. A function $f : \Omega \times \mathbb{R} \to \mathbb{R}$ (finite-valued) is a normal integrand, if and only if $f(\omega, x)$ is $\mathcal{A}$-measurable in $\omega$ for each $x$ and is continuous in $x$ for each $\omega$.

**Remark 8.** *A few examples of decomposable spaces are $\mathcal{L}^p(\mathcal{S} \times \mathcal{A}, \Sigma(\mathcal{S} \times \mathcal{A}), \mu)$ for any $p \ge 1$ and $\mathcal{M}(\mathcal{S} \times \mathcal{A}, \Sigma(\mathcal{S} \times \mathcal{A}))$, the space of all $\Sigma(\mathcal{S} \times \mathcal{A})$-measurable functions.*

**Lemma T.2** (([Rockafellar & Wets](), [1998]()), Theorem 14.60). *Let $\mathcal{X}$ be a space of measurable functions from $\Omega$ to $\mathbb{R}$ that is decomposable relative to a $\sigma$-finite measure $\mu$ on the $\sigma$-algebra $\mathcal{A}$. Let $f : \Omega \times \mathbb{R} \to \mathbb{R}$ (finite-valued) be a normal integrand. Then, we have*

$$\inf_{x \in \mathcal{X}} \int_{\omega \in \Omega} f(\omega, x(\omega))\mu(d\omega) = \int_{\omega \in \Omega} \left( \inf_{x \in \mathcal{X}} f(\omega, x) \right) \mu(d\omega).$$

*Moreover, as long as the above infimum is not $-\infty$, we have that*

$$x' \in \arg\min_{x \in \mathcal{X}} \int_{\omega \in \Omega} f(\omega, x(\omega))\mu(d\omega),$$

*if and only if $x'(\omega) \in \arg\min_{x \in \mathbb{R}} f(\omega, x) \mu$ almost surely.*

**Lemma T.3** (Freedman's inequality (e.g., ([Agarwal et al.](), [2014]()))). *Let $\{M_t\}_{t \le T}$ be a real-valued martingale difference sequence w.r.t. filtration $\{\mathcal{G}_t\}$ with $|M_t| \le b$ a.s. and let $S_T = \sum_{t=1}^T \mathbb{E}[M_t^2 \mid \mathcal{G}_{t-1}]$. Then for any $\delta \in (0, 1)$,*

$$\Pr\left( \sum_{t=1}^T M_t \ge \sqrt{2S_T \ln(1/\delta)} + \tfrac{b}{3}\ln(1/\delta) \right) \le \delta.$$

**Lemma T.4** (Robust DE counting bound ($\phi$-residual class)). *Fix $h \in [H]$. Let $\Xi = \Xi_h^\xi$ be a function class on $\mathcal{X} := \mathcal{S} \times \mathcal{A}$ and let $\Pi = \Pi_h$ be a family of probability measures on $\mathcal{X}$. Suppose a sequence $\{\xi_k\}_{k=1}^K \subseteq \Xi$ and $\{\mu_k\}_{k=1}^K \subseteq \Pi$ satisfies that for all $k \in [K]$, $\sum_{t=1}^{k-1} \left( \mathbb{E}_{\mu_t}[\xi_t] \right)^2 \le \beta$. Then for any $\varepsilon > 0$ and any $k \in [K]$,*

$$\sum_{t=1}^k \mathbf{1}\big\{ |\mathbb{E}_{\mu_t}[\xi_t]| > \varepsilon \big\} \le \left( \frac{\beta}{\varepsilon^2} + 1 \right) \dim_{\mathrm{DE}}(\Xi, \Pi, \varepsilon). \tag{88}$$

*Proof.* The proof follows the same argument as in Jin et al. (2021)[Proposition 43]. We include it for completeness.

Fix $\varepsilon > 0$.

**Step 1: Disjoint dependence.** Fix any $k$ such that $|\mathbb{E}_{\mu_k}[\xi_k]| > \varepsilon$. By the definition of DE dimension in Definition 3, if $\mu_k$ is $\varepsilon$-dependent on a subsequence $\{\nu_1, \ldots, \nu_\ell\}$ of $\{\mu_1, \ldots, \mu_{k-1}\}$ (with respect to $\Xi$), then there exists some $\xi \in \Xi$ (in particular we may take $\xi = \xi_k$) such that

$$\sum_{t=1}^\ell \left( \mathbb{E}_{\nu_t}[\xi_t] \right)^2 \ge \varepsilon^2.$$

Therefore, if $\mu_k$ is $\varepsilon$-dependent on $M$ *disjoint* subsequences of $\{\mu_1, \ldots, \mu_{k-1}\}$, then summing the above inequality over these $M$ disjoint subsequences yields

$$\sum_{t=1}^{k-1} \left( \mathbb{E}_{\mu_t}[\xi_k] \right)^2 \ge M \varepsilon^2.$$

Combining with $\sum_{t=1}^{k-1} \left( \mathbb{E}_{\mu_t}[\xi_k] \right)^2 \le \beta$ gives $M \le \beta/\varepsilon^2$. Hence, for any $k$ with $|\mathbb{E}_{\mu_k}[\xi_k]| > \varepsilon$, the measure $\mu_k$ can be $\varepsilon$-dependent on *at most* $\beta/\varepsilon^2$ disjoint subsequences of $\{\mu_1, \ldots, \mu_{k-1}\}$.

**Step 2: Pigeonhole argument using** $\dim_{\mathrm{DE}}$. Now consider any sequence $\{\nu_1, \ldots, \nu_\kappa\} \subseteq \Pi$. We show that there exists some index $j \in [\kappa]$ such that $\nu_j$ is $\varepsilon$-dependent on at least

$$M := \left\lceil \frac{\kappa - 1}{\dim_{\mathrm{DE}}(\Xi, \Pi, \varepsilon)} \right\rceil$$

disjoint subsequences of $\{\nu_1, \ldots, \nu_{j-1}\}$. To see this, run the following procedure:

- Initialize $M$ disjoint blocks $B_1 = \{\nu_1\}, \ldots, B_M = \{\nu_M\}$ and set $j = M + 1$.

- For each $j$, if $\nu_j$ is $\varepsilon$-dependent on all blocks $B_1, \ldots, B_M$, we stop.

- Otherwise, pick a block $B_i$ such that $\nu_j$ is $\varepsilon$-independent of $B_i$ and update $B_i \leftarrow B_i \cup \{\nu_j\}$.

- Increase $j$ and continue.

By the definition of $\dim_{\mathrm{DE}}(\Xi, \Pi, \varepsilon)$, each block $B_i$ can be extended by $\varepsilon$-independent insertions at most $\dim_{\mathrm{DE}}(\Xi, \Pi, \varepsilon)$ times. Therefore, the procedure must stop by time $j \le M \cdot \dim_{\mathrm{DE}}(\Xi, \Pi, \varepsilon) + 1 \le \kappa$, which implies the claimed existence of such a $j$.

**Step 3: Combine Steps 1–2.** Fix any $k \in [K]$ and let $\{\nu_1, \ldots, \nu_\kappa\}$ be the subsequence of $\{\mu_1, \ldots, \mu_k\}$ consisting of those indices $t \le k$ for which $|\mathbb{E}_{\mu_t}[\xi_t]| > \varepsilon$. Applying Step 2 to this subsequence implies there exists some $j$ such that $\nu_j$ is $\varepsilon$-dependent on at least $\lceil (\kappa - 1)/\dim_{\mathrm{DE}}(\Xi, \Pi, \varepsilon) \rceil$ disjoint subsequences of its predecessors. By Step 1 (applied to that $j$), the number of such disjoint dependent subsequences is at most $\beta/\varepsilon^2$. Hence,

$$\frac{\kappa - 1}{\dim_{\mathrm{DE}}(\Xi, \Pi, \varepsilon)} \le \frac{\beta}{\varepsilon^2},$$

which yields

$$\kappa \le \left( \frac{\beta}{\varepsilon^2} + 1 \right) \dim_{\mathrm{DE}}(\Xi, \Pi, \varepsilon).$$

This is exactly eq. 88. $\qquad\square$

