# OpenReview forum: "Online Robust Reinforcement Learning with General Function Approximation"
_ICML.cc/2026/Conference — ICML 2026 regular_

### Official Review · Reviewer_jcDS · 2026-03-02

**Soundness:** 3
**Presentation:** 3
**Significance:** 2
**Originality:** 2
**Overall Recommendation:** 4
**Confidence:** 3

**Summary:**

This paper presents an online reinforcement learning (RL) framework designed to handle distributional shift between training and deployment environments. The authors study a purely online distributionally robust RL (DR-RL) setting, where no pre-collected offline data or prior knowledge of the transition dynamics is assumed. The method is supported by a detailed theoretical analysis, introducing an intrinsic complexity measure and establishing regret guarantees under general function approximation. Empirical evaluation is conducted on the CartPole-v1 environment under perturbations. Overall, the paper provides strong theoretical contributions, while the experimental validation is relatively limited.

**Compliance With Llm Reviewing Policy:**

Affirmed.

**Final Justification:**

While the paper itself and the rebuttal presented by the author evolves around theoretical guarantees, I think the paper will benefit from more simulation experiements. Therefore, I tend to keep my original score.

**Key Questions For Authors:**

1. What is the computational complexity of the proposed algorithm? Is it practical for large-scale or real-time applications?

2. How does the method perform on environments beyond CartPole, particularly more complex or high-dimensional benchmarks?

3. Can the proposed framework be extended to continuous-action control settings, and if so, how would the analysis and algorithm change?

**Limitations:**

The authors are encouraged to include a discussion on limitations in the conclusions or the appendix.

**Strengths And Weaknesses:**

Strengths:

1. The problem addressed is important and timely, especially in the context of robust RL under distributional shift.

2. The paper provides strong and detailed theoretical analysis, introducing an intrinsic complexity measure and regret guarantees.

3. The manuscript is well written, clearly structured, and easy to follow despite its technical depth.

Weaknesses:

1. The experimental evaluation is limited to the CartPole environment, without validation on additional or more complex benchmarks.

2. While the theoretical results are strong, the experimental section is relatively limited.

3. More ablation studies are needed to analyze the sensitivity to key algorithmic parameters.

4. The numerical computational complexity and scalability of the method are not clearly discussed.

5. The framework is only evaluated in discrete space settings, without exploration of continuous environments.

---

> ### Author Rebuttal · Authors · 2026-03-30
>
> We thank the reviewer for recognizing the strength of the theoretical contribution and for the detailed review. We address each point below.
>
> **1. W1 \& W2. Experiments.** We appreciate the reviewer's positive evaluation of our theory. The primary contribution of this work is theoretical: establishing regret guarantees for purely online robust RL under general function approximation. Accordingly, the experimental section is designed to validate the algorithmic mechanism rather than to provide a comprehensive, large-scale benchmark study. We thus adopt the CartPole, aiming to validate the core algorithmic mechanism of the proposed framework rather than to provide a broad empirical benchmark. Specifically, they isolate and test the two key components of our method:
> - (1) learning the robust Bellman operator from nominal interaction via the dual-functional formulation
> - (2) integrating this learned operator with optimism-based exploration in an online setting.
>
> We also highlight that CartPole provides a controlled yet nontrivial setting for this purpose. Its state space is continuous, requiring our function approximation technique. Importantly, compared to the tabular robust baselines (e.g., OPROVI-TV) implemented via discretization, our function approximation algorithm performs much better, validating the effectiveness of our method.
>
> On the other hand, implementing our algorithm in more complex environments is straightforward. We have further developed experiments on ArcRobot from OpenAI Gym, which has a continuous state space. We similarly implement our algorithm with neural networks, and the results are shown in https://anonymous.4open.science/r/anonymous-icml26-B6D4. Similarly, compared to baseline DQN, our algorithm remains robust against environment mismatches, validating our theory. We will develop additional experiments and include them in the final version.
>
> **2. Ablation and sensitivity.** We highlight that our current submission already includes partial sensitivity analysis (e.g., Fig. 1 and parameter sweeps). In particular, we sweep both the robustness radius $\sigma \in \\{0.0,0.2,0.3,0.4,0.5,0.6\\}$ and the slack parameter $\beta \in \\{0.0,0.5,1.0\\}$.
>
> We further provide a concise numerical summary that makes the dependence on both $\sigma$ and $\beta$ explicit in Table 1 of https://anonymous.4open.science/r/anonymous-icml26-B6D4. The table reports representative performance under action, force-magnitude, and pole-length perturbations for different $(\sigma,\beta)$ pairs.
>
> *Key takeaways from Table 1:*
> - (i) The robustness radius $\sigma$ is the primary driver of performance: increasing $\sigma$ consistently improves robustness across all perturbation families.
> - (ii) The slack parameter $\beta$ plays a secondary but meaningful role: the optimal $\beta$ depends on the perturbation type, with stricter enforcement ($\beta=0$) preferred under stochastic action noise and moderate slack ($\beta=0.5$) under structured dynamics shifts.
> - (iii) Overall, the sensitivity is structured and interpretable rather than brittle, aligning with the roles of $\sigma$ (uncertainty size) and $\beta$ (constraint enforcement).
>
> In the final version, we will include this ablation in the appendix with a brief discussion and explicit mapping of weak/moderate/severe perturbations, making the dependence on key parameters fully transparent.
>
> **3. Computational complexity and practical applicability.** Both new parts, the duality optimization and fitting step, of our algorithm can be implemented efficiently via gradient descent over neural networks. Thus, their computational complexity depends on the network size, with cost comparable to DQN (please also see our response—Reviewer fRbL, Point 5). To numerically verify this, we report runtimes under CartPole in Table 2 in https://anonymous.4open.science/r/anonymous-icml26-B6D4, which shows $1.5$ times runtime and implies the practical applicability.
>
> **4. Continuous-action extensions.** Although we present our works for finite RMDPs, the framework could extend naturally to continuous-action settings, with modifications to both the algorithm and analysis. Algorithmically, the discrete maximization $\max\_{a\in\mathcal A} f\_h(s,a)$ will be replaced by approximate optimization (e.g., actor parameterization). The dual-based robust Bellman update remains unchanged.
>
> In analysis, our discrete-action arguments will be replaced by complexity control for infinite action spaces (e.g., covering or smoothness assumptions over $(s,a)$). A main additional source of error is the policy optimization error from approximately solving $\max_{a'} f(s', a')$. This error can be absorbed into the completeness condition (Assumption 1) via an approximate completeness assumption, as is standard in continuous-action RL theory. The robust BE dimension framework naturally accommodates this, since it depends on the function class $\mathcal F$ rather than the action space directly.

---

> > ### Author Rebuttal · Reviewer_jcDS · 2026-04-02
> >
> > Thanks for the authors feedback.
> >
> > While the authors clarify that their main contribution is theoretical, verfying the proposed method through experiments is critical. The authors mention that adapting their algorithm to different environments or to continous action spaces is straightforward; however, practical deployment often expose limitations that theoretical guarantees do not.

---

> > > ### Author Response · Authors · 2026-04-02
> > >
> > > We appreciate the reviewer's feedback and overall positive evaluation of our work.
> > >
> > > We first want to clarify that our study focuses on **large-scale robust RL with discrete action spaces** (as we specified in Sec 2.1), and we do not claim any results for continuous robust RL in our paper. We apologize for the earlier unclear claim in our response on the natural extension to continuous action problems, where our major statement on extending to continuous-action settings is **primarily theoretical and conjecture**: continuous-action settings require replacing discrete maximization with approximate optimization (e.g., actor parameterization) and handling the resulting policy optimization error, which is expected to be effective under our algorithm framework. However, continuous control or RL is generally significantly harder, and its full algorithmic realization/validation is non-trivial, and additional algorithm designs like policy gradient are generally needed, which is beyond our paper's scope. We view continuous-action extensions as an important direction for future work, requiring additional algorithmic and theoretical developments, and we will clarify this limitation in the revision.
> > >
> > > However, we agree that empirical validation is important and appreciate the reviewer’s point regarding practical deployment. Our experiments are designed to demonstrate that the method is **practically implementable and robust in non-trivial settings**, including continuous-state environments and diverse perturbation regimes (but not continuous action spaces). We are also developing more comprehensive experiments (including to extend our newly added ArcRobot experiments to more comprehensive settings) and will include these results in the revised version due to the limited discussion period.

---

### Official Review · Reviewer_hvcr · 2026-03-12

**Soundness:** 3
**Presentation:** 3
**Significance:** 3
**Originality:** 3
**Overall Recommendation:** 5
**Confidence:** 2

**Summary:**

This paper revisits the robust RL problem and offers two main generalizations relative to prior work: (i) first, unlike prior papers that consider either offline data or the presence of a generative model, this paper studies the online setting; and (ii) to account for large state-action spaces, the paper considers general function approximators. An UCB-style algorithm is proposed that maintains confidence estimates of the robust Bellman operator and employs the principle of optimism under uncertainty. The main technical innovation is in the creation of these confidence bonuses that drive exploration. The authors also introduce a new structural component called the robust Bellman eluder dimension to characterize the regret of their algorithm.

**Compliance With Llm Reviewing Policy:**

Affirmed.

**Final Justification:**

The rebuttal adequately addressed my main comments, and hence, I have raised my score accordingly.

**Key Questions For Authors:**

No other questions than the ones above.

**Limitations:**

Yes

**Strengths And Weaknesses:**

**Strengths:**

- While robust RL is not a new topic, this paper appears to be the first to simultaneously tackle the challenges of online data and function approximation in this context.

- The paper is well written overall, and does a good job of explaining the high-level ideas leading up to their algorithmic development. While I am not an expert on this specific topic, the core intuition behind the algorithm is coherent, and complies with the general strategy of UCB-based algorithms. The main new technical challenge that is addressed seems to be that of ensuring that the robust optimal value function lies in the estimated confidence set.

- The authors introduce a new structural property for their setting, namely the robust Bellman Eluder dimension, and show that the regret of their algorithm can be characterized in terms of this new object. This is an interesting finding.

I enjoyed reading the paper overall, and I think that the technical contributions are valuable to the theory RL community. I do have some follow-up questions listed below; I would be willing to raise my score if the authors adequately address them.

**Main Comments:**

- [Q1] Since I am not an expert in this area, I am not entirely certain of how much extra work is needed beyond the prior work of Panaganti et al., who introduce much of the dual functional techniques leveraged in this paper. Remark 4 in this regard seems a bit inadequate. What is the major challenge in the analysis in going from a positive regularizer $\lambda >0$ to $\lambda=0$?

Also, could the authors elaborate on the new algorithmic elements in their approach relative to the prior work of Panaganti et al.?

- [Q2] In the introduction of the paper, the authors mention that the online robust RL problem poses an information bottleneck issue: data collected from the nominal environment may not adequately represent the ``worst" probability transition kernel. How is this issue specifically bypassed? Do the definition of the uncertainty sets in Definition 1 help circumvent this issue? It seems that such a definition ensures that all the transition kernels within the uncertainty set are constrained to be not too far from the transition kernel of the nominal MDP. As such, as long as the data collected online can be used to estimate the latter, one can maintain an empirical estimate of the uncertainty set for decision-making. Is this the core intuition?

- [Q3] In the online setting, the data collected is correlated unlike the generative sampling model. How is this issue tackled in the analysis? Typically, one needs some kind of mixing property from the Markov chains induced by the data-generating policies. How do such mixing effects show up in the regret bounds?

- [Q4] Typically, when studying RL algorithms like TD/Q-learning under function approximation, to obtain convergence guarantees, one needs to assume that the resulting operator is contractive. Is the robust Bellman operator considered in this paper also contractive for the parametric function classes considered?

- [Q5] Can the optimization problem in (15) and the confidence set construction in line 8 of Algorithm 1 be implemented efficiently?

- [Q6] The authors mention that their regret guarantee is sharp. However, can a formal lower bound be proven indicating that the robust Bellman eluder dimension is indeed the fundamental measure of complexity for this problem? It seems that the sharpness aspect only comes in when the authors specialize their results to either tabular MDPs or linear MDPs.

---

> ### Author Rebuttal · Authors · 2026-03-30
>
> We thank the reviewer for the overall positive review and thoughtful questions. We address each point below.
>
> **1A. Distinction from (Panaganti et al.,).**  Our novelty lies in two aspects:
> - *(1) Setting:* Prior works study offline/hybrid RL with coverage assumptions, whereas we consider a purely online setting with no prior data, requiring strategically active data collection.
> - *(2) Analysis:* Panaganti et al. (2024) analyze $\phi$-regularized RMDPs with a Lagrangian penalty $\lambda > 0$ as a smoothing parameter. Their analysis fundamentally requires $\lambda > 0$ controlling the dual variables and ensuring contraction-type properties. But setting $\lambda = 0$ does not recover our setting as a special case because the dual variable ranges become unbounded as $\lambda \to 0$, and their error decomposition relies on the regularized objective having stronger smoothness properties that vanish at $0$. Our analysis handles $\lambda = 0$ directly by exploiting the specific structure of divergence duality and establishing uniform approximation guarantees that hold without regularization.
>
> **1B. New algorithmic elements.** Our method differs from prior work in three key aspects:
> - *(1) Optimism-driven exploration via the dual:* Prior works do need exploration due to coverage assumption. In ours, the dual function approximates the robust Bellman operator and induces the global confidence sets that drive optimistic exploration. This coupling between dual learning and exploration is absent in prior work.
> - *(2) Robust BE dimension as the complexity measure:* We introduce the robust BE dimension and show it governs the regret. Prior robust RL works rely on coverage coefficients. Our analysis requires a new regret decomposition (Eq. 28) that cleanly separates exploration error from operator approximation error, and new concentration arguments (Lemma K.2) that account for the evolving data distributions across episodes.
> - *(3) Uniform operator approximation under evolving policies:* Lemma 2 shows that the dual-based operator approximation holds simultaneously for all policies and their induced distributions. The analogous results in Panaganti et al. hold only for the fixed offline distribution.
>
> **2. Information bottleneck.** Closeness of kernels alone does not resolve the bottleneck: distributions with small TV distance can differ in support, causing unreachable states and exponential hardness (Lu et al., 2024). We enforce absolute continuity (Line 146), ensuring all worst-case transitions are supported under nominal kernel (automatic for KL/$\chi^2$, enforced for TV via Assumption 4). Under this condition, the reviewer’s intuition is partially correct: although one can estimate the nominal kernel and construct uncertainty set, it is infeasible in large-scale problems. Instead, our approach is model-free, directly expressing the worst-case value through a dual reformulation, enabling tractable and statistically efficient learning.
>
> **3. Correlated data and mixing; Contraction.** Mixing is generally needed in online learning with infinite single-trajectory data collection, yet in our finite-horizon episodic setting, transitions at each step are conditionally independent across episodes, eliminating the need for mixing. The challenge here is that behavior policies are adaptively chosen based on history, so the data across episodes are not i.i.d. We address this using martingale concentration, without independence/mixing requirements.
>
> Contraction is typically required in infinite-horizon settings with fixed-point iteration; our finite-horizon setting uses stage-wise backward recursion, so no contraction is needed. Moreover, unlike stochastic approximation-based TD/Q-learning, which requires projection to function space, ours follows fitted-style and the completeness ensures inclusion.
>
> **4. Computational tractability.** Both duality optimization and confidence construction can be efficiently implemented in practice (please see response-Reviewer fRbL, Point 5). Numerically, in CartPole, our running time is $~1.5\times$ standard DQN (https://anonymous.4open.science/r/anonymous-icml26-B6D4).
>
> **5. Interpretation of sharpness and lower bounds.** Our regret bounds are near-optimal in structured settings (e.g., tabular, linear). However, a lower bound is hard in general settings:
> - *(1) Non-robust case.* Even in vanilla RL, minimax lower bounds with general approximation are unknown, as they require constructing realizable, Bellman-complete instances with regret scaling in BE dim, which is challenging due to its dependence on policy-induced distributions. Existing results thus focus on upper bounds, with matching lower bounds only in structured settings.
> - *(2) Robust setting.* This is further complicated by (i) value-dependent worst-case transitions, (ii) dependence on policy-induced distributions across algorithms, and (iii) the need to jointly control nominal dynamics, adversarial perturbations, and the function class.

---

> > ### Author Rebuttal · Reviewer_hvcr · 2026-04-02
> >
> > I thank the authors for their rebuttal which has adequately addressed my main comments. I have adjusted my score accordingly.

---

> > > ### Author Response · Authors · 2026-04-02
> > >
> > > Thank you for your positive feedback. We appreciate your time and are glad that our responses addressed your concerns.

---

### Official Review · Reviewer_fRbL · 2026-03-15

**Soundness:** 2
**Presentation:** 3
**Significance:** 2
**Originality:** 3
**Overall Recommendation:** 5
**Confidence:** 4

**Summary:**

This paper introduces an online robust RL algorithm under general function approximation. The approach focuses on $phi$-divergence-based uncertainty sets. It outlines a fitted learning algorithm inspired from the dual formulation of the robust Bellman update. Regret bounds are established without realizability assumption or parameterization, and the problem complexity is measured according to a distributional Eluder dimension term. Under tabular or linear MDP assumptions, the same regret bounds are tight or match sota results.

**Compliance With Llm Reviewing Policy:**

Affirmed.

**Final Justification:**

Strong rebuttal with convincing answers.

**Key Questions For Authors:**

- The motivation for this work is a bit unclear to me. Fitted learning is standard in classical RL under general function approximation, and I don't understand why it doesn't directly extend to robust RL when we interact with the nominal. I understand that there is no concentrability assumption required here, and no coverage (even though, in some sense, the eluder dimension relates to it). Could the authors elaborate on why a direct extension to robust fitted learning is insufficient?

- A fitted learning algorithm appears in [4] for coherent risk-sensitive measures. It looks similar to robust FTL because of the dual robust formulation applying there. I understand that the sample complexity bounds in that work assume finite coverage, finite concentrability. Beyond that, what fundamental obstacles do the authors see in transposing their techniques to robust RL?

- What value does this study bring in terms of computational complexity for robust MDPs?

[4] Yu, Pengqian, William B. Haskell, and Huan Xu. "Approximate value iteration for risk-aware Markov decision processes." IEEE Transactions on Automatic Control 63.9 (2018): 3135-3142.

**Limitations:**

Explicit claims on the limitations of this work are missing.
Numerical experiments are weak: they are applied on one toy domain. It would have been much more convincing to see extensive experiments on large MDPs, showing that the contribution is not only theoretical but also empirical.

**Strengths And Weaknesses:**

**Strengths**

- The paper is well-written and structured. The narrative is easy to follow. I haven't checked the proof of the results, but the theoretical approach is sound.


**Weaknesses**
- The authors make a confusion that I recurrently see during reviews, namely, indifferently talk about robust MDPs or distributionally robust MDPs. Even though mixing the two notions is sound under general state-spaces + $\phi$-divergence uncertainty sets, the present work takes a minimum over transitions probabilities (min over $P\in\mathcal{P}$) as in [Iyengar, 2005], **not** over distributions  of transition kernels $\mu\in \mathcal{U}$ where $P\sim\mu$, as in [Xu and Mannor, 2010]. The two notions are very much related, but require different structural assumptions on the set of interest. At least in the optimization literature, these sets are called under different names (uncertainty set versus ambiguity set).

- The functional optimization and its advantage over the dual (11) is unclear to me. The authors mention previous work from [Panaganti et al, 2022, 2024] but their explanation lacks some argumentation besides the citation.

**Typos/minor comments**
- A formal definition of $\phi$ divergence would be appreciated in the text.
- l. 131 should include the subscript $D_{\phi}$ in the uncertainty set, even though the authors immediately claim that they will focus on $D=D_{\phi}$ from there on. Otherwise, it looks inconsistent with the superscripts $\mathcal{U}^{\phi, \sigma}$ appearing earlier.
- Eq. (10) shouldn't it be $-\nu$ at the numerator?
- Should replace "the optimal policy" with "an optimal policy" (there can be many, even though all of them reach the same optimal value, ie the max may exist but argmax may not be unique)
- A formal definition of (classical) Eluder dimension would be helpful. So would an explanation/intuition as to how it extends to distributional Eluder.
- Experiments: What is OPROVI-TV?
- Experiments: Other relevant baselines are missing, e.g., [1, 2]

--> More generally, the paper lacks self-containment. Even though I am familiar with the robust RL literature, I found hard to grasp the technical concepts manipulated there. Citations are provided here and there, without further argumentation or explanation.

[1] Mankowitz, Daniel, et al. "Learning robust options." Proceedings of the AAAI Conference on Artificial Intelligence. Vol. 32. No. 1. 2018.
[2] Derman, Esther, et al. "Robustness and regularization in reinforcement learning." NeurIPS 2023 Workshop on Generalization in Planning. 2023.

---

> ### Author Rebuttal · Authors · 2026-03-30
>
> We thank the reviewer for careful reviews and will revise for better clarity and exposition. Key concerns are addressed below.
>
> **1. (Distributionally) robust MDPs.**  We apologize for the misuse of terminology. Our setting is a robust RL as in (Iyengar, 2005). While “distributionally RMDP” is often used to refer to both formulations in RL (e.g., Panaganti et al., 2022; He et al., 2025), we will clarify this and use robust RL instead.
>
> **2. Functional optimization beyond the pointwise dual form.** Pointwise optimization in Eq. 11 requires solving duality optimization for $SA$ pairs in tabular setting, which is computationally expensive and statistically inefficient in online settings due to independent estimation for each (s,a). Our formulation (Lemma 1) lifts this to a *single global ERM* over $\mathcal G$, yielding:
> - *(i) Computational gains:* we only need to solve one regression instead of  $SA$ optimizations;
> - *(ii) Statistical efficiency:* the functional minimizer provides a single, global approximation to the dual variables across all $(s,a)$ simultaneously, enabling uniform error control (Lemma 2) without SA-dependence. Moreover, the global error quantification enables us to design efficient explorations instead of UCB-typed pairwise comparison.
>
> **3. Fitted learning extends to online robust RL.** The extension is non-trivial due to three key obstacles:
> - *(1) Not directly observable Bellman operator.* Unlike standard fitted methods (e.g., FQI), which estimate the Bellman operator from a single next-state sample, the robust operator involves an infimum over an uncertainty set and cannot be directly evaluated from nominal data. Learning it and quantifying the fitted errors is new to standard fitted learning. We thus introduce the duality functional formulation.
> - *(2) New error decomposition.* Our regret then decomposes into a dual-based Bellman residual controlled by the robust BE dimension and a dual approximation error from learning the operator; yet in standard fitted, there only exists a nominal Bellman residual, and it can be directly quantified with nominal concentration.
> - *(3) Coupled confidence sets for online learning.* Unlike non-robust fitted learning, where confidence sets depend only on data collection, our setting must additionally account for uncertainty in the dual function $\underline{g}_f$. This coupling necessitates a new analysis based on the robust BE dimension, where we need to quantify and reduce both uncertainties.
>
> **4. Relation to [4].** While [4] uses dual formulations for risk-sensitive MDPs, two key differences prevent direct extension:
> - *(i) Different Settings:* [4] assumes a generative model with i.i.d. sampling and relies on coverage conditions to control approximation errors. In contrast, we study the online setting, where agent needs to collect and learn from trajectory-wise data actively. Free exploration with comprehensive coverage as in [4] will lead to a large regret, thus our techniques to control and analyze regret are fundamentally different and new from [4].
> - *(ii) Dual formulation and operator learning:* [4] uses pointwise scalar dual optimization (tabular style) and does not address its computational or statistical challenges. In contrast, we learn the robust operator from trajectory data via a functional lifting of the dual problem with a uniform approximation framework.
>
> **5. Computational value of our study.** Our online setting is strictly more challenging than solving a known robust MDP. At each episode k, Algorithm 1 performs—(i) ERM to compute $\underline g\_f$ on the dataset $\mathcal D^{(k)}\_h$; (ii) confidence set update via the squared dual-based residual. With neural network parameterizations of  $\mathcal F$ and $\mathcal G$, both reduce to SGD with per-step cost $O(|\mathcal D^{(k)}_h| \cdot$(network size)), comparable to standard DQN-style methods. Unlike tabular robust RL ($O(S^2 A)$ Bellman updates), our method scales with the function class complexity rather than |S| or |A|, enabling large-scale settings (Sec. 6, App. C). Compared to non-robust methods (DQN/GOLF), the overhead is modest (one extra network and update), yielding only $~1.5\times$ runtime in CartPole. The runtime result is provided in Table 2 in https://anonymous.4open.science/r/anonymous-icml26-B6D4.
>
> **6. Experiments and Baselines.**
> - OPROVI-TV is a tabular online robust RL baseline (Lu et al., 2024). Note that CartPole is a continuous problem, and thus tabular methods are implemented with discretized space. Our approach directly handles continuous space and achieves better performance.  We also include another continuous tasks—ArcRobot (please see response-Reviewer jcDS, Point 1).
> - Suggested works [1,2] address *hierarchical policies* or *regularization-based* robustness, not worst-case Bellman optimization, and are therefore not directly comparable. We focus on baselines that are online, robust to transition uncertainty, and aligned with the same objective.

---

> > ### Author Rebuttal · Reviewer_fRbL · 2026-04-03
> >
> > I raise my score to 5 as the authors fully addressed my concerns and clarified my misunderstandings.

---

> > > ### Author Response · Authors · 2026-04-03
> > >
> > > We sincerely thank the reviewer for the detailed reviews and raising the score. We are also happy to address your concerns.

---

### Decision · Program_Chairs · 2026-04-30

**Decision:**

Accept (regular)

**Comment:**

This paper studies robust reinforcement learning with general function approximation. The theoretical results show that the proposed optimism-based algorithm achieves sublinear regret when the robust eluder dimension of the function class is small. The authors also provide experiments suggesting that the algorithm is robust to several types of perturbations.

Overall, the reviewers agree that the paper makes a solid contribution to the field. However, it also appears that the technical ingredients used in this work are fairly standard, which makes the theoretical novelty seem somewhat limited. In my view, the paper would be strengthened by more clearly explaining why this problem setting is intrinsically challenging, or alternatively by placing greater emphasis on the experimental section to better demonstrate the practical significance of the contribution.